# Skin-inspired, sensory robots for electronic implants

Lin Zhang[1], Sicheng Xing [2], Haifeng Yin[3], Hannah Weisbecker [4], Hiep Thanh Tran[2], Ziheng Guo[5], Tianhong Han[6], Yihang Wang[1], Yihan Liu[1], Yizhang Wu [1], Wanrong Xie[1], Chuqi Huang[1], Wei Luo[2], Michael Demaesschalck[5], Collin McKinney [5], Samuel Hankley[5], Amber Huang [4], Brynn Brusseau[4], Jett Messenger[7], Yici Zou[4] & Wubin Bai [1] ✉

Drawing inspiration from cohesive integration of skeletal muscles and sensory skins in vertebrate animals, we present a design strategy of soft robots, primarily consisting of an electronic skin (e-skin) and an artificial muscle. These robots integrate multifunctional sensing and on-demand actuation into a biocompatible platform using an in-situ solution-based method. They feature biomimetic designs that enable adaptive motions and stress-free contact with tissues, supported by a battery-free wireless module for untethered operation. Demonstrations range from a robotic cuff for detecting blood pressure, to a robotic gripper for tracking bladder volume, an ingestible robot for pH sensing and on-site drug delivery, and a robotic patch for quantifying cardiac function and delivering electrotherapy, highlighting the application versatilities and potentials of the bio-inspired soft robots. Our designs establish a universal strategy with a broad range of sensing and responsive materials, to form integrated soft robots for medical technology and beyond.

The dynamically changing environments drive living organisms to evolve toward inseparable integration of motor and sensor functions[1–3]. Especially, coherent integration between skeletal muscles and sensory skins in vertebrate animals enables their rational and well-organized cooperation orchestrated by neural systems to execute perceptive actions with intelligence. A diverse variety of receptors (mechano, thermal, pain, and others) embedded in the soft skin gathers and encodes tactile data, which not only guides muscular motions to the optimum but also interprets the environment for enhanced awareness and cognition[4–8]. Such motor-sensor integration established in the biological systems inspires development of intelligent robotic systems mimicking skin softness to safely explore and interact with dynamic, unstructured, and often uncertain environments, particularly when robots interface with biological tissues and

organs to enable precision therapeutics[9–12]. However, existing robots often lack a seamless integration among actuators, sensors, and controllers, that naturally preserve physical softness and biocompatibility[13,14].

Creating such bio-inspired somatosensory soft robots as implants holds promising potential to innovate medical technology, especially in surgery, diagnosis, drug delivery, prostheses, artificial organs, and tissue-mimicking active simulators for rehabilitation[15–19]. Conceptually distinct types of soft robotic implants take the form of shape-morphing and functionalization, are capable of compliance matching biological tissues, retrieving their functional signature, and offering therapeutic treatments[12,15,16,20–23]. For example, an integrated bladder system consisting of interdigitated capacitive sensors capable of continuous bladder volume detection, and a shape memory

[1]Department of Applied Physical Sciences, University of North Carolina, Chapel Hill, NC 27514, USA. [2]Department of Biomedical Engineering, University of North Carolina, Chapel Hill, NC 27514, USA. [3]MCAllister Heart Institute Core, University of North Carolina, Chapel Hill, NC 27514, USA. [4]Department of Biology, University of North Carolina, Chapel Hill, NC 27514, USA. [5]Department of Chemistry, University of North Carolina, Chapel Hill, NC 27514, USA. [6]Joint Department of Biomedical Engineering, North Carolina State University, Raleigh, NC 27606, USA. [7]Weldon School of Biomedical Engineering, Purdue University, West Lafayette, IN 47907, USA. ✉e-mail: wbai@unc.edu

alloy-based actuator with strong emptying force for urine voiding, allows for real-time bladder control[20]. A soft gripper based on a shape memory polymer with integration of silver nanowires and a crack-based strain sensor enables conformable contact with a carotid of swine model and measuring its blood pressure[21]. Taking inspiration from the ventricular teeth of hookworm, thera-grippers made of a metal-polymer hybrid actuator and a drug-eluting patch can latch onto the mucosal tissue inside gastrointestinal GI lumen and extend drug release[22]. The combination of sensing and actuation not only enhances diagnostic and/or therapeutic precision for implants via dynamically modulating the structural interface to targeted tissues, but also enables possibility to become artificial organs that offer both needed structural transformation and physiological functions (e.g., electrical signaling, and hormone secretion)[19,22,24]. Despite the promising progress in soft robotic implants, grand challenges remain in designing materials and manufacturing technologies, to leverage multi-faceted

requirements of device performance, including compliant mechanics to match tissue softness, biocompatibility of constituent materials to ensure implantation safety, structural adaptability to prolong device longevity, and biomimicry to enhance device functionality[13,14,20,25].

Here we present concepts and device designs to achieve untethered, soft robots that follow a biomimicry integration of actuators, sensors, and stimulators, to enable structural adaption and dynamic reconfiguration that minimize tissue damage during implant deployment, release stress at biotic-abiotic interface, increase biocompatibility, and enhance device multi-modal performance with spatiotemporal precision (Fig. 1 and Fig. S1). Demonstrated examples of utilizing such bio-inspired robots include: (i) a robotic gripper that wraps around a bladder to enable coordinated, closed-loop operation of bladder volume evaluation and electrical stimulation to treat underactive bladder, (ii) a robotic cuff that can enclose around a blood vessel for measuring blood flow and pressure, (iii) an ingestible robot

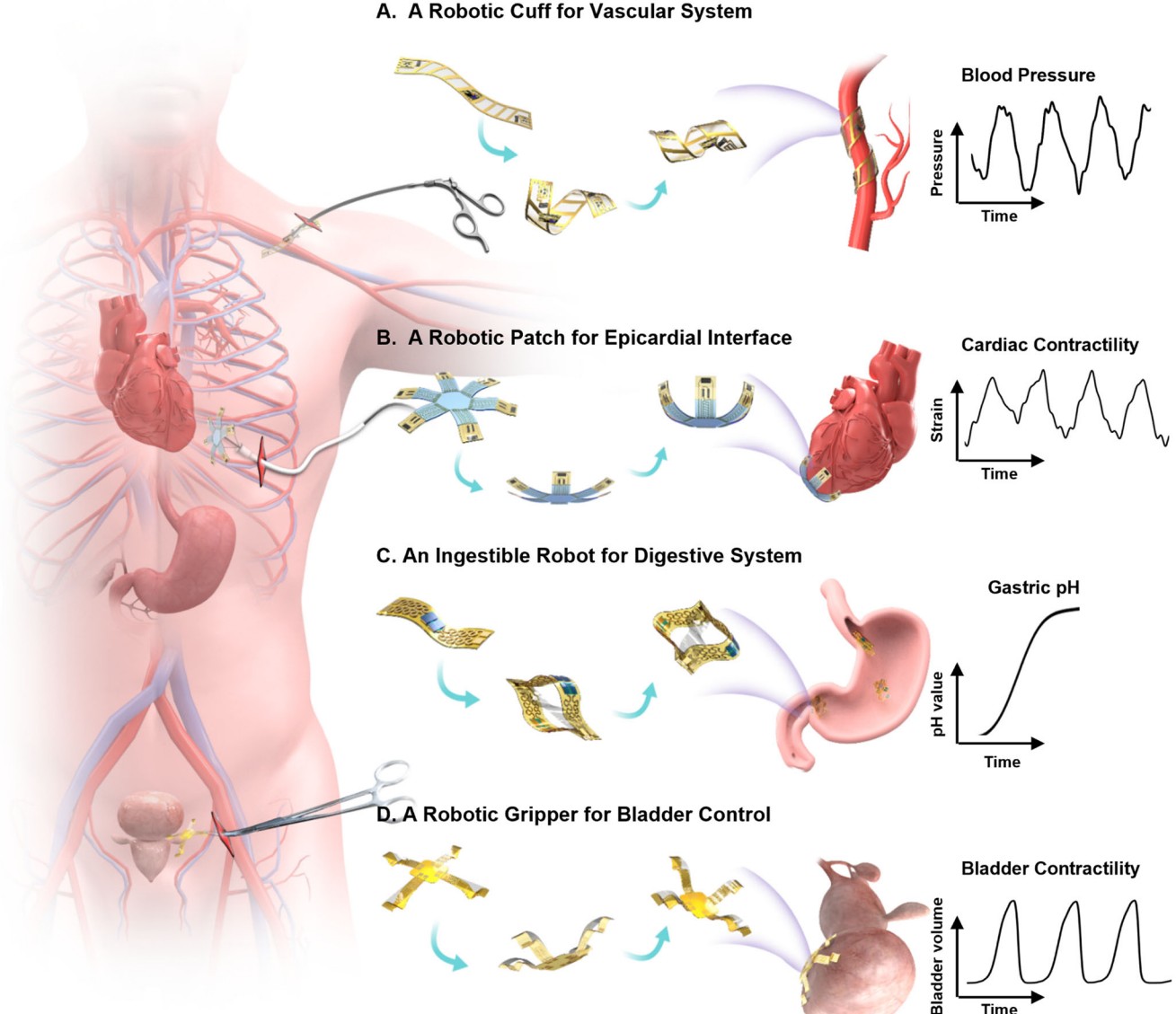

**A. A Robotic Cuff for Vascular System**

Blood Pressure

**B. A Robotic Patch for Epicardial Interface**

Cardiac Contractility

**C. An Ingestible Robot for Digestive System**

Gastric pH

**D. A Robotic Gripper for Bladder Control**

Bladder Contractility

**Fig. 1 | Schematic illustration showing bio-inspired sensory robots as minimally invasive smart implants for diagnosis, stimulation, and drug delivery.** **A** A robotic cuff for vascular system. The twisting motion provides physical enclosing around a blood vessel for precise detection of blood pressure and structural support. **B** A robotic patch for epicardial interface. The gripping motion enables gentle contact with a beating heart without residual straining, to provide real-time

quantification of cardiac contractility and temperature, and electrotherapy. **C** An ingestible robot for digestive system. This structural transition from the shape of a miniaturized pill to a 3D expanded hoop enables extended stay inside stomach to provide both pH sensing and drug delivery. **D** A robotic gripper for bladder control. The adaptive motion of gripping onto a bladder provides precise tracking of bladder volume and targeted stimulation for treating urological disorders.

that can expand when arriving in a stomach for prolonged monitoring and drug delivery, and (iv) a robotic patch that can actively grasp and release a beating heart for epicardial sensing and electrical stimulation (E-stim), which collectively highlight potential impacts of the bio-inspired robotic designs that naturally integrate actuation, sensing and stimulation within a coherent entity as next-generation electronic implants with physical intelligence.

The soft robots primarily consist of two integrated, functional layers that emulate relations between sensory skin and underlying muscles. Specifically, one layer represents an electronic skin (e-skin), that is made of functional nanocomposites predominantly based on an in situ solution-based fabrication approach. The other layer represents an artificial muscle, that is based on poly(N-isopropylacrylamide) (PNIPAM) hydrogel, which can reversibly contract and relax upon activation trigger. Significantly, hydrogels, known for their exceptional softness, low activation temperature and nonfibrotic biocompatibility, are generally preferred over other stimuli-responsive materials in implantable applications[26–29]. The bilayer design composed of the e-skin and artificial muscle represents a heterogeneous configuration with a variety of responsiveness upon exposure to environmental stimuli, which orchestrates its robotic motion. Our in situ solution-based method successfully embeds multiple sensing materials (e.g., silver nanowires -AgNWs, reduced graphene oxide -RGO, MXene and poly(3,4-ethylenedioxythiophene) polystyrene sulfonate -PEDOT:PSS) into a polymer matrix (e.g., polyimide -PI, and polydimethylsiloxane -PDMS), enabling the e-skin a versatile platform that highly mimics the skin with complex receptors, and accurately detects the external signals. These functionalities encompass touch, pressure, temperature, and chemical sensing, surpassing the integrative complexity and heterogeneity achievable with 3D printing or other conventional approaches[30–32]. Moreover, inspired by nature (e.g., starfish and chiral seedpods), we can vary designs of the soft robots, enabling various motions (e.g., bending, expanding, and twisting), and corresponding 3D deformed configurations. In addition, the soft robots allow both on-demand transformation and local-region actuation via embedded control circuits, which further increases structural versatility and capability. Moreover, the soft robots can move, sense, and communicate in a wireless closed-loop fashion via integration of control module and data analytics, enabling minimally invasive operations with safe and stable access to enclosed small spaces inside human body.

## Results

### Bio-inspired multi-modal sensory soft robot

The integrated architecture between the skin and skeletal muscles enables safe and closed-loop interactions with surrounding environment, bringing a feeling of touch and acting of motion seamlessly together in space and time[33–35]. A feature of particular interest in skin is the mechanoreceptors localized at the interface between the epidermis and dermis of skin, which are responsible for detecting a variety of mechanical stimuli, including the fast-adapting (FA) receptors that respond to dynamic forces and slow-adapting (SA) receptors that respond to static pressures (Fig. 2A)[36–38]. By mimicking the hierarchical architectures of skin and muscle with associated biological functions, our soft robots integrate multi-electronic modules and thermally actuatable hydrogels, realizing both receptor-like sensing functions to detect various stimuli, and on-demand muscle-like contraction to generate physically adaptive motion, from a single integrated platform, endowing soft robots with intelligence in navigating through real-world environments autonomously (Fig. 2B). Figure 2C shows a fabricated multi-modal sensory soft robot with geometry emulating a starfish. The robot consists of three primary layers: a flexible nanocomposite layer as a multi-modal electronic-skin (e-skin) embedded with distinct sensors (strain, pressure, pH, and temperature) and stimulators (thermal and electrical), a thermally responsive

hydrogel layer as an artificial muscle generating actuation force, and a thin bio-adhesive layer as a cushioning medium to form interfacial binding between the e-skin and artificial muscle. Our approach for fabricating the flexible nanocomposite layer can be generally applicable to a wide variety of soft materials and nanomaterials heterogeneously composited within a single matrix material, which enables the potential to form highly integrated systems with a broad range of sensors and stimulators. Specifically, here we demonstrate (1) a thermal sensor made of a nanocomposite of reduced graphene oxide (RGO) and polyimide (PI), (2) a strain sensor made of a nanocomposite of silver nanowires (AgNWs) and polydimethylsiloxane (PDMS), (3) sensing and stimulation electrodes made of a nanocomposite of poly(3,4-ethylenedioxythiophene): polystyrene sulfonate (PEDOT:PSS) and PI, and (4) a thermal heater made of a nanocomposite of AgNWs and PI. The flexibility and versatility of this strategy allow the as-fabricated functional units to be easily integrated into each arm of the starfish robot in a monolithic fashion, as illustrated in Fig. 2D, realizing a sensory skin for the robot to enable environmental awareness. Figure 2E displays an example of an ultrathin multi-modal e-skin equipped with six nanocomposite sensors (The detailed fabrication process appears in Fig. S2A.). The e-skin is further encapsulated with a parylene layer (thickness ~2 μm) to enhance its durability during prolonged applications[39,40]. This is subsequently attached onto a piece of predesigned thermally responsive PNIPAM hydrogel that serves as an artificial muscle for the soft robot (Fig. S2B). The PNIPAM hydrogel can undergo a dramatic volumetric reduction of about 90% as the temperature shifts from 25 °C to 60 °C. Notably, this significant and rapid deswelling behavior is initiated only when temperature is beyond its lower critical solution temperature (LCST 32–34 °C), enabling versatile actuation capabilities within biological environments (Fig. S3A, B, and Fig. 2H)[41,42]. Additionally, we can adjust the LCST by incorporating acrylamide (AAm) into the PNIPAM hydrogel to form poly(NIPAM-co-acrylamide) (P(NIPAM-AAM)), enabling us to tailor the LCST to align with varying application scenarios that may require operations at distinct temperature ranges (Fig. S4)[43–45]. The as-fabricated soft robot can actively form a conformal and stress-free interface with curvilinear biological surfaces, indicating its inherent mechanical softness and high biocompatibility. This adaptability minimizes potential risks related to mechanical incompatibility, facilitating its smooth integration with targeted tissues/organs (Fig. 2F, G, and Fig. S3C). At an elevated temperature (40 °C), the robot consisting of the stimuli-responsive artificial muscle (PNIPAM layer) and non-stimuli-responsive e-skin (multi-modal layer) tends to bend toward the muscle side on the basis of asymmetrical responsive properties (Fig. 2H, Fig. S5A–C)[17,46].

Figure 2I, Fig. S5D, and Supplementary Movie S1 provide an example of a robotic starfish reversibly closing and opening its rays under a temperature shift between 23 °C and 40 °C. Furthermore, such multi-layer integration allows a diverse collection of robotic designs that undergo various types of actuations. Fig. S6, 7 and Supplementary Movie S2 highlight soft sensory robots featuring starfish-like structures with an arbitrary number of rays (e.g., four and six) and a fishbone-like structure with four pins. In addition, tuning the design layout of the multi-layer structure can yield complex deformation and structural reconfiguration beyond simple bending motion. Here, inspired by the self-twisting of chiral seedpods, we constructed a geometry with parallel strips in the e-skin layer (Additional details appear in Fig. S8A)[47,48]. Upon thermal stimulation, the PNIPAM hydrogel contracts while the PI maintains stability, generating a differential contraction across the structure. The differential thermal response induces a tilting torque, leading to local saddle-like curvature and twisting motion of the integrated robotic systems (Fig. 2J, Fig. S8B and Supplementary Movie S3). Our findings closely align with the simulation results as shown in Fig. S8C. We also develop a soft robotic pill based on an anchored hydrogel/nanocomposite tri-layer structure, where a hydrogel-muscle

 

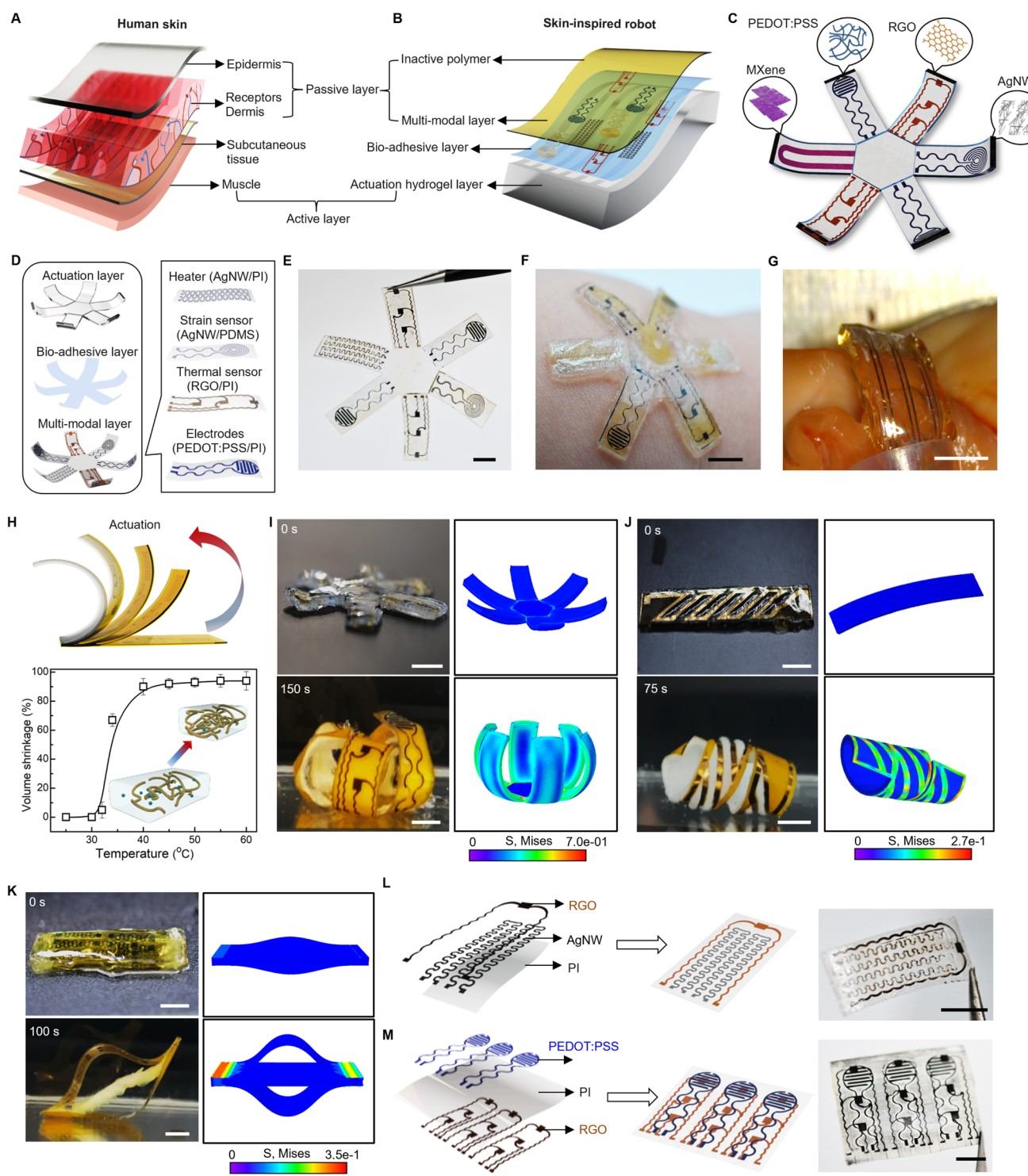

layer is surrounded by two separate e-skin layers that bond to the muscle layer at the edges (Fig. S9A shows the detailed structure design.). Upon thermal actuation, the pill can self-expand into a 3D ring shape, since the contraction of the muscle layer drives the e-skin layers to buckle out of the plane, as shown in Fig. 2K, Fig. S9B, and Supplementary Movie S4. Furthermore, compared with a make-and-transfer method[30–32], our in situ solution-based fabrication approach enables the integration of sensors into the e-skin matrix in a single step, enhancing mechanical and electrical performance by reducing interfacial resistances and improving mechanical conformity, thereby significantly improving sensitivity and responsiveness. As aforementioned, this approach also offers a versatile platform that can be constructed using a broad range of functional nanomaterials hybridized with a polymeric matrix to form a multi-modal sensing system. By selecting materials that offer unique functional attributes, from AgNWs known for their conductivity and flexibility to graphene and MXene for their high surface area, electrical conductivity, and hydrophilicity[49–52], this system provides a possible emulation to the skin that contains complex somatosensory units, where various mechanoreceptors and thermoreceptors distributed in the epidermal and dermal layers enable the spatiotemporal recognition of the magnitude and location of touch and temperature stimuli[53,54]. Fig. S10 provides a representative example of in situ integration of AgNW/PDMS-based strain sensor, in which the strain sensor has a good

**Fig. 2 | Multi-materials Integration for Multi-modal Sensory Soft Robot.**
**A** Schematic illustration of epidermis-dermis-muscle structure of skin. **B** Bio-inspired structure of soft robot from skin. **C** Conceptual illustration of the integrated multi-modal sensory soft robot with distinct nanocomposite sensors functionalized into each arm. **D** Schematic illustration of a starfish-inspired multi-modal sensory soft robot. Left: An exploded view highlighting 3 primary constituent layers, including a flexible multi-modal layer, a bio-adhesive layer, and an actuation hydrogel layer. Right: Schematic illustration highlighting the multi-material integration within the multi-modal layer including: (i) Silver nanowires (AgNWs) and polyimide (PI) as a flexible heater; (ii) AgNWs and PDMS as a strain sensor; (iii) Reduced graphene oxide (RGO) and PI as a temperature sensor; (iv) Poly(3,4-ethylenedioxythiophene) polystyrene sulfonate (PEDOT:PSS) and PI as sensing and stimulation electrodes. **E** Optical image of the flexible e-skin with six nanocomposite sensors. Optical image showing conformal attachment of the soft sensory robot onto human skin (**F**) and porcine tissue (**G**) with high mechanical

compliance. **H** Top: Schematic showing a temperature-responsive bending from a bilayer of poly(N-isopropylacrylamide) (PNIPAM) and polyimide-based nanocomposite. Bottom: Volumetric shrinkage of PNIPAM across a temperature range of 25–60 °C. The data points represent the mean value from $n = 3$ independent experiments, and the error bars are in S.D. **I–K** Optical images and corresponding finite element modeling of thermal-triggered structural reconfiguration of soft robots. Colors indicate von Mises stress magnitude. **I** Biomimicry soft gripper encloses upon heating at 40 °C. **J** Chiral seedpod-inspired robot reverses helix at 40 °C. **K** Soft robotic pill expands upon heating at 40 °C. **L, M** Schematic and optical images of anisotropic integration of various functional materials into a polymeric matrix to form a multi-modal sensing system using an in situ solution-based approach. **L** A RGO/PI-based temperature sensor and an AgNW/PI-based heater on the same polyimide side. **M** PEDOT:PSS/PI-based electrodes and RGO/PI temperature sensors integrated on the two opposite sides of a PI layer, respectively. Scale bars, 5 mm.

response to the stretching. Figure 2L shows the proposed solution-based approach enables a freestanding PI film to integrate multi-functional modalities including an RGO/PI-based temperature sensor and an AgNW/PI-based heater (Fig. S11A), endowing soft robots with both thermal sensing and stimulation. Moreover, Fig. 2M displays a more complicated integration paradigm with multi-layer stacking, where different electronic components (e.g., PEDOT:PSS/PI-based conductive electrodes and RGO/PI temperature sensors) can be distributed in different layers of the e-skin to achieve simultaneous functional versatility and compactness. This assembly technique ensures the e-skin remarkable thinness and flexibility, enhancing its effective performance for implantable applications (Fig. S11B). The X-ray photoelectron spectroscopy (XPS) characterization on the e-skin layers reveals the precise nanoscale integration of active materials within a polymer matrix, as detailed in Figs. S12–S14 and Supplementary Note S1. It showcases the optimal distribution and intermolecular bonding of the composite components, effectively addressing the common challenge of uneven dispersion of nanomaterials, which usually undermines the performance of conventional composites. Our approach minimizes the polymer amount required to integrate nanomaterials into composite functional modules and utilizes excess polymer as an insulating layer to separate modules, preventing interference between their electrical and chemical signals, thereby ensuring that each functional module operates independently and effectively. This simple approach combines the distinct properties of each constituent, achieving a balance between structural integrity and functional versatility[55–59]. This advanced level of integration would be of great value for soft robots that seek to achieve multifunctionality and local sensing capabilities approaching skin.

## On-demand robotic actuation with spatiotemporal control

Programmable stimuli-responsive soft robotic systems capable of working in enclosed or confined spaces and adapting functions under changing situations hold great promise as next-generation medical robots[60,61]. The realization of versatile morphing modes through local-actuation control is crucial for enhancing on-demand actuation. Here, we develop soft robots with cognitive capabilities via unifying programmable actuation and in situ sensing. Figure 3A shows a sensory robotic arm primarily consisting of a PNIPAM-hydrogel-based muscle layer and a multi-modal e-skin layer that is designed with an AgNW/PI nanocomposite actuation heater and a RGO/PI nanocomposite thermal sensor. Fig. S15A shows layer-by-layer stacking as a simple and effective approach for fabricating the e-skin. This approach stands out for its capability to fabricate multi-layered e-skin integrating diverse functionalities within an unified e-skin framework, offering a sophisticated level of device customization. Figure 3B and Fig. S15B demonstrate that the multifunctional nanocomposite film is highly flexible and can be tightly bonded onto the hydrogel layer via n-butyl cyanoacrylate adhesive. Importantly, the bio-adhesive layer exhibits

robust adhesion under both dry and wet conditions, as displayed in Fig. S16. This feature ensures the reliability of implantable devices in the dynamic and moist human body environment. When the liquid PI is cast onto the nanomaterial film (e.g., AgNWs and RGO), the liquid PI penetrates into the interconnected pores of the three-dimensional (3D) network, owing to the low viscosity and low surface energy of the liquid PI[62]. The scanning electron microscope (SEM) images, including both top and side views as shown in Fig. 3C, D, Fig. S17A–F, and Fig. S21A–C, present that the curing process fully buries all the nanomaterials inside the PI matrix, resulting in a uniform composite free from observable voids. This prevents separation of the nanomaterials from the polymeric matrix, thus minimizing untended side effects to the neighboring tissues. The Fourier-transform infrared spectroscopy (FTIR) and X-ray powder diffraction (XRD) results (Fig. S17G, H, Fig. S21D, E, and Supplementary Note S2) also confirm the chemical structure of AgNW/PI and RGO/PI nanocomposites suggesting a successful fabrication of high-quality nanocomposites of functional materials and polymer matrix[63–66]. Figure 3E demonstrates that the resultant AgNW/PI nanocomposite conductor is highly conductive, twistable, and bendable, and thus used as a highly flexible heater for electrothermal actuation. Thermal images in Fig. 3E show the infrared thermograph of a Joule-heated e-skin in the resting state and under both bending and twisting conditions. The electrical heater distributed inside the e-skin exhibits a stable and uniform temperature distribution without degradation of the temperature level at the deformed points, ensuring effective thermal transport from the heater to the hydrogel-based muscle layer. Figure 3F and Fig. S18A depict the transient electrothermal response of the flexible heater applied by various powers in an ex vivo environment. The saturation temperature of AgNW/PI nanocomposite heater increases with the supplied power as more Joule heat is generated, and the lower critical solution temperature (LCST) of PNIPAM hydrogel at 34 °C can be obtained at low input power (<~1 W) (Fig. S18A). As shown in Fig. S18B, the flexible electrical heater exhibits steady heating and cooling cycles, indicating high repeatability and remarkable heating stability using AgNW/PI nanocomposite. Figure 3G shows varying the amount of electric power applied to the heater effectively modulates the resultant bending angle $\theta$ (as defined in Fig. S19A) of the robotic arm, thus realizing precise, on-demand access to intermediate morphologies along the bending pathway. Figure 3G also indicates that the actuator reaches its peak deformation upon achieving thermal equilibrium, and importantly, this maximum bend is maintained as long as there are no changes in thermal conditions. We further evaluated the mechanical force generated by the soft robotic finger which incorporates a PNIPAM hydrogel layer roughly 1 mm thick, under various input powers. Fig. S19B shows that the static force exhibits a noticeable increase with rising temperature. At a temperature of 40 °C, the force reaches a maximum of 32 mN. Additionally, it is observed that the generated force remains consistent throughout 40 cycles of alternating power on

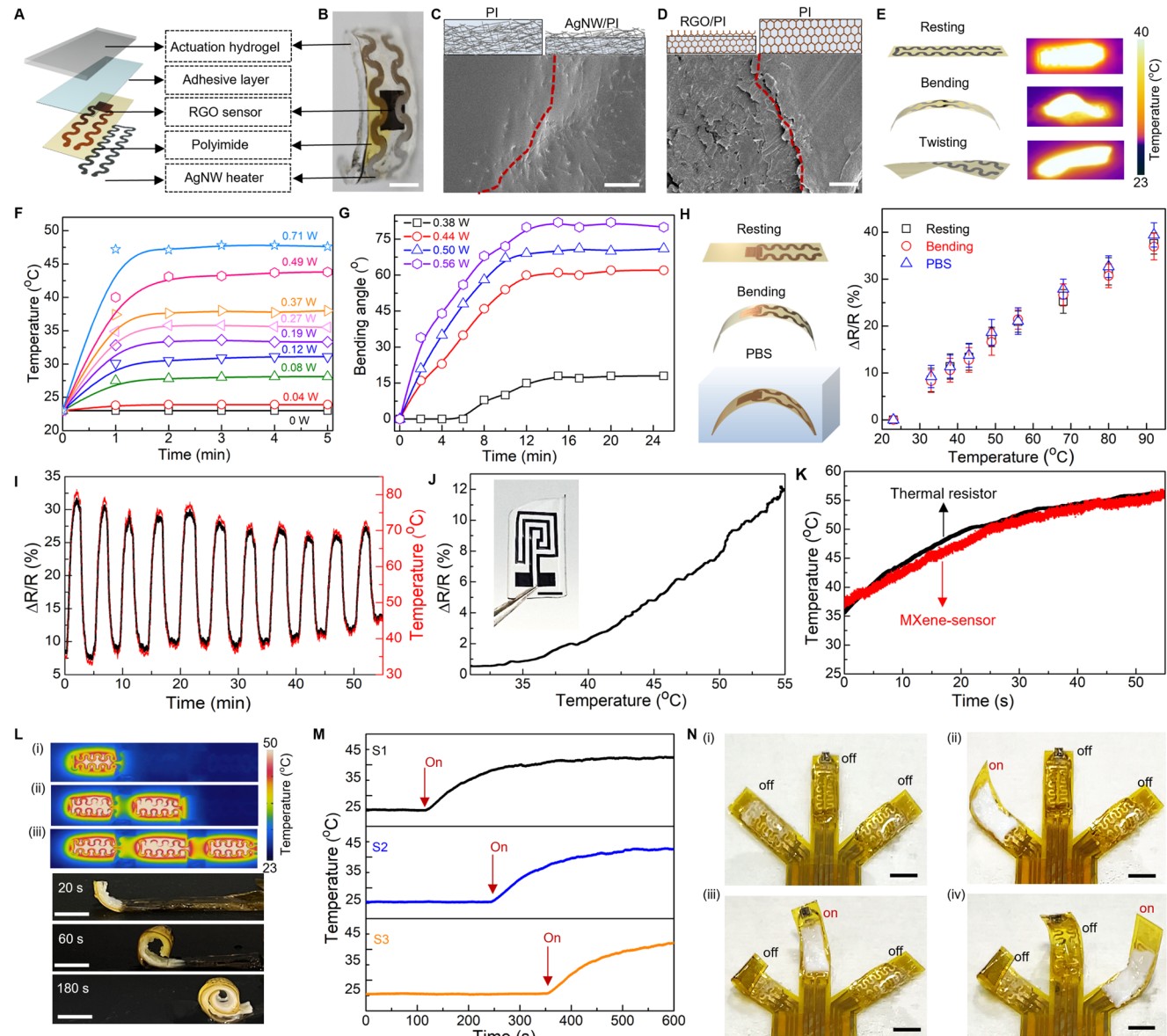

**Fig. 3 | Anisotropic integration of nanocomposites for on-demand robotic actuation with spatiotemporal control.** Illustration of an exploded view (**A**) and optical image (**B**) of a soft robotic arm comprising a PNIPAM hydrogel layer, a RGO/PI thermal sensor, and an AgNW/PI actuation heater, bonded using n-butyl cyanoacrylate adhesive. Joule heating from the AgNW/PI heater triggers bending of the robotic arm. Scale bar, 5 mm. **C**, **D** SEM images of nanocomposite film surfaces. Here, deep reactive ion etching (DRIE) of partial regions reveals the anisotropic integration within the films, distinguishing pristine PI from AgNW/PI (**C**) and RGO/PI regions (**D**). AgNWs and RGO, with higher surface-free energy than PI, are uniformly dispersed inside the PI matrix, enhancing wetting and binding properties. Scale bars, (**C**) 1.5 μm, (**D**) 10 μm. **E** Infrared thermograph of an AgNW/PI heater demonstrating consistent heating performance even after 1000 bending and twisting cycles. **F** Surface temperature of this heater varies with input electric power, functioning efficiently at low power. **G** The bending angle of a soft robotic arm changes with varying input electric power. **H** Resistive response of the RGO/PI thermal sensor during bending, twisting, and PBS immersion, across temperatures from 23 °C to 92 °C. Data points represent mean values from $n = 3$ independent experiments, error bars in S.D. **I** Static cycling test of the RGO/PI thermal sensor. Here, the left y-axis represents resistance changes, while the right y-axis shows corresponding temperature changes. **J** Resistive response of an MXene-based thermal sensor across temperatures from 23 °C to 55 °C. **K** Temperature measurement comparison between the MXene/PI thermal sensor and a commercial thermal resistor (ERT-J0ET102H). **L** Stepwise coiling actuation of a soft robotic probe via sequential thermal stimulus. Top shows infrared thermograph and bottom shows structural changes from a flat to coiled state. Scale bars, 10 mm. **M** Integrated RGO/PI thermal sensors enable temperature measurement of localized regions for proprioceptive sensing. **N** Optical images demonstrating on-demand motion control of a three-arm soft robotic gripper via sequentially programming input power. Scale bars, 5 mm. 3 measurements are repeated with similar results to (**C**), (**D**), (**I**), (**J**), (**K**), (**L**).

and off (0.35 W), indicating the robust reversibility of the soft robot (Fig. S19C). When compared to similar hydrogel-based soft actuators, our design consistently achieves a relatively high output force, as shown in Table S1[67–74]. Moreover, our study demonstrates that the thickness of the hydrogel layer determines the actuation performance of the device. As depicted in Fig. S20A, thicker hydrogel layers induce more pronounced shape morphing at a set activation

temperature, while a slower response due to their increased mass. The thicker layers are also capable of generating higher actuation force, thereby enhancing the actuation capability of the device (Fig. S20B). However, the increased device volume may limit its applicability in minimally invasive implantable devices, where compactness is a key factor. In the device configuration, embedded sensors are strategically positioned within a single, thin e-skin layer, thus minimizing the

required thickness of the hydrogel layer for efficient actuation and allowing the integrated system to be relative compact (Fig. S20C). Our fabrication technique accommodates customization of the hydrogel layer thickness to optimize device performance for specific applications, demonstrating the method's flexibility and adaptability to various biomedical needs. The utilization of a RGO/PI-nanocomposite temperature sensor enables real-time tracking of environmental temperature during muscle motion. The in situ solution processing ensures RGO network is uniformly distributed in PI matrix to provide precise pattern registration with good electrical conductivity (~400 S/m) and stable performance in aspects of electrical linearity, sensitivity, and repeatability. Figure 3H illustrates the resistive change in a relatively linear relation with temperature for the RGO/PI thermal sensor. The temperature coefficient of the resistance (TCR) of the RGO/PI thermal sensor is >0.5%/°C, featuring its high thermal sensitivity. On the other hand, the RGO/PI-based thermal sensor exhibits a stable performance after 1000 bending cycles, and even after immersing in PBS solution. Figure 3I and Fig. S21F show performance of the thermal sensor in response to cycles of temperature rise and drop, indicating good sensing stability. In addition, Fig. S21G shows consistent measurements of RGO/PI nanocomposite sensing performance in comparison with a commercial thermal resistor (ERT-J0ET102H), indicating excellent sensing accuracy. To exemplify this versatility, we have also successfully fabricated MXene-based sensors, as illustrated in Fig. 3J, K and Fig. S22. MXene-based materials possess inherent biocompatibility, rendering them ideally suited for incorporation into implantable biomedical devices without eliciting adverse reactions or compromising the surrounding biological environment[52,75].

Well-controlled shape morphability with high spatiotemporal resolution is essential for soft robots to execute complicated tasks safely in biological environments[76,77]. Here, the integration of e-skin and artificial muscle allows the bio-inspired soft robots to realize motions of local regions independently, which collectively can lead to a wide range of shape morphability. Figure 3L demonstrates the locally controlled morphability with a soft robotic finger, of which the e-skin includes a series of AgNW/PI-based heaters for localized thermal activation, and RGO/PI-based thermal sensors for temperature monitoring (Fig. S23A). As shown in Fig. 3L and Fig. S23B, the robotic finger undergoes a stepwise self-coiling motion via a sequential stimulus and perceives its local temperature simultaneously by the embedded thermal sensors (Fig. 3M). More complex actuation modes or configurations are accessible owing to the combined ease of processing and ability to introduce multifunctional units. Figure 3N presents a three-arm soft robotic gripper where each arm contains various types of sensors, including an optical sensor, a thermal sensor, and a strain sensor, respectively, and an electrical heater (Fig. S24A), to enable independent control of motion for each arm, thus allowing on-demand gripping and interaction with targeted objects (Supplementary Movie S5). Here, the observed variability in bending configuration can be attributed to differences in the positioning and depth of the functional modules within the e-skin layer. Moreover, the soft robotic systems can be structurally tailored to support a range of motions. For example, Fig. S24B, C highlights a soft starfish-like robot with four sensory arms, and a soft robotic cuff with optoelectronic sensors, respectively. The electrothermal stimulus along with distributed sensing capabilities enables programmed actuation not only on demand but also regulated simultaneously by the sensing feedback (e.g., Supplementary Movie S6).

Furthermore, our soft robotic system exemplifies advanced sensory-motor integration, leveraging the synergistic relationship between embedded sensors and actuators to achieve dynamic adaptivity and responsiveness to environmental changes. A prime example is a temperature-sensitive control system, as shown in Fig. S25A, which utilizes real-time sensory feedback to dynamically adjust heating in response to environmental temperature changes. The operational principle, as detailed in Fig. S25B and Supplementary Note S3, involves a microcontroller-driven algorithm that interprets temperature input collected by a resistive temperature sensor, and modulates the electric heater's current accordingly, enabling rapid adaptations to achieve and maintain a preset temperature. Fig. S26 presents a soft robotic finger's real-time response to temperature variations, ensuring stable shape adaptation through this regulatory mechanism. Moreover, this intelligent control significantly improves safety by preventing the risk of overheating, thereby ensuring the system's safe operation in various thermal conditions, highlighting our device's ability to provide precise thermal management, enhancing both efficacy and safety in its applications[78,79].

## Wireless sensing and actuation of soft sensory robot

Wireless operation of actuation as well as therapeutic and/or diagnostic functions is essential for implantable robots to minimize tissue damage and implant infection[76,80–82]. However, existing schemes for wireless operation of robots mostly require sophisticated circuitry for energy harvesting and storage, which may introduce non-negligible heat (~80 °C), unfavored space occupation, system complexity, limited lifetimes, and high cost[83,84]. To overcome the preceding challenges of actuation/sensing integration in soft implantable robots, we report a bio-inspired soft robot as a representative example to demonstrate remote, battery-free operation and communication in both sensing and actuation.

Figure 4A–C show schematic illustrations of the system design consisting of sensor and actuator components. Our strategy for wireless sensing relies on a passive inductor-capacitor (LC) resonance circuit formed by a planar inductor coil and a parallel plate based on capacitor polyacrylamide (PAAm) hydrogel, as illustrated in Fig. 4B. A change in a biomechanical event (e.g., vessel filling, cardiac contraction/relaxion, and bladder filling/voiding) can lead to a corresponding change in capacitance of the hydrogel-based capacitor, which can be quantified by recording shift in resonance frequency ($f_s$) of the LC circuit according to the equation $f_s = 1/2\pi\sqrt{LC}$, where $L$ and $C$ are the inductance and capacitance of the resonance circuit, respectively[85,86]. The inductor coil couples to an alternating electromagnetic field through a readout probe, enabling quantitative measurement of the input return loss (S11) using a vector network analyzer (VNA) (Fig. 4D and Supplementary Note S4). The proposed PAAm-based sensor with relatively low modulus, intrinsic stretchability, and biocompatibility, can detect the pressure through variation of capacitance between the two electrodes. Figure 4F shows a linear correlation between the measured $\Delta C/C_O$ and applied pressure. The slope of the linear fitting curve reports gauge factor (GF) of the sensor reaching ~3%/kPa in the clinically relevant range of pressure (0–4 kPa). Figure 4G&H shows the resonant frequency decreases from 340 MHz to 260 MHz in response to the applied pressure decreasing from 0 to 3.5 kPa, and the pressure sensitivity of the sensor is ~26.7 kHz/kPa.

Moreover, wireless electromagnetic power transmission is an attractive solution to overcome limitations imposed by implantable systems with discrete batteries, especially for devices operating in enclosed places such as the human body[86–88]. Here, we demonstrate the bio-inspired, sensory robot allows for electrically controlled locomotion in this wireless, battery-free manner (Fig. 4A). Figure 4C shows layout of the wireless actuation design including a PNIPAM actuation hydrogel, a flexible electrical heater, and a radio-frequency (RF) power harvester based on a Cu coil. The RF harvester is made of a triple-layer structure (Fig. S27A–C). Magnetic coupling between the transmitting and receiving coil generates electric currents to the heater that thermally stimulates the artificial muscle (Fig. 4E & Fig. S27D). As shown in Fig. 4J and Fig. S28, the robotic finger can undergo a bending motion upon receipt of electromagnetic energy from a power-transfer module. Figure 4I shows an infrared image of the robotic finger highlighting

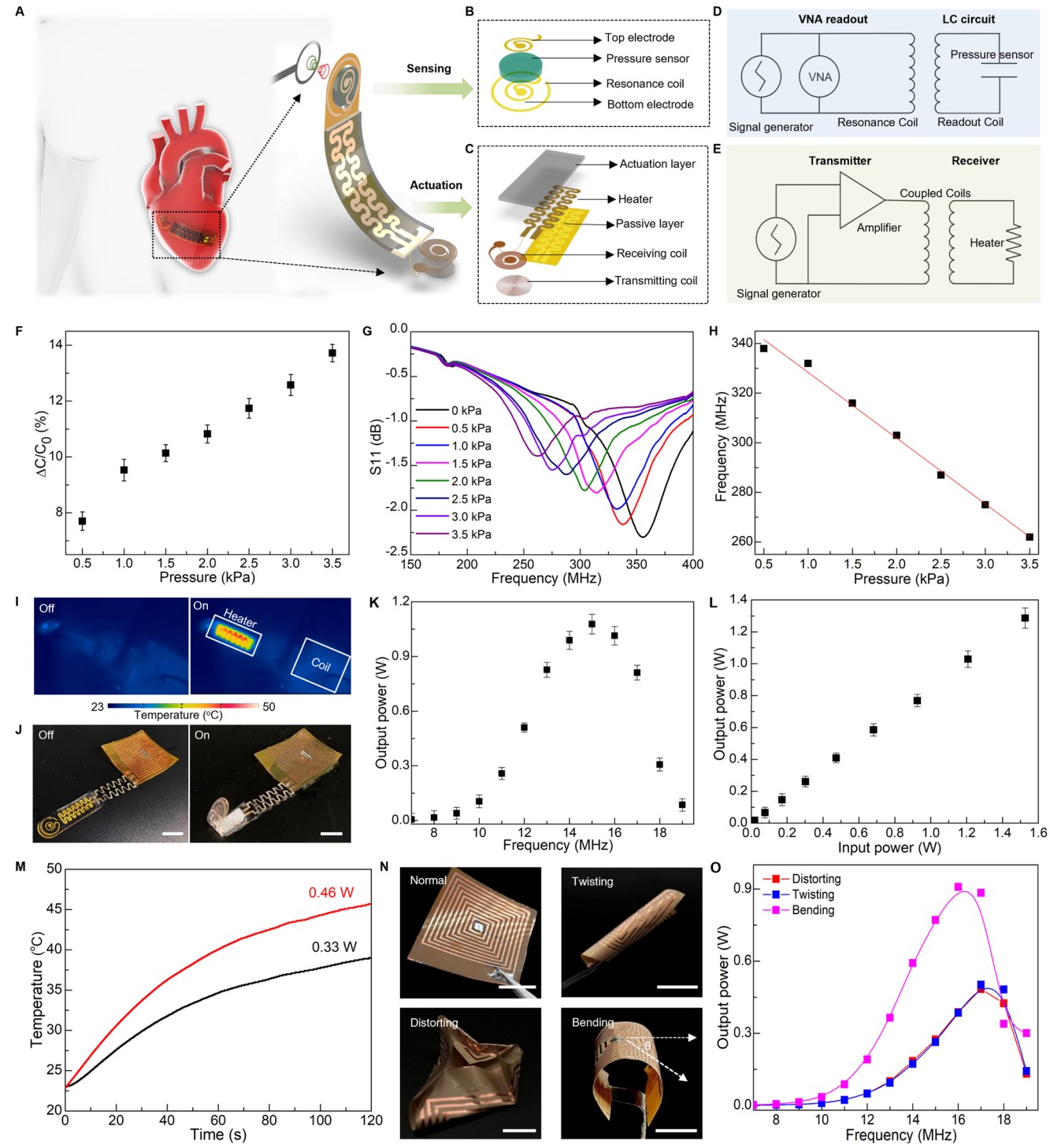

concentrated heat on the electrical heater and minimum heat on the receiving inductor, which collectively ensures a stable power supply and minimum heat damage to surrounding bio-environments in potential usage as implants. Figure 4K shows the output power as a function of frequency. The wireless power transfer system achieves the highest harvested power -1.05 W at a frequency of 15 MHz. Figure 4L indicates the harvested power can be tuned via varying the input power at the optimal frequency 15 MHz, and the power efficiency reaches around 80%, which is sufficient to trigger the heater for raising local temperature to drive motions of the soft robot (Fig. 4M).

The soft robotic system features spatiotemporal locomotion wirelessly controlled by RF signal modulation. To demonstrate the

working principle of this method, we fabricate a soft robotic gripper with three arms as an illustrative example. This strategy employs a three-lead LC receiver coil designed based on the frequency-response characteristics of magnetic resonance coupling (Fig. S29A, B), allowing for the formation of two different configurations with distinct inductive values. These configurations are paired with corresponding capacitors to form LC circuits with different resonant frequencies (Fig. S29C). By coupling with different transmission coils operating at distinct transmission frequencies, the coils can selectively deliver power to specific heaters, thereby enabling the activation of individual robotic arms (Fig. S29D–F). Furthermore, we conduct a further investigation into the influence of shape deformation, including bending,

**Fig. 4 | Design and construction of a soft sensory robot for wireless sensing and actuation. A** Schematic of a soft sensory robot featuring an electrical heater, a polyacrylamide (PAAm)-based pressure sensor, and two inductive coils for transmission sensing signals (**B**) and electrical power (**C**). **B** Exploded view of the sensing components containing a capacitor with two electrodes, a PAAm-hydrogel dielectric layer, and a copper (Cu) inductive communication coil. **C** Exploded view of actuation components consisting of a PNIPAM actuation hydrogel, a flexible electrical heater, and a radiofrequency (RF) power harvester based on a copper coil. **D** Equivalent circuit diagram of wireless pressure sensing, where pressure variations alter the capacitance and, thus, resonance frequency which is captured wirelessly through a vector network analyzer (VNA). **E** Equivalent circuit diagram of wireless actuation, where a transmitting coil connected to an RF power amplifier energizes the receiving coil, powering the heater for robotic motion. **F** Measured capacitive change of the PAAm-based pressure sensor in response to applied pressure. **G** Measured shift of resonance curves of the PAAm-based pressure sensor

in response to applied pressure. **H** Change of the LC resonant frequency as a function of applied pressure serving as a signal-transduction scheme for wireless pressure detection. **I** Thermal distribution during wirelessly harvesting energy exhibits minimal heating in the receiving coil and efficient power consumed by the electrical heater, ensuring minimal heat damage to surrounding bio-environments. **J** Optical images of a soft sensory robot undergoing a wireless actuation to transform from a flat state to a bent state. **K** The output power as a function of frequency, optimized at ~15 MHz. **L** The output electrical power varies with the input power. **M** The temperature change of the electrical heater over time under various output powers used for wireless actuation. **N** Optical images of the deformed RF coils including bending, twisting, and distorting. **O** Measured resonance frequency of RF coil under various shape deformations. Here the bending angle $\theta$ is 60°. Scale bars, 5 mm. Data are presented (**F**), (**K**), (**L**) from $n = 3$ independent experiments, and the error bars are in S.D.

twisting, and distorting, on the transfer performance of the RF harvester (Fig. 4N). Fig. S27E demonstrates that the zeros of the reactance of the receiving circuit remain constant throughout the entire process, indicating minimal disruption to the coil's resonance frequency[89,90]. Additionally, Fig. 4K and Fig. 4O exhibit a reduction in transmission efficiency resulting from decreased mutual inductance between the transmission and the receiving coils due to shape deformation. For all the deformations shown here, the resonance frequency of the coil remains within an acceptable range for power harvesting, and the power transmitted remains above the minimum level required for the robotic actuation. Supplementary Note S5 and Figs. S30–S32 further demonstrate the effect of various design and operational parameters on the power transfer system[91–93]. This yields systematic metrics for designing a well-tuned WPT system, capable of fulfilling the specifications of the targeted application, concurrently decreasing power expenditure, and evading possible risks to living organisms[78,94]. Such wireless robotic systems may serve as a promising solution to safe, real-time monitoring of internal pressure needed for various medical procedures.

## Soft sensory robots interfacing with various internal organs

Soft sensory robots have significant potential for medical-device applications that warrant safe, synergistic interaction with humans[95,96]. Here, we develop robots with the capability to actively morph into 3D configurations to generate a stress-free and stable interface with targeted organs for enhanced sensing, stimulation, and drug delivery. In vitro tests with artificial organ models demonstrate the versatilities and potential applications of our soft robots.

Urinary bladder dysfunction is one of the emerging issues in an aging society, which not only leads to loss of voluntary control over the bladder muscles, but also cuts off sensorial feedback to central nervous system[97]. In most cases of bladder dysfunction, the patients are not able to sense the fullness of bladder with urination, making it a challenge to time the action for voiding treatment (e.g., electrical stimulation)[98]. Therefore, realizing the voiding treatment in a timely manner requires a monitoring system that measures the bladder status continuously.

Here, we develop a soft robotic gripper for both real-time assessment of bladder volume and voiding treatment in a wireless closed-loop control fashion. The as-prepared robotic system, illustrated in Fig. 5A–C and Fig. S33A, B, consists of a flexible hydrogel-based actuator, a 3D buckled strain sensor, an electrical stimulator, and a control module. The actuator includes a passive layer made of a patterned Au/PI bilayer as an electrical heater, and an active layer of PNIPAM hydrogel. Upon an electrical trigger delivered via an inductive coil, the robotic gripper can bend and wrap around the bladder conformally and gently (Fig. 5A) to ensure precise measurement of bladder volume and minimize stress at the interface. The strain sensor integrated into the robot can detect bladder volume continuously.

Here, the strain sensor includes an elastic PAAm hydrogel film and a serpentine Au/PI resistor to form a buckled 3D structure for enhanced sensitivity. The fabrication steps for the buckled sensor appear in Materials and Methods. We use a balloon model to mimic the natural bladder behavior of filling and emptying to validate performance of the integrated soft robotic gripper. Fig. S33C shows the sensor conformally attaches onto the balloon surface with the biocompatible adhesiveness of PAAm hydrogel[99]. The softness of PAAm hydrogel in the robot achieves minimal strain disruption to the bladder during the device operation. Injecting and extracting water into the balloon with a syringe pump realizes the behavior of bladder filling and emptying (Fig. 5D). The change of sensor resistance exhibits a strong linear correlation with bladder volume during its expansion and shrinkage (Fig. 5E and Fig. S33D, E), thus serving as an indicator in bladder-volume control. Figure 5F demonstrates the system repeatability in real-time monitoring during multiple cycles of bladder filling and emptying.

The readout of the bladder volume is achieved through a voltage divider circuit (VDC) consisting of a reference resistor connected in series with the as-fabricated strain sensor. This configuration has minimal impact on the wireless power transmission between coils (Fig. S34). To achieve a closed-loop control of electrical stimulation in the treatment of a dysfunctional bladder system, we further program the Bluetooth-Low-Energy (BLE) System-on-Chip (SoC) (BLE SoC) to enable a pulse-width modulation (PWM) instance. The instance, together with on-board power amplification and filtering circuits, facilitates the application of programmable electrical stimulation in response to an increased strain in the bladder. This enables on-demand electro-therapy and closed-loop control of the robotic implant, as illustrated in Fig. 5B, C. A demonstration of the control scheme can be found in Fig. S35 and Fig. 5G. When the balloon's volume reaches a pre-determined threshold, set here at 100 mL, the control system initiates electrical stimulation. After successful voiding below the threshold, the system automatically deactivates the stimulation. While electrical stimulation has shown promising results in enhancing bladder control in various studies and clinical trials, its efficacy can differ across individuals[100,101]. The effectiveness of electrical stimulation for bladder voiding and its required voltage levels requires further investigation beyond the scope of our current study. However, our prototype showcases the potential of integrating sensing and actuation mechanisms to facilitate timely and adaptive interventions for bladder dysfunction.

The on-demand motion provided by the soft robot facilitates device implantation and ensures benign and stable contact with targeted tissue or organs[12,102,103]. Here we develop a soft robotic cuff that can enclose around a blood vessel upon thermal stimulation (Fig. 5H), to enable real-time measurement of blood pressure. As shown in Fig. S36A, the soft robotic cuff consists of a muscle layer based on a PNIPAM and an e-skin layer embedded with a strain

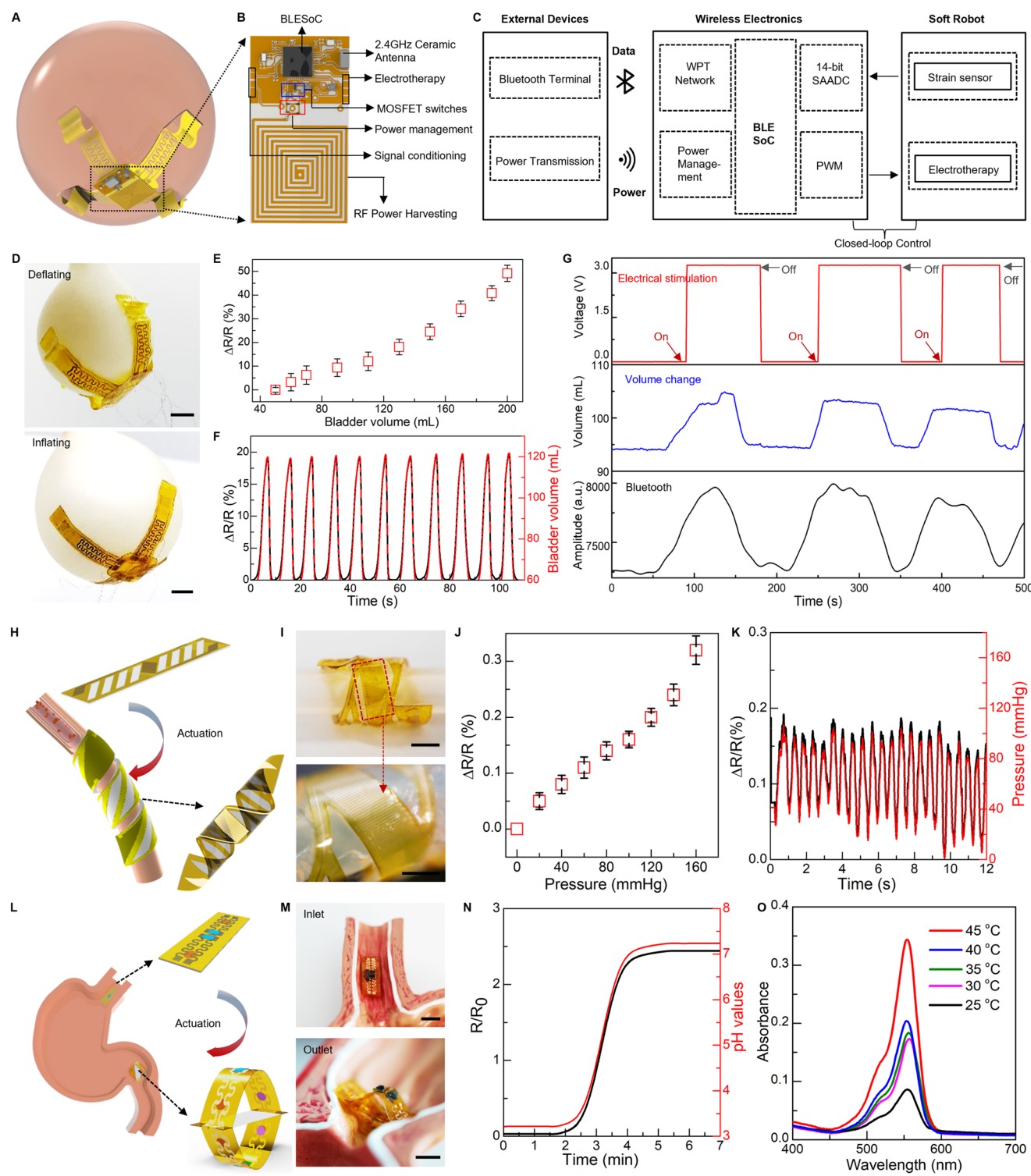

sensor based on a serpentine Au ribbon. Notably, the e-skin layer uses a pattern of parallel strips in the PI film, that forms a 45° angle with the longitudinal direction of the device, which, by coupling with the muscle layers, facilitates formation of a helix structure. An in vitro model of artery that uses a rubbery tube with a pulsatile flow of water to create a simulated pulsation pattern validates performance of the soft robotic cuff (Fig. S36B–D and Fig. 5I). The helix formation of the robotic cuff provides a gentle and stable coupling with the artificial artery, of which the measured signal (resistive change) exhibits a linear relation with the inner fluid pressure (Fig. 5J). Figure 5K demonstrates the capability in continuous

measurement that captures patterns of simulated pulsation, further confirming its utility in measuring blood pressure.

Furthermore, such bio-inspired designs of soft sensory robots can also configure into an ingestible platform. Here, we show that a soft ingestible robot is capable of prolonged residence in the stomach for pH monitoring and drug delivery. Fig. S37A depicts the proposed structure of the ingestible robot based on a slab-shaped muscle layer sandwiched by two e-skin layers consisting of a drug delivery module and a pH sensing module (Fig. 5L, more details appear in the Materials and Methods.). The miniaturized size of the robot facilitates swallowing and transport through the esophagus to stomach (Fig. 5L). Once in

**Fig. 5 | Soft sensory robots interfacing with various internal organs. A** A fully implantable, soft robotic gripper precisely measuring bladder volume and providing electrical stimulation through wireless closed-loop control. **B** The control platform including a wireless power harvesting network, a full bridge amplifier, a voltage regulator, and signal conditioning circuits integrated with a Bluetooth System-on-Chip, and a MOSFET switch. **C** Block diagram for the bladder stimulation module. **D** A soft robotic gripper deployed onto an artificial bladder, demonstrating deflation and inflation cycles. **E** Measured resistive characteristics of the buckled strain sensor on the artificial bladder correlated with volume changes. **F** The 3D buckling strain sensor monitoring real-time volumetric changes during cyclic bladder operations, with resistance and volume changes displayed on dual axes. **G** Programmed electrical stimulation (top) and measured volume of an artificial bladder (middle and bottom). This demonstration is conducted using a volume threshold of ~100 mL, and amplitude of 3 V. A slight delay in the deactivation process could be attributed to the microcontroller unit's response time. **H** A soft robotic cuff enclosing around a blood vessel for monitoring blood pressure. **I** Optical image of the cuff on an artificial vessel stimulating blood circulation. **J** Measured resistive changes of the strain sensor at various simulated blood pressures. **K** Fluidic pressure measurement in an artificial artery system using the soft robotic cuff, displaying resistance changes on the left y-axis and pressure changes on the right y-axis. **L** A soft ingestible robot designed for continuous stomach pH monitoring and extended drug delivery. **M** Optical images showing the robot entering, expanding and blocking in the stomach. **N** Electrical response of PEDOT:PSS/PVA hydrogel to pH change ranging from 3 to 7 over time, with resistance on the left y-axis and pH changes on the right y-axis. **O** Rhodamine-B embedded into the poly lactic-co-glycolic acid (PLGA) matrix to form a drug delivery patch concealed inside the robot. Its release is measured by UV-vis absorbance over an hour at different temperatures. Scale bars, 5 mm. Data are presented (**E**), (**J**) from $n = 3$ independent experiments, and the error bars are in S.D.

the stomach, the robot self-expands that prevents passage through the pylorus for an extended duration in the gastric environment (Fig. 5M and Fig. S37B). The soft pH sensors based on PEDOT:PSS/poly(vinyl alcohol) (PVA) embedded onto the e-skin of the robot provide continuous measurements of gastric pH (Fig. S37C and Fig. 5N). The pH sensor exhibits a linear relationship between the resistive change and pH of the fluidic environment within the range of acidic (pH ~ 3) to basic (pH ~ 8) (Fig. S37E). To realize prolonged drug release, the integrated drug patches use poly(lactic-co-glycolic acid) (PLGA) as a matrix to load drugs and the robotic motion as a trigger mechanism that exposes the drug-loaded patches to stomach fluid for initiating drug release. As a demonstration, we use a biocompatible dye, rhodamine-B (RB), as a model drug (5 mg of RB per 0.5 g PLGA). Figure 5O and Fig. S37H show the UV-vis absorption spectra and the corresponding dosages, respectively, of the RB-loaded drug patch, immersed in PBS solution (at pH ~ 5) for 1 h under various temperatures ranging from 25 °C to 45 °C, (Fig. S37F, G show the calibration curve based on the measured UV-vis absorption spectra.). Such ingestible soft robots highlight a synergistic combination between sensing function and robotic motion to realize on-demand, prolonged control of drug delivery.

Additionally, our device offers versatile adaptability for diverse application scenarios through customizable dimensions, sensor positioning, and geometric layouts, ensuring it aligns with the unique morphologies and functional demands of targeted tissues/organs. This flexibility is essential for enabling minimally invasive deployment. As illustrated in Figs. S38, S39, Table S2, and Supplementary Note S6, the design's adaptability enhances soft robotic technologies for effective integration in a broad spectrum of biomedical applications[104–111]. We also explored the bioadhesive behavior of our device on targeted tissues/organs. We observed that hydrogel's inherent adhesiveness is significantly related to its water content and temperature[112]. As shown in Fig. S40, there is a decline in adhesive strength as temperatures approach the hydrogel's LCST. While this inherent adhesive capability contributes to the initial secure placement of the device, it's noteworthy that solely relying on this property might not guarantee a durable bond, especially as the hydrogel experiences dehydration. However, this temperature-responsive adhesiveness can play a complementary role in enhancing the device's grasp by counterbalancing any potential decrease in force due to hydrogel reswelling.

## A soft robotic thera-gripper for epicardial sensing and electrical stimulating

Cardiac implants that can monitor and regulate heart rhythms are critical for patients with severe cardiac diseases[113,114]. Despite the broad implementation of cardiac implants or pacemakers in clinical settings, however, existing devices are usually rigid, undeformable, and lack structural reconfigurability to adapt to dynamic motions of beating heart, which precludes optimum performance chronically and safely[115].

Here, we report a soft robotic thera-gripper that can gently envelop a heart to perform spatiotemporal monitoring of electrophysiological activity, temperature, and strain, and provide therapeutic capabilities (e.g., electrical stimulating). The soft robotic thera-gripper contains four multi-functional arms, emulating a starfish, to ensure effective latching onto the epicardial surface of which the soft mechanics ensures negligible disruption to cardiac dynamics. Fig. S41A presents an exploded view of the robotic thera-gripper, which contains actuators based on PNIPAM actuation hydrogel, two temperature sensors made of thermal resistors, two electrical stimulating electrodes made of Au, and four strain sensors made of serpentine Au/PI resistors (The fabrication approach appears in Materials and Methods, and Fig. S41B.). Figure 6A shows the thera-gripper at its resting state features minimally invasive insertion with a medical catheter, and forms a bowl shape at the actuation state, via a slightly raised temperature, to gently hold a beating heart for a safe, stable interface (Fig. 6C, D). The design employed PNIPAM hydrogel with a LCST 34 °C that is closely aligned with natural body temperature to achieve necessary shape deformation. The initial heating serves primarily to accelerate the actuation, but after achieving the desired state, continuous electrical heating becomes unnecessary. This feature allows the device to effectively adapt and function within the physiological temperature range without the need for ongoing thermal input. Figure 6B shows the corresponding FEA result of an actuated soft robotic thera-gripper. Moreover, we examined the biocompatibility of the soft robots in vitro and in vivo. Here, mouse 3T3-J-2 cells exposed to complete soft robotic devices with constituent materials including pure PNIPAM hydrogel and functional nanocomposites (e.g., AgNWs, RGO, PEDOT:PSS & MXene), remain robust healthy and maintain stable viability (Fig. 6E, F, Figs. S42, S43)[116,117]. Additionally, histological analysis reveals that the soft robots implanted inside chest chamber of mice induce no observable inflammation or other adverse effects to surrounding tissues, indicating good biocompatibility for long-term operation (Figs. S44, 45). The incorporated temperature sensors provide feedback information on device temperature that govern the structural transformation upon contact with cardiac tissues (Fig. 6G). The E-stim electrodes integrated into our thera-gripper can generate electrical impulses to regulate cardiac functionality (Fig. 6H and Fig. S41C, D). Fig. S46 shows measured ECG traces, highlighting effective transmission of electrical impulses (voltage ranges from 500 mV to 2 V with 1 ms width at 2.65 Hz) onto cardiac tissues. The e-skin layer consists of microelectrodes for capturing electrical activity of the heart, which serves as essential guidance in operating electrical stimulation (Fig. S47). Figs. S48, S49 showcase the simultaneous sensing and stimulation capabilities on a beating heart with an in vivo mouse model, demonstrating its capability in a broad spectrum of potential therapeutic applications[118,119].

Cardiovascular disease is one of the leading causes of death worldwide affecting more than 17.9 million people per year[120].

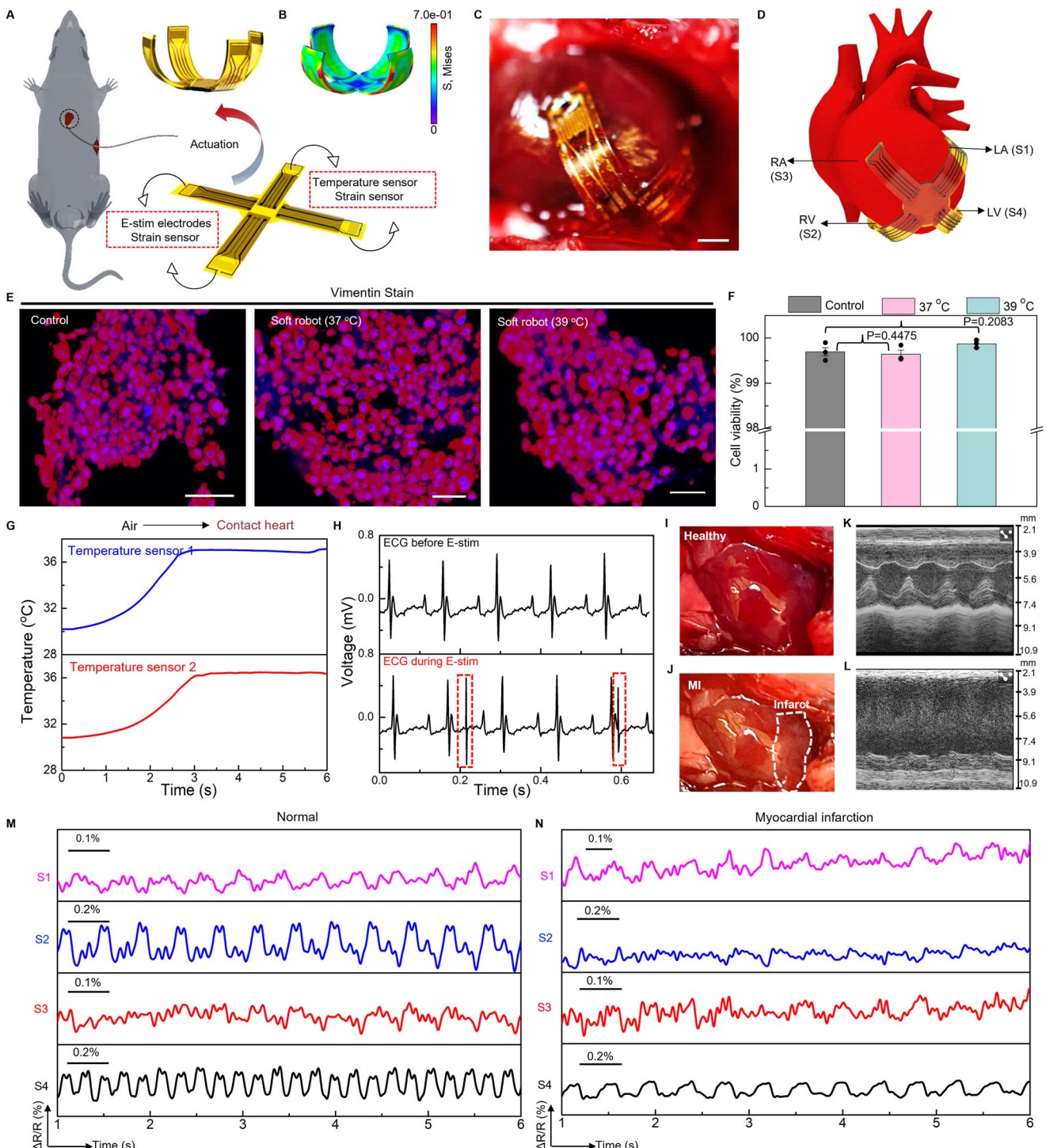

**Fig. 6 | In vivo validation of a soft robotic thera-gripper for epicardial sensing and electrical stimulation (E-stim). A** Schematic illustration showing the thera-gripper features minimally invasive insertion at the resting state and wraps onto the surface of a beating heart at the actuation state. The thera-gripper contains four strain sensors made of serpentine Au/PI resistors, two E-stim electrodes based on Au, and two temperature sensors made of thermal resistors. **B** Finite element modeling of the actuation state. The colors in the legend indicate the magnitude of the von Mises stress. **C** Image of a soft robotic thera-gripper grasping on the epicardial surface of a living mouse heart. Scale bar, 5 mm. **D** Schematic illustration showing the thera-gripper position on a mouse heart, where the strain sensors (labeled as S1, S2, S3, and S4) are located onto different heart chambers for locally monitoring of dysfunctional tissue. **E** Confocal microscope images of 3T3-J-2 cells before (control) and after exposure to as-prepared soft robots integrated with an

e-skin layer and a PNIPAM hydrogel-muscle layer, incubated at 37 °C and 39 °C for 48 h. Scale bars, 50 μm. **F** Comparative cell viability before and after soft robot's exposure, indicating that 3T3-J-2 cells exposed with the soft robot and cultured at the elevated temperature (39 °C) have no decreased viability. Scale bars, 50 μm. Mean ± S.D. $n = 3$. $P$ value by Unpaired $t$-test. **G** Temperature measurements from the thera-gripper during its deployment onto the mouse heart, demonstrate the device's capability to monitor thermal variations in real-time. **H** The surface ECG trace during electrical stimulation using a pair of Au E-stim electrodes. Optical images of a healthy heart (**I**) and an injured heart 2 weeks after myocardial infarction (MI) (**J**). The MI area is shown by the white dashed circle in (**J**). M-mode echocardiographic images from a healthy (**K**) and post-MI heart (**L**). Representative measurements of local cardiac contractions before (**M**) and after MI (**N**) using a soft robotic thera-gripper wrapping onto a living mouse heart.

Real-time and continuous monitoring of myocardial functions (e.g., contractility) is desired for patients with high risks of cardiac arrest, which not only ensures proactive treatment prior to occurrence of adverse events, but also provides comprehensive evaluations of therapeutic effects during postoperative care[121–123]. A soft thera-gripper, integrated with strain sensors distributed on its arms, can provide continuous, spatially resolved quantification of myocardial strains, holding great potential in precision treatment for cardiac diseases. Here, a chronic model of myocardial infarction (MI) with permanent ligation of the left coronary artery (LCA) of a mouse enables assessment of the sensing capability of a thera-gripper post to its robotic motions to establish optimized sensing interfaces (details appear in Materials and Methods.). Figure 6I, J shows the infarcted area that turns into pale myocardium. Echocardiographic strain imaging provides continuous quantification of regional myocardial function. Fig. S50A show the short-axis views in echocardiographic imaging (B-mode) from a normal and post-MI left ventricle (LV) of heart, respectively. The M-mode images, as shown in Fig. 6K, L, display the fraction shortening (FS) decreases from 75% to 15% upon occurrence of MI, indicating the heart exhibiting moderate dysfunction accompanied by regional contractility loss (Supplementary Note S7 and Fig. S50B)[124,125]. Although echocardiography is an invaluable tool allowing for reliable diagnostic information necessary for the clinical decision, it often lacks convenient accessibility and usually takes more than 30 min to capture the movement and function of the heart muscle and heart valves[126]. Figure 6D and Fig. S51A show that the multi-strain sensors (labeled as S1, S2, S3, and S4) of the implanted thera-gripper distributed across four chambers of the heart, respectively, highlighting the key advantage of our thera-gripper with real-time strain sensing over traditional imaging tools in recording local contractions continuously and simultaneously. Fig. S51B shows the response of a representative strain sensor, revealing mechanical rhythms of the cardiac cycles, consistent with ECG recordings. Ligation of coronary artery leads to myocardial infarction that causes ST-segment elevation, as shown in Fig. S51C. Figure 6M, N shows the contractility patterns of the right and left atria (RA and LA), and the right and left ventricles (RV and LV) under normal and MI conditions, respectively, measured by an implanted thera-gripper where the output features of the sensors are determined by their experienced strain that significantly correlates to their positions on epicardial surface. The S4 (LV) displays the largest amplitude due to a maximum experienced strain, indicating the intrinsic highest myocardium strength in the LV chamber. Figure 6J reveals the area of infarction 2 weeks after the MI surgery. The results in Fig. 6N demonstrate that the LV outflow obstruction changes patterns of ventricular contraction. The sensors S2 (RV) and S4 (LV) experience a reduced strain change due to the consequent loss of contractile myocardium, which, correspondingly, reduces force of myocardial contractility and decreases heart rate. Collectively, such bio-inspired design establishes a foundation for implantable robots to harness on-demand motion for structural adaptation inside body and sensory functions for real-time optimization of therapeutic outcomes.

Our post-implantation evaluation revealed that the hydrogel-based thera-gripper remained intact and securely attached to cardiac tissue as intended, demonstrating its durability and effectiveness over time (Fig. S52A). Notably, the E-stim electrodes and thermal sensor maintained optimal performance, effectively delivering electrical stimulation (Fig. S52B, C) and precisely sensing temperature fluctuations (Fig. S52D) even after a 2-week period. These results support its feasibility for long-term therapeutic and diagnostic applications.

## Discussion

In this study, we report concepts and device designs to achieve untethered soft robots that highly emulate biological systems and seamlessly integrate actuators, sensors, and stimulators to enable structural adaption and reconfigurable interfaces that minimize tissue damage in vivo, enhance mechanical match at biotic-abiotic interface, increase biocompatibility, and improve multimodal functionality with spatiotemporal precision. Our soft robots primarily consist of an e-skin layer made of multi-modal nanocomposites that mimic receptors in biological skin to perceive various external stimuli, and an artificial muscle layer made of thermally responsive PNIPAM hydrogel to generate adaptive motion. We employ an in situ solution-based approach to fabricate the flexible multi-modal e-skin. This facile method represents a versatile platform, for a broad range of functional materials (e.g., AgNWs, RGO, and PEDOT:PSS) to be incorporated into a polymeric matrix (e.g., PDMS and PI) to form various types of sensors (e.g., temperature, pressure, and strain) with high spatiotemporal resolution. Such biomimicry design of soft robots offers versatility in mechanical motions including bending, twisting, and expanding as well as diversity in structural deformation, including configurations resembling starfish, fishbone, chiral seedpod, and others. On-demand actuation triggered by electrothermal stimulation from electrical heaters embedded in the e-skin allows precise, independent control of regional parts of the soft body. Furthermore, integration with wireless modules enables the robots to be controlled and communicated without tethering, even when implanted inside body. To demonstrate the broad utility, we develop soft robots that are tailored for specific application scenarios. Specifically, we fabricate a soft robotic gripper that can wrap around a bladder to enable coordinated, closed-loop operation of bladder-volume evaluation and electrical stimulation to treat an overactive bladder, a robotic cuff that can twist around a blood vessel for measuring blood pressure, and an ingestible robot that can reside in a stomach for prolonged pH sensing and drug delivery. In vivo studies with a mouse model demonstrate capabilities of a soft robotic thera-gripper in gently enveloping a beating heart, spatiotemporal assessment of electrophysiological activity, quantification of cardiac contractility, and supplying electrical stimulation for functional regulation. These demonstrations showcase the potential applications of such soft robots as next-generation biomedical implants with structural intelligence and multi-functionalities. Future advancements could further enhance the synergistic interaction between soft implantable robots and biological tissues, to achieve long-term biocompatibility and stability in dynamic physiological environments for improving treatment of chronic diseases.

## Methods

### Materials
N-isopropylacrylamide (NIPAM, 98%) was purchased from TCI. Poly (vinyl alcohol) (PVA, 99 + % hydrolyzed), poly(D,L-lactide-co-glycolide) (Mw 50,000-75,0000), acrylamide (AAm, 99%), N, N'-Methylenebisacrylamide (BIS, 99%), N,N,N',N'-Tetramethyl-ethylenediamine (TMEDA, 99%), and ammonium persulfate (APS, >98%) were purchased from Sigma-Aldrich. Rhodamine B and silver nitrate ($AgNO_3$ 99.9%) was purchased from Themo Scientific. Ethylene glycol (EG, >99%) was purchased from BDH Chemicals. Polyvinylpyrrolidone ($M_W \approx 55,000$, PVP) and poly(3,4-ethylenedioxythiophene)-poly(styrenesulfonate) (PEDOT:PSS, 3.0–4.0%) were purchased from Sigma-Aldrich. Copper chloride ($CuCl_2$) was purchased from Ward's Science. The graphite powder was purchased from Spectrum Chemical Manufacturing Corp. The sodium nitrate ($NaNO_3$, >99%), hydrogen peroxide ($H_2O_2$, 30% w/w), the potassium permanganate ($KMnO_4$, >99%), hydrochloric acid (HCl, 36.5–38%) and sulfuric acid ($H_2SO_4$, 95–98%) were purchased from BDH chemicals. Ethylenediamine (EDA, 99%) was purchased from Alfa Aesar.

### Materials characterization
The SEM images were taken by the field emission scanning electron microscope (Hitachi S-4700 with EDS). The XPS spectra were obtained by Kratos Axis Supra x-ray photoelectron spectrometer, allowing to determine the elemental composition of the top ~10 nm of the sample

surface. The XRD patterns of functional nanocomposites were obtained using the Rigaku SmartLab theta-theta diffractometer. The FTIR spectra were recorded using Hyperion 1000 with Tensor 27 spectrometer. The thermal images were taken with an infrared (IR) camera (FLIR ETS320). The performance of the drug-release patch was characterized with a Ultraviolet-visible (UV-vis) spectrometer (VWR UV-1600PC).

**Multi-modal sensory soft robots with bio-inspired designs.** The fabrication of thermo-responsive hydrogel: Poly(N-iso-propylacrylamide) (PNIPAM) was synthesized based on precipitation polymerization. In a typical example, 500 μL NIPAM as monomer (25 wt%), 50 μL BIS as crosslinker (0.5 wt%), 140 μL APS (1 wt%)/200 μL TEMED (2 wt%) as initiator were mixed to form the precursor solution. After 30 min, the precursors can polymerize, in situ, to form the thermos-responsive PNIPAM hydrogel.

The synthesis of poly(NIPAM-co-acrylamide) (P(NIPAM-AAm)) hydrogel: The synthesis of the PNIPAM-co-PAAm was conducted through a free radical polymerization method. In a typical procedure, a mixture of 450 μL NIPAM monomer (25 wt%) and 50 μL AAm monomer (at concentrations of 5 wt%, 10 wt%, 15 wt%, 20 wt%, and 25 wt%) was prepared. Additionally, 50 μL of BIS crosslinker (0.5 wt%), 140 μL of APS initiator (1 wt%), and 200 μL of TEMED (2 wt%) were added to form the precursor solution. After 30 min, the precursors can polymerize to form the thermos-responsive P(NIPAM-AAm) hydrogel.

The fabrication of AgNWs: AgNWs were synthesized based on a modified polyol method. 100 mL of EG containing NaCl (~0.05 mM), PVP (~189 mM), $AgNO_3$ (~0.0014 mM), and $CuCl_2$ (~0.017 mM) were added to a round-bottom flask and heated at 185 °C for 1 h in an oil bath. Then, 30 mL $AgNO_3$ EG solution (~0.12 M) was added dropwise with vigorous stirring. After the reaction was completed, the flask was cooled to room temperature. The AgNW suspension in the EG was diluted with 30 mL water and sedimented for 14 h. The supernatant was decanted, the water was added to reach 160 mL in total volume of the solution. The suspension was sedimented for 14 h and decanted. Then, 50 mL water was added to this mixture, and 100 mL acetone was slowly added with gentle mixing. After centrifugation, the as-formed AgNW pellets was fully resuspended in a 20 mL PVP aqueous solution (~0.5 wt%). The cleaning process was repeated 4 times. Finally, the AgNW pellets were stored in water for later use.

The fabrication of reduced graphene oxide: The graphene oxide (GO) was synthesized based on a modified Hummers method. In general, 5 g of graphite powder and 2.5 g of $NaNO_3$ were added into 120 mL sulfuric acid in an ice bath under stirring for 2 h. Subsequently, 15 g of $KMnO_4$ was slowly added into the solution under a temperature <20 °C. After 1 h, the reaction temperature was raised to 35 °C for overnight reaction. 150 mL $H_2O$ was added to the mixture for 12 h reaction under 98 °C. Then, 500 mL of $H_2O$ and 20 mL of $H_2O_2$ were added to the mixed solution. Finally, the solution was washed with HCl (1 M) and $H_2O$ until the pH was natural. The EDA was used as the reducing agent to reduce the graphene oxide.

The fabrication of a soft robot inspired by a starfish: The fabrication process of the multi-modal functional electronic-skin (e-skin) is shown in Fig. S2A. A polyimide (PI) substrate (thickness ~ 10 μm) was patterned on a pre-cleaned and plasma-pretreated glass slide using laser-cutter (six rectangular, each is 15 mm × 6 mm). The AgNWs solution (50 wt%), the RGO suspension (5 wt%), and PEDOT:PSS solution were drop cast onto the patterned glass slide and heated at 50 °C for drying. After the solution was dried, the structured functional materials were fabricated with laser cutting (SFX-50GS). The thermal effects of laser patterning ensure minimal residual materials remain, as localized heating effectively eliminates any leftovers, facilitating seamless progression to subsequent processing steps. Then, a thin layer of liquid PI was spin-coated onto the patterned thin film of functional nanomaterials and cured at 150 °C for 1 h. Finally, the resultant film was cutted out with a laser beam and peeled off from the glass slide. The as-formed multi-modal e-skin was encapsulated with a layer of parylene (thickness ~ 2 μm) and bonded onto PNIPAM hydrogels via the adhesive glue (3 M Vetbond 1469c) (Fig. 2C) to generate the multi-modal sensory soft robot with a bio-inspired starfish design. Here, laser hatching parameters for patterning nanomaterials are set: for AgNW/PEDOT:PSS/RGO, an infrared laser power of 12% (50 W) with a hatching speed of 5000 mm/s and frequency of 40 kHz; for Au/MXene, the settings are adjusted to an infrared laser power of 10% (50 W), a hatching speed of 1000 mm/s, and frequency of 40 kHz, ensuring precision in the material's functional structuring.

The fabrication of a soft robot inspired by chiral seedpods: The gold nanomembrane (Au, thickness ~ 200 nm) and adhesive chromium layer (Cr, thickness ~ 10 nm) were deposited by the magnetron sputtering on the PI film (thickness ~ 10 μm). The patterns for Au electronics and PI substrate shown in Fig. S8A were formed using the laser cutting machine. Then the e-skin made of patterned Au/PI film and encapsulated by a parylene layer (thickness ~ 2 μm), is attached to the PNIPAM hydrogel (15 mm × 6 mm) using the bioadhesive layer.

The fabrication of a soft robotic pill: Firstly, the PEDOT:PSS solution was dropcast onto a pre-treated glass slide spacer, dried under 50 °C and patterned using a laser-cutter. Secondly, the PEDOT:PSS/PI nanocomposite thin film was formed by spin-coating liquid PI onto the patterned PEDOT:PSS, and fully curing at 150 °C for 1 h. Next, the e-skin layer received a protective coating of parylene, ~2 μm in thickness. Finally, two pieces of PEDOT:PSS/PI nanocomposite film were adhered to a piece of PNIPAM hydrogel at the edges (Fig. S9A).

**Static finite element analysis for various soft robots.** 3D finite element analyses (FEA) in commercial software ABAQUS were utilized to predict the shape transformation process of soft robots with different patterns and dimensions. The elastic modulus ($E$) and poison's ratio ($v$) used in the simulations were $E_{PNIPAM}$ = 90 KPa, $v_{PNIPAM}$ = 0.30 for PNIPAM hydrogel, and $E_{PI}$ = 2.5 GPa, $v_{PI}$ = 0.34 for PI.

**Mechanical characterization.** Adhesion force was tested by the standard 180° peel test with the Instron machine (Mark-10 ESM303). All tests were conducted with a constant peeling speed of 13 mm/min.

Mechanical tests were conducted on rectangular-shape specimens with the dimensions of 10 mm in width, 2 mm in thickness, and 16 mm in length, using the Instron machine (Mark-10 ESM303).

**The sensory-motor integration within the soft robotic system.** The robotic gripper and the external circuitry were connected in series with an NI DMM amperemeter set for DC current measurement. The device was cooled to 22 °C in ambient temperature before the system was started up to capture its response to a sudden decrease in ambient temperature. The temperature readouts recorded by the device's integrated sensor were logged via a microcontroller unit (MCU) and cross-referenced with data from FLIR thermal camera. Both the current and temperature data were analyzed using custom Python script designed specifically for this purpose.

**Wireless sensing and actuation of soft sensory robot.** The fabrication of polyacrylamide (PAAm) hydrogel pressure sensor: In poly-acrylamide (PAAm) hydrogel synthesis, 500 μL AAm as monomer (25 wt%), 50 μL BIS as crosslinker (0.5 wt%), 140 μL APS (1 wt%)/200 μL TEMED (2 wt%) as initiator were mixed to form the precursor solution. After 10 min, the precursor can polymerize, in situ, to form the PAAm hydrogel. Then a parallel-plate capacitor-based pressure sensor was formed through sandwiching PAAm hydrogel between two electrodes made of Au/PI bilayer (Fig. 4B). Here, to mitigate potential stability issues regarding hydrogel swelling and its impact on pressure sensing, we propose two mitigation solutions. Firstly, we incorporate a protective encapsulation layer around the hydrogel. This layer is

engineered to be permeable enough to facilitate pressure transmission while simultaneously shielding the hydrogel from direct exposure to body fluids that may induce excessive swelling. Secondly, we refine the hydrogel's composition by increasing the crosslinker concentration to diminish swelling sensitivity without impairing the hydrogel's ability to sense pressure. By applying these strategies will enhance the stability of the hydrogel as a pressure sensor for implantable applications.

The fabrication of the soft sensory robot: Laser cutting of the Au/PI bilayer formed the electrical heater that was connected to a Cu receiver coil (Fig. 4C). This assembly was then covered with a thin layer of parylene (thickness ~ 2 μm). Following this, the PNIPAM muscle was bonded to the electrical heater to form the bilayer structure. Finally, the actuation component was integrated with the PAAm-based pressure sensor (Fig. 4A). Here, the circuits for sensing and actuating can be integrated together onto a single piece of Au/PI bilayer film (Fig. S27C).

The characterization of wireless power transmitting and pressure sensing: The detailed power transfer performance and pressure sensing data acquisition are explained in Supplementary Note S4.

Wireless control of locomotion: We implement a three-lead LC receiving network based on the frequency response characteristics of magnetic resonance coupling (Fig. S29A). In this system, two leads are connected to the full length of the coil, while a third electrode is connected to the middle of the coil loops, resulting in the formation of a smaller inductor with the common electrode (Fig. S29B). The two coil loops with different inductive values are paired with different capacitors (Full length ~ 200 pF; Half length ~ 47 pF) to form LC circuits with different resonant frequencies yet similar small quality factors (Fig. S29C). Each soft robotic arm is connected to an individual pair of leads to harvest RF power transmitted at different frequencies.

## LCST tunability of PNIPAM-based hydrogel.

The synthesis of poly(NIPAM-co-acrylamide) (P(NIPAM-AAm)) hydrogel: The synthesis of the PNIPAM-co-PAAm was conducted through a free radical polymerization method. In a typical procedure, a mixture of 450 μL NIPAM monomer (25 wt%) and 50 μL AAm monomer (at concentrations of 5 wt%, 10 wt%, 15 wt%, 20 wt%, and 25 wt%) was prepared. Additionally, 50 μL of BIS crosslinker (0.5 wt%), 140 μL of APS initiator (1 wt%), and 200 μL of TEMED (2 wt%) were added to form the precursor solution. After 30 min, the precursors can polymerize to form the thermosresponsive P(NIPAM-AAm) hydrogel.

Measurement of bending angle of soft robots based on PNIPAM-co-PAAm hydrogel: The bending angles were measured as a function of time under different electrical powers in both a simulated in vivo environment (37 °C, PBS solution) and an in vitro condition (room temperature, PBS solution). The measurements were performed for three different levels of AAm incorporation: 0 wt%, 5 wt%, and 10 wt%.

## A soft robotic gripper for bladder control.

The fabrication of a robotic gripper: The process began with the fabrication of e-skin layer via laser cutting Au/PI bilayer film and applying a parylene coating. The e-skin layer includes electrical heaters, electrical stimulators, and serpentine Au resistors (Fig. S33A). Then, the PNIPAM hydrogel was adhered onto the electrical heater, and the Au/PI resistor was anchored onto one piece of PAAm hydrogel at the edges to form a strain sensor with a buckled structure (Fig. S33C).

Measurement of biomimetic bladder volume: A balloon model was used to mimic the natural bladder to evaluate the actuation and sensing performance of the soft gripper. After thermal stimulation, the soft gripper can wrap around the balloon and the strain sensor can attach onto the balloon surface via the PAAm hydrogel. A syringe pump was used to inject and extract water into the balloon to imitate the filling and emptying behavior of natural bladder. The PowerLab (Model 16/35, AD Instruments) allowed the recording of output signals.

Wireless sensing of resistive strain sensor and close-looped control of bladder electrical therapy robot: A power harvesting and signal conditioning circuit was fabricated and soldered using the method of soft PCB fabrications (Supplementary Note S4). We developed an alternative approach for powering and signal conditioning. This strategy enables the accurate readout of the resistive strain sensor without significantly affecting the wireless power transmission between coils. Specifically, we fabricate a full-bridge rectifier with surface-mount diodes, enabling the conversion of the alternative current (AC) obtained via the receiver coil into direct current (DC). Subsequently, the rectified DC is fed into a 3.3 V low-dropout (LDO) regulator. The output from this regulator serves as the power source for both a Bluetooth-Low-Energy (BLE) System-on-Chip (SoC) and a voltage divider circuit. The voltage divider circuit consists of a reference resistor connected in series with the as-fabricated resistive strain sensor. The voltage divider circuit consists of a reference resistor connected in series with the as-fabricated resistive strain sensor. The voltage across the resistive strain sensor through the voltage divider circuit was sampled by a 14-bit on-chip Successive Approximation Analog to Digital Converter (SAADC). A custom BLE service transmits the sampling value to host terminals acting as BLE clients and parsers to allow wireless readout of strain values. For achieving a closed-loop control of electrical stimulation in the treatment of dysfunctional bladder system, we further programmed the BLE SoC by enabling a pulse-width modulation (PWM) instance. The instance, combined with on-board power amplification and filtering system, enables the programmed electrical stimulation corresponding to detected bladder strain variations.

## A robotic cuff for vascular system.

The fabrication of a robotic cuff: Firstly, we used laser cutter to fabricate the e-skin layer with a pattern of parallel strips that exhibit a 45° angle with the longitudinal direction of the device. The e-skin layer contains a serpentine Au ribbon as a strain sensor (Fig. S36A). Subsequently, this layer was coated with a parylene film ~2 μm in thickness. Finally, the PNIPAM hydrogel-muscle layer was bonded to the e-skin layer.

Measurement of biomimetic blood pressure: As shown in Fig. S36B, to mimic the arterial environment, we employed a silicone tube (Transparent Silicone Tube 4 mm ID × 5 mm OD, wall thickness ~ 0.5 mm) with large stretchability and flexibility as an arterial artery. The pulsatile water flow (30–260 mL/min) is generated with two flow rate controllable pumps. Pump 1 maintains a constant flow rate to establish a baseline pressure, while Pump 2, connected to a solenoid valve, is regulated by a relay. This valve opens and closes periodically, replicating the pulsatile pressure of blood flow. The relay is further controlled with a pre-programmed microcontroller. All the parts for the setup were purchased from local venders.

## An ingestible robot for digestive system.

The fabrication of a drug-releasing patch: In brief, 5 mg poly(lactic-co-glycolic acid) (~PLGA) was dissolved in 5.1 g acetone containing 5 mg Rhodamine B (~RhB). Then the mixture was poured into a poly(dimethylsiloxane) (~PDMS) spacer, and acetone was removed from the mixture under vacuum condition. Finally, the as-formed film of PLGA/RhB was cut into small pieces of drug-release patch with a diameter ~2 mm.

The fabrication of PEDOT:PSS/PVA hydrogel sensor: For hybrid hydrogel fabrication, first a 10 wt% PVA solution was made by dissolving PVA powder in water. Then 5 wt% PEDOT:PSS aqueous dispersion was added into the PVA solution followed by slow mixing for 24 h. The prepared PEDOT:PSS/PVA solution was poured into glass spacer, followed by freezing at −20 °C for 8 h and thawing at 25 °C for 3 h for three times.

The fabrication of a soft ingestible robot: The process began with the formation of e-skin layer. The circuits for sensing were fabricated on Au/PI bilayer film with a laser cutter, while the remaining parts

received a parylene coating. The PEDOT:PSS/PVA hydrogel pH sensors were adhered onto the side of Au/PI film with conductive electrodes, while the drug-release patches were integrated onto the other side. Then, two pieces of e-skin layer were adhered to a piece of PNIPAM hydrogel at the edges, as shown in Fig. S37A.

Quantification of RhB release from the drug patch: The drug-release patches of PLGA/RhB mixture were immersed into phosphate-buffered saline (-PBS) solutions (pH ~ 5) for 1 h under different temperatures ranging from 25 to 45 °C. The UV-vis absorbance spectrometer was used to analyze the amount of the RhB released from the drug-patch.

Measurement of pH sensitivity: The PEDOT:PSS hybrid hydrogels were submerged into PBS solutions with different pH values ranging from 3 to 8. In 5 min, hydrogels were removed from PBS solutions and their resistance was measured with PowerLab. The effect of pH on the electrical properties of the PEDOT:PSS/PVA hydrogel is attributed to the ionic interaction between PEDOT and PSS polymer chains. Under acidic conditions, PEDOT chains are uniformly distributed along the PSS polymer chains, ensuring the formation of continuous electrical connections between PEDOT segments. While pH shifts from acidic to more alkaline, the homogenous distribution of PEDOT along the PSS polymer chains is interrupted by negatively charged hydroxy groups, and buried inside the insulating PSS phase, as illustrated in Fig. S37D.

**Cell morphology analysis, cell viability test, and histological analysis.** Cell morphology analysis: Swiss 3T3-J-2 cells are seeded in a 96-well plate with 10,000 cells per well and cultured for 48 h at 37 °C. Here, the cells are exposed to pure PNIPAM hydrogel, AgNWs/PI nanocomposite, RGO/PI nanocomposite, PEDOT:PSS/PI nanocomposite, MXene/PI nanocomposite, and the integrated soft robot. Notably, each of these functional units was encapsulated with a parylene film (thickness ~ 2 µm). For examining the effect of thermal stimulation on cell viability, the 3T3-J-2 cells with a soft robot are subjected to an environment at 39 °C for a duration of 48 h. To detect Vimentin antigen, chromogenic Immunohistochemistry (IHC) is performed on paraffin-embedded cells that were sectioned at 5 microns. This IHC is carried out using the Leica Bond Rx Autostainer system. Slides are dewaxed in Bond Dewax solution (AR9222) and hydrated in Bond Wash solution (AR9590). Heat-induced antigen retrieval is performed at 100 °C in Bond-Epitope Retrieval solution 1 pH-6.0 (AR9961). After pretreatment, slides are incubated with Vimentin Antibody (5741, Cell Signaling Technologies) at 1:1000 for 30 m followed with Novolink Polymer (RE7260-CE) secondary. Antibody detection with 3,3′-diaminobenzidine (DAB) is performed using the Bond Intense R detection system (DS9263). Stained slides are dehydrated and coverslipped with Cytoseal 60 (8310-4, Thermo Fisher Scientific). A positive control tissue is included for this run. High-resolution acquisition of IF slides is performed with the Aperio Versa 200 scanner (Leica Biosystems Inc.) at an apparent magnification of 20X. Immunofluorescence reaction (IF) is performed on paraffin-embedded cells that are sectioned at 5 microns. This assay is carried out on the Bond Rx fully automated slide staining system (Leica Biosystems) using the Bond Research Detection kit (DS9455). Slides are dewaxed in Bond Dewax solution (AR9222) and hydrated in Bond Wash solution (AR9590). Heat induced antigen retrieval is performed at 100 °C in Bond-Epitope Retrieval solution 1 pH-6.0 (AR9961) for 30 min. After pretreatment, slides are incubated with Vimentin Antibody (5741, Cell Signaling Technologies) at 1:1000 for 30 m. Ready-to use secondary antibody, Leica's Novolink Polymer (RE7260-CE) is used followed by TSA Cy5 (SAT705A001EA, Akoya Biosciences) to visualize the target of interest. Nuclei were stained with Hoechst 33258 (Invitrogen). The stained slides are mounted with ProLong Gold antifade reagent (P36930, Life Technologies). Positive controls are included for each assay. High resolution acquisition of IF slides is performed with the Aperio Versa 200 scanner (Leica Biosystems Inc.) at an apparent magnification of 20X.

Cell viability test: Swiss 3T3-J-2 cells are seeded in a 96-well plate with 10,000 cells per well and cultured for 48 h. Here, the cells are exposed to pure PNIPAM hydrogel, and the integrated soft robot. For examining the effect of thermal stimulation on cell viability, the 3T3-J-2 cells with a soft robot are subjected to an environment at 39 °C for a duration of 48 h. We use the Aperio Cytoplasmic version 2 algorithm on image regions that were annotated to exclude artifacts. The algorithm input parameters included Clear Area Intensity, Optical Density values (RGB) for both counterstain and biomarker detection, intensity threshold values, and minimum/maximum size and smoothing values for cell segmentation. Several of these parameters were adjusted to apply to the specific marker (Vimentin) and cells.

Histological analysis: Explanted organs were bisected and stored in 10% buffered formalin inside 50 mL conical tubes. Following this, these tissue samples were subsequently prepared for histological evaluation with H&E staining and were imaged using Leica Biosystems.

**A soft robotic thera-gripper for epicardial sensing and electrical stimulating.** The fabrication of a robotic thera-gripper: The e-skin layer of the soft robotic gripper is formed using the laser cutting machine, consisting of four strain sensors made of serpentine Au/PI resistors, two Au E-stim electrodes, and two temperature sensors made of thermal resistors, as shown in Fig. S47A, B. Following the application of a parylene film, four pieces of PNIPAM hydrogel were bonded onto the e-skin layer to form the actuators. The soft robotic gripper can gently hold a mouse heart during its beating movement.

In vivo animal experiment: Procedures used in the study were reviewed and approved by the Institutional Animal Care and Use Committee and Research Animal Resources at the North Carolina University Chapel Hill (IACUC ID:21-241.0). Female mice (weight 20–30 g; age, 10 weeks) were purchased from the Jackson Laboratory. The detailed surgery process is described in Supplementary Note S7. The electrocardiography (ECG) and heart function were monitored simultaneously using commercial equipment (PowerLab). The myocardial infarction surgery was conducted by permanently occluding the left coronary artery (LCA). Echocardiography was performed to evaluate cardiac function and histological analysis was performed to assess inflammatory effect of the implantable robotic thera-gripper.

## Data availability
All data needed to evaluate the conclusions in the manuscript are present in the manuscript and/or the Supplementary Information. The source data is provided with this manuscript. Source data are provided with this paper.

## Code availability
The custom codes are available. https://doi.org/10.5281/zenodo.11095018.

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

## Acknowledgements

This work was supported by the start-up funds from University of North Carolina at Chapel Hill and the fund from National Science Foundation (award # ECCS-2139659). We thank Yongjuan Xia in the Pathology Ser-vices Core (PSC) for expert technical assistance with Histopathology/ Digital Pathology including tissue sectioning, immunohistochemical staining, and imaging. The PSC is supported in part by an NCI Center Core Support Grant (P30CA016086). We thank Dr. Amar S. Kumbhar in the Chemistry Department at UNC Chapel Hill for their help with SEM imaging. Research reported in this publication was also supported by the National Institute of Biomedical Imaging and Bioengineering at the National Institutes of Health under award number 1R01EB034332-01. This work was performed in part at the Chapel Hill Analytical and Nanofabrication Laboratory, CHANL, a member of the North Carolina Research Triangle Nanotechnology Network, RTNN, which is supported by the National Science Foundation, Grant ECCS-2025064, as part of the National Nanotechnology Coordinated Infrastructure, NNCI.

## Author contributions

L.Z. and W.B. conceived the ideas and designed the research. L.Z., S.X., H.W., H.T., Z.G., T.H., C.H., Y.Z.W., Y.L., W.L., M.D., S.H., A.H., and B.B. fabricated and characterized the samples. L.Z. performed the mechan-ical modeling and simulation. L.Z., H.W., H.T., Z.G., and H.Y. performed the ex-vivo and in-vivo studies. L.Z., S.X., H.W., Y.L., Z.G., Y.H.W., W.X., Y.Z., J.M., H.T., and C.M. performed the data analysis. L.Z. and W.B. wrote the manuscript with input from all authors.

## Competing interests

The University of North Carolina at Chapel Hill (No. 035052/602726: filed on Oct. 13, 2023) filed a provisional patent application, titled "Skin-inspired, sensory robots for electronic implants", surrounding this work, where W.B. and L.Z. are the co-inventors. The remaining authors declare no competing interests.
