## [Peer Review File · Nature Communications]

Skin-inspired, sensory robots for electronic implantsREVIEWER COMMENTS

Reviewer #1 (Remarks to the Author):

The article entitled "Skin-inspired, sensory robots for electronic implants" by Zhang et al. introduces an approach for soft robots that can sense multiple parameters and provide on-demand motions. Although the proposed concept would be useful for the development of associated technologies, there are many questionable issues and experimental shortcomings that should be addressed before further progress.

The authors need to check the overall figures again. The order of the figures in the manuscript is a total mess. Also, there is so much information about SI that it is very difficult to understand the main text or what the author is trying to say. All unnecessary SI should be discarded, and only the content that is directly related to the key points should be mentioned.

Please understand that the comments have been randomly organized rather than in the order of the manuscript.

1. The authors demonstrated the adhesion performance of the bio-adhesive layer using the bending image in Figure 3B, but this referee can't find any quantified data set of the adhesion force, did the authors studied how strong it is under various conditions? (e.g. dry, wet environments, etc)

2. While the thickness of other layers was specified, the actuation hydrogel's thickness was missing (it seems to be considerably thick). Increase of the thickness might enhance actuation performance but could potentially affect sensing and stimulation properties.

2-1) What is the optimal thickness of the actuation layer for achieving reliable shape-morphing?

2-2) How does the thickness of the actuation layer affect sensing and stimulation performance?

3. In Figure 3N, bending configurations appear to be different and uncontrollable although three arms had the same composition. Can the authors explain why? And, it seems those arms are not reversible since the off-arms don't seem to return to the original state

4. In Figure 4A, there might be stability issues with the PAAM hydrogel as a pressure sensor. Considering the implant environment, swelling of the hydrogel might hinder pressure measurement due to instability in weight loss and capacitance changes. Any proposed solutions to address this concern would be necessary.

5. The authors discussed the adhesion and actuation force of the hydrogel in Fig. S46, Supplementary Note S8. However, it seems to be considered in a dry environment. Given the nature of hydrogels, there might be swelling and degradation issues when operating in a wet environment.

6. In Figures 5 and S47, the hydrogel was designed to face biological tissue (please correct me if not). In this case, some of physical sensors (e.g., strain and pressure gauges) would be fine, but how to figure out operation of other sensors/stimulators that need direct contacts to tissues? For example, the authors mentioned here electrical stimulation for the bladder, but is it possible to stimulate the bladder through the adhesive layer and PAAM hydrogel? That would be not. Can the authors explain how to achieve such functional operations?

7. In Figure 5g, it is unclear what the author intended to demonstrate through the programming. Can the authors provide significance of the programming for the control of the bladder. (It is quite strange that E-stim ON when the bladder is void, and E-stim OFF after expansion/contraction cycle... Shouldn't it turn on when the volume reaches maximum?).

8. In Figs. 6 and S52A, it appears difficult or impossible for the electronic components at the end of each arm to form a strong coupling to living tissues, particularly to tissues whose volume changes over time, such as the bladder. How did the authors control or attach the robotic arms?

9. Page 3, line 86

It is not 'overactive', but 'underactive' bladder.

10. In fig. 2c, is there a motivation for using such diverse materials for different components? This referee can't find outstanding characteristics of those materials for sensing components.

11. In figs. 2f and 2g, the authors claimed that soft robots can form conformal contacts to tissues, but did the authors study or compare mechanical modulus between devices and tissues?

12. In fig. S12B, such layered structures are likely to interfere with each other's signals. is there any reason to measure both parameters at the same time? Or, even if simultaneous measurements are made, does each sensor operate independently?

13. there's no description for Figs. 3H&3I

14. We usually do not include information about SI figures in main figure captions. For example, in fig.3, there is information about fig. S18 and 25, but these should be removed.

15. please do double check all the figures including SI, whose order is a total mess

Reviewer #2 (Remarks to the Author):

Lin Zhang et al. developed various types of soft robots, utilizing an electronic skin (e-skin) made of materials like silver nanowires, reduced graphene oxide, MXene, and conductive polymers integrated into a polymer matrix. This e-skin is designed to mimic skin receptors, enabling the perception of various stimuli. The robots, inspired by natural forms like starfish, chiral seedpod, and others, are equipped with features for bending, twisting, and expanding, and include an artificial muscle for dynamic movement. This study also integrated wireless modules to control and communicate without tethering. Demonstrated in various medical scenarios, these devices showed capabilities such as blood pressure sensing and adapting to organ surfaces for diagnostic and therapeutic functions, exemplifying their potential as multifunctional, minimally invasive medical tools.

However, the devices used materials commonly used in practical applications, with the exception of an in situ solution-based fabrication approach, showing no significant material novelty. This article is also lack of either long-term feasibility or detailed information about the mechanism of each sensor like the relationships between pH, pressure and resistance.

Despite such some disadvantages, the positive aspect lies in its multifunctional ability to be implanted in real scenarios, possibly leading to successful demonstrations for various organ shapes. In specific, massive integration of multifunctional soft robots entailing high-performance functional nanomaterials and actuating modules is highly impressive. Additionally, this article demonstrates a high level of systemic completeness, considering real clinical situations. Therefore, the reviewer believes that with some modifications addressing specific comments included in the following, this paper has the potential to be accepted in Nature Communications.

*Major comments

1. In the introduction part, some references (Nature Materials volume 15, pages937–950 (2016), Nature Nanotechnology volume 9, pages397–404 (2014), and Nature Electronics volume 6, pages779–789 (2023)) regarding stable tissue-interfacing performance of wearable/implantable devices should be cited to justify the authors' research purpose.

2. The reviewer believes that sensory-motor integration is valuable when the robot can control motors in response to signals received from sensors. However, it seems that controlling the motors and performing sensor functions operate independently. In that case, can this robot still be

considered as the presence of responsiveness as mentioned in the introduction?

3. It seems good that the robot is designed differently for each target organ. The reviewer is curious about the variation in curvature among different organs. For instance, in the case of organs like the heart and bladder, squeezing too much might make it challenging to grasp effectively. How was this consideration taken into account in the design?

4. Why does the parallel strips configuration result in twisting? Is there any reference for this, or was it an original concept?

5. On page 6, if the transfer method is well-designed, it could potentially enable multi-modal implementation. Is there a particular advantage in this solution-based approach over transfer methods?

6. On Page 7, Figure 2H, the volume of the PNIPAM hydrogel in Figure 2H is shown to dramatically decrease between approximately 35°C and 45°C. Given that the normal human body temperature is around 36-37°C, even if actuation is induced by an electrothermal method, maintaining its shape seems crucial for attaching it to various organs within the body. To substantiate this point, it would be necessary for the authors to demonstrate volume changes starting from a range lower than 35°C.

7. It was confirmed that the composite was implemented through XPS analysis. Is there a concern that this might result in lower performance compared to a full composite?

8. After nanomaterials are created through the laser pattern, the inherent properties of the material make handling difficult. How was the surrounding material removed or treated?

9. In Figs 3C, D, please indicate in the caption whether SEM images represent the surface or cross-section. Additionally, to determine the presence of voids, it may be necessary to examine cross-sectional TEM images.

10. In Fig 3E, it is understood that accurately observing the exact temperature by performing twisting and bending may be challenging with an infrared camera. However, even considering this, it seems that the temperature distribution is not uniform.

11. In Fig 3F, G, the temperature converges over time, but what is the reason for the continuous change in bending force? Additionally, once bent, how long does the bending force persist?

12. In addition to comment 9, is it necessary to keep the heater on continuously to maintain actual bending when attaching it to an organ?

13. The dual-axis graphs presented in this figure (Figure 3I, Figure 5F, K, N etc.) are difficult to intuitively comprehend. It is unclear whether the changes on the left axis correspond to the results on the right axis or if they were determined through variations in resistance, pressure, or pH values. Therefore, it seems necessary for the authors to modify the graphs for better intuitive understanding and provide explanations regarding the extent of numerical changes made.

14. Using heat as a stimulus might potentially cause damage to the tissue. However, considering the hydrogel is thick enough, it may prevent such damage. How efficiently does heat transfer throughout the entire hydrogel, and is the heat dissipation effective enough to be considered safe?

15. While applying temperature, can the thermal sensor accurately measure the temperature from the tissue?

16. What is the reason for the difference in the transfer method in S12(b) compared to the dual printing method in S16?

17. On Page 11, the author mentioned that "The electrothermal stimulus along with distributed sensing capabilities enables programmed actuation not only on demand but also regulated simultaneously by the sensing feedback". How was actuation implemented based on sensing feedback?

18. On Page 14, Figure S31, in Figure S31B, it is evident that there is a tendency for the coupling coefficient to decrease as the separation distance increases. However, in Figure S31D, there is a trend showing an increase in output power relative to input power as the distance increases. This observation appears to be theoretically inconsistent. If this graph is accurate, the authors need to provide additional explanations for these results.

19. On Page 16, in order for the sensory robots developed by the authors to conformally adhere to tissues or organs, they require not only curvilinear surfaces but also effective adhesion. While this article mentions the adhesiveness of PNIPAM, a recently published paper indicates that PNIPAM shows tissue adhesion at low temperatures (around 25°C), but loses almost all adhesive strength when it transitions to a gel form near 40°C (Nature Biotechnology 2023, 41, 652-662). To showcase the tissue adhesion capability of the authors' device, it becomes imperative to gauge the adhesive strength of the hydrogel by varying temperatures. This would confirm that there are no issues with adhesion even after attaching it to an organ.

20. In Figs 5, 6, the use of Au electrodes for electrical stimulation in electrotherapy seems somewhat divergent from the other sensing materials of this paper (Fig. S52B).
21. In Fig 5G, the electrical stimulation off voltage is not 0.0 V. Is there a specific reason for this? To help readers to understand, it appears that the authors should provide additional explanations regarding the correlation between electrical stimulation and volume changes. Specifically, a clear clarification is needed on whether the specific electrical stimulation is causing an actual increase in volume or if it is transmitting signals to patients having urinary dysfunction.
22. As the authors mentioned in the context, PNIPAM hydrogel has reversibly contract and relax property upon electrothermal trigger. If that's the case, is there no risk of delamination if the device is not well-adhered to the tissue when the internal body temperature decreases? Additionally, wouldn't there be a possibility that continuous electrothermal stimulation, applied to prevent deformation of the device after it is attached to the tissue, could have adverse effects on the tissue?
23. On Page 24, Figure 6F, what is the indication T1 and T2 in Figure 6F account for? The authors need to explain these in the caption.
24. In Fig 6G, it is challenging to consider it as functioning like a pacemaker, as pacing does not seem to lead to actual cardiac capture.
25. In Figs 6 M, N, it was mentioned that changes were observed in S2 and S4. However, it seems that only changes in frequency are visible along the x-axis. Shouldn't the focus be on the resistance changes along the y-axis, which may represent actual strain changes?
26. The paper lacks details on the system used to generate water flow simulating blood flow in the vascular system, including names of the equipment and the setting of flow rate. Information on the properties and thickness of the rubber used is also insufficient.
27. This work validated through H&E staining data that a single-day stimulation has a minimal impact on the tissue. However, it remains unclear what effects may occur with continued stimulation over longer period.
28. The 2-week histological analysis of cardiac tissue with a hydrogel-based thera-gripper, as presented in the study, raises questions. It would be insightful to know the condition of the device after this period –whether the device remained in an operable state and the device remained properly attached as intended while withstanding repeated pulsations.

*Minor comments

1. Caption in figure S11 (a) : AgNW/PI -> AgNW/PDMS
2. Page 21, the third line from the bottom spelling error : PINPAM -> PNIPAM
3. Figure S51D, indication spelling error : Basi -> Basic
4. There should be a space before the units in the text and figures.

Responses to comments of Referee #1

Comments from Referee #1:

Summary Comment: The article entitled “Skin-inspired, sensory robots for electronic implants” by Zhang et al. introduces an approach for soft robots that can sense multiple parameters and provide on-demand motions. Although the proposed concept would be useful for the development of associated technologies, there are many questionable issues and experimental shortcomings that should be addressed before further progress. The authors need to check the overall figures again. The order of the figures in the manuscript is a total mess. Also, there is so much information about SI that it is very difficult to understand the main text or what the author is trying to say. All unnecessary SI should be discarded, and only the content that is directly related to the key points should be mentioned.

Our response: We thank the reviewer for these positive comments and for these helpful suggestions for revision. We carefully addressed the issues, as listed below, and revised our manuscript accordingly.

Modification to the manuscript: We have modified both the manuscript and supporting information based on the reviewer’s concerns and suggestions.

Comment 1: The authors demonstrated the adhesion performance of the bio-adhesive layer using the bending image in Figure 3B, but this referee can’t find any quantified data set of the adhesion force, did the authors studied how strong it is under various conditions? (e.g. dry, wet environments, etc).

Our response: We appreciate the reviewer’s comment regarding the lack of quantified data on the adhesion performance of the bio-adhesive layer. We have conducted 180° peeling tests under both dry and moist environments at physiological temperature ~37 °C.

Fig. R1. Interfacial adhesion characterization via peeling tests. (A) Adhesive force-displacement curves and (B) calculated interfacial energy from the 180° peeling test of the bio-adhesive layer interacting with hydrogel. The adhesion force is higher under a moist condition, mimicking the human body environment, favorable for potential implantable biomedical device applications.

Fig. R1 presents that the bio-adhesive layer maintains robust adhesion in both dry and wet conditions. Notably, the adhesion force shows an enhancement in a moist condition that simulates the internal human body environment. This enhancement is likely due to the hydrophilic nature of the hydrogel and adhesive, which forms additional hydrogen bonds in the presence of moisture. Such behavior is advantageous for implantable devices that must maintain reliable adhesion in the moist internal conditions of the human body. We have clarified these points in the revised manuscript and supporting information, providing a clearer view of the bio-adhesive layer's performance.

Modification to the manuscript:

- (1) On page 26, in the revised supporting information, we added **Fig. R1** as **Fig. S15**.
- (2) On page 9, in the revised manuscript, we added, "Importantly, the bio-adhesive layer exhibits robust adhesion under both dry and wet conditions, as displayed in **Fig. S16**. This feature ensures the reliability of implantable devices in the dynamic and moist human body environment."
- (3) On page 28, in the revised manuscript, we added, "Adhesion force was tested by the standard 180° peel test with the Instron machine (Mark-10 ESM303). All tests were conducted with a constant peeling speed of 13 mm/min."

Comment 2: While the thickness of other layers was specified, the actuation hydrogel's thickness was missing (it seems to be considerably thick). Increase of the thickness might enhance actuation performance but could potentially affect sensing and stimulation properties.

2-1) What is the optimal thickness of the actuation layer for achieving reliable shape-morphing?

2-2) How does the thickness of the actuation layer affect sensing and stimulation performance?

Our response: We thank the reviewer's comment regarding the thickness of the actuation hydrogel layer and its potential impact on the device's performance. We have conducted a series of experiments to investigate the effects of the actuation layer's thickness on both actuation performance and its interplay with sensing and stimulation functionalities.

2-1) Optimal thickness for shape-morphing:

Our investigation reveals that thicker hydrogel layers exhibit more pronounced shape morphing at a given activation temperature, but with a slower response rate due to

increased mass, as shown in **Fig. R2A**. Correspondingly, these thicker layers generate greater forces, enhancing actuation performance. However, this comes at the cost of increased overall device volume, which may not be ideal for minimally invasive applications where compactness is crucial. Our approach can tailor the hydrogel layer's thickness based on specific application requirements, highlighting our fabrication method's versatility.

Fig. R2. Effect of hydrogel thickness on actuation performance. (A) Bending angle over time for varying hydrogel thicknesses at 41°C. (B) Measured static force changes at 41°C across different hydrogel thicknesses. (C) Resistive response of a thermal sensor (ERT-JOET102H) across a temperature range of 22 °C to 60 °C for different hydrogel thicknesses.

2-2) Clarification on sensing and stimulation performance:

In our design, the sensors are embedded in a distinct sensing layer and this design strategy ensures that modifications to the thickness of the hydrogel actuation layer are primarily focused on optimizing actuation performance without affecting the sensitivity or functionality of the sensing and stimulation elements. To experimentally validate this design principle, we conducted a series of experiments focusing on the performance of temperature sensors embedded within hydrogel layers of varying thicknesses. These results indicate no significant difference in sensor performance across different hydrogel thicknesses, further supporting our design strategy.

Modification to the manuscript:

- (1) On page 31, in the revised supporting information, we added **Fig. R2** as **Fig. S20**.
- (2) On pages 10-11, in the revised manuscript, we added, “Moreover, our study demonstrates that the thickness of the hydrogel layer determines the actuation performance of the device. As depicted in **Fig. S20A**, thicker hydrogel layers induce more pronounced shape morphing at a set activation temperature, while a slower response due to their increased mass. The thicker layers are also capable of generating higher actuation force, thereby enhancing the actuation capability of the device (**Fig. S20B**). However, the increased device volume may limit its applicability in minimally invasive implantable devices, where compactness is a key factor. In the device configuration, embedded sensors are strategically positioned within a single, thin e-skin layer, thus minimizing the required thickness of the hydrogel layer for efficient actuation and allowing

the integrated system to be relative compact (**Fig. S20C**). Our fabrication technique accommodates customization of the hydrogel layer thickness to optimize device performance for specific applications, demonstrating the method's flexibility and adaptability to various biomedical needs. The utilization of a RGO/PI-nanocomposite temperature sensor enables to precisely identify the real-time information of temperature during muscle motion.”

Comment 3: In Figure 3N, bending configurations appear to be different and uncontrollable although three arms had the same composition. Can the authors explain why? And, it seems those arms are not reversible since the off-arms don't seem to return to the original state.

Our response: We appreciate the reviewer's comment regarding the bending configurations in **Fig. 3N**. Despite the uniform material composition of the three arms, slight variations in environmental conditions during the actuation process such as minor differences in temperature and hydrogel thickness contributed to the non-uniform bending behavior. In addition to the factors already mentioned, the design embedded thermal heaters could also influence the bending behavior of each arm. The embedded heater on e-skin generates temperature control and causes the hydrogel's expansion and contraction. Consequently, the slight differences in the positioning and embedding depth of these heaters can cause variability in bending configurations. We have explained this phenomenon in the revised manuscript.

Regarding reversibility, we would like to provide further clarification. Our experiments (**Fig. S5B**, and **Fig. S7-S9**) indicate that the recovery phase indeed requires a longer duration than the actuation process. This phenomenon is common in many smart materials. As shown in **Fig. 3N**, the observed variation in recovery rates of the arms back to their original shapes is directly related to the sequence and timing of their activation. Importantly, given sufficient time for recovery, our experiments have consistently shown that the arms can return to their original state (**Fig. S7-S9**).

Modification to the manuscript:

On page 11, in the revised manuscript, we added, “Here, the observed variability in bending configuration can be attributed to differences in the positioning and depth of the functional modules within the e-skin layer.”

Comment 4: In Figure 4A, there might be stability issues with the PAAM hydrogel as a pressure sensor. Considering the implant environment, swelling of the hydrogel might hinder pressure measurement due to instability in weight loss and capacitance changes. Any proposed solutions to address this concern would be necessary.

Our response: We appreciate the reviewer's comment on the stability challenges with the use of PAAM hydrogel as a pressure sensor within an implant environment. To address concerns about hydrogel swelling affecting pressure measurement stability, we propose two strategic solutions. First, we suggest the integration of a protective

encapsulation layer around the hydrogel to mitigate the effects of swelling and environmental fluctuations. This encapsulation layer would be designed to be permeable enough to allow for pressure transmission while minimizing direct hydrogel exposure to body fluids that could cause excessive swelling. Secondly, optimizing the hydrogel synthesis recipe (e.g., increasing crosslinker density) to reduce its sensitivity to swelling without compromising its pressure-sensing capabilities. These measures are capable of enhancing the stability and reliability of the hydrogel as a pressure sensor within the implantable environment. We have discussed the possible solutions regarding the future implantable application in the revised manuscript.

Modification to the manuscript:

On page 29, in the revised manuscript, we added, “Here, to mitigate potential stability issues regarding hydrogel swelling and its impact on pressure sensing, we propose two mitigation solutions. Firstly, we incorporate a protective encapsulation layer around the hydrogel. This layer is engineered to be permeable enough to facilitate pressure transmission while simultaneously shielding the hydrogel from direct exposure to body fluids that may induce excessive swelling. Secondly, we refine the hydrogel’s composition by increasing the crosslinker concentration to diminish swelling sensitivity without impairing the hydrogel’s ability to sense pressure. By applying these strategies will enhance the stability of the hydrogel as a pressure sensor for implantable applications.”

Comment 5: The authors discussed the adhesion and actuation force of the hydrogel in Fig. S46, Supplementary Note S8. However, it seems to be considered in a dry environment. Given the nature of hydrogels, there might be swelling and degradation issues when operating in a wet environment.

Our response: We appreciate the reviewer’s comment on the potential swelling of the hydrogel in our device, especially when exposed to a wet environment. The primary objective of our characterization was to understand the baseline mechanical properties and actuation capabilities of hydrogel. Hydrogels, by their nature, are highly responsive to environmental conditions, particularly moisture. However, PNIPAM hydrogel exhibits a thermal-responsive property. Once the external temperature exceeds its LCST, the hydrophobic interaction of the isopropyl group dominates, leading to a conformational transition from coil to globule. This process causes the water inside to be squeezed out, resulting in the volumetric shrinkage of the system (**Fig. R3A**). The temperature-induced molecular structural change significantly impacts the hydrogel’s shape morphing and actuation force (**Fig. R3B&C**), far more than minor humidity variations typical of human body environments. Therefore, this rationale underpins our decision to not specifically investigate the hydrogel’s actuation behavior in wet environments. Our investigations are thus aligned with understanding how temperature variations, rather than moisture changes, influence the actuation capabilities essential for the intended applications of our device.

Fig. R3. Actuation behavior of bilayer soft robots under ambient conditions. (A) Measured volume shrinkage of PNIPAM hydrogel during a heating process with temperature changing from 25 °C to 60 °C. (B) The resultant bending angle θ of a soft robotic arm as a function of the input electric power. (C) Measured static force changes of the soft robotic finger under different temperatures induced by different electrical powers.

In response to the reviewer's note on adhesion and actuation force evaluations potentially occurring in dry conditions, we wish to clarify that our bioadhesive behavior assessments were conducted under simulated physiological conditions. All samples were incubated overnight to ensure a representative *in vivo* environment is achieved, maintaining the integrity of tissue/organ surfaces for accurate assessment (Fig. R4). This approach ensures a thorough evaluation of the hydrogels' performance, including their reaction to swelling and degradation, under conditions that closely emulate their intended operational environment.

Fig. R4. Demonstration of hydrogel adhesion to biological tissues. (A&B) Adhesion behavior of a PNIPAM-co-PAAm hydrogel with an LCST of 36 °C across different temperatures. Below the LCST, the hydrogel exhibits robust adhesion below its LCST, which gradually loses above this

threshold. All tests were conducted after overnight incubation simulating physiological conditions at 34 °C, 37 °C, and 40 °C. Scale bars, 5 mm.

Modification to the manuscript:

(1) On pages 5-6, in the revised manuscript, we modified, “The PNIPAM hydrogel can undergo a dramatic volumetric reduction of about 90% as the temperature shifts from 25 °C to 60 °C. Notably, this significant and rapid deswelling behavior is initiated only when temperature is beyond its LCST (32 °C~34 °C), enabling excellent actuation capabilities within biological environments (**Fig. S3A&B**, and **Fig. 2H**).”

(2) On page 29, in the revised supporting information (**Fig. S19**'s caption), we added, “Here, all characterizations were conducted under ambient conditions because of the unique temperature-responsive nature of the PNIPAM hydrogel. Its actuation force and shape morphing capabilities are predominantly influenced by temperature-induced molecular structural changes, rather than slight humidity changes typical in human body environments. This characteristic underlines the hydrogel's efficacy in translating thermal stimuli into mechanical actions, essential for the intended biomedical applications.”

(3) On page 52, in the revised supporting information, we added **Fig. R4** as **Fig. S40**.

(4) On page 21, in the revised manuscript, we added, “We also explored the bioadhesive behavior of our device on targeted tissues/organs. We observed that hydrogel's inherent adhesiveness is significantly related to its water content and temperature. As shown in **Fig. S40**, there is a decline in adhesive strength as temperatures approach the hydrogel's LCST. While this inherent adhesive capability contributes to the initial secure placement of the device, it's noteworthy that solely relying on this property might not guarantee a durable bond, especially as the hydrogel experiences dehydration. However, this temperature-responsive adhesiveness can play a complementary role in enhancing the device's grasp by counterbalancing any potential decrease in force due to hydrogel reswelling.”

(5) On page 28, in the revised manuscript, we added, “Adhesion force was tested by the standard 180° peel test with the Instron machine (Mark-10 ESM303). All tests were conducted with a constant peeling speed of 13 mm/min.”

Reference

Y. Jiang, A.A. Trotsyuk, S. Niu, D. Henn, K. Chen, C.C. Shih, M.R. Larson, A.M. Mermin-Bunnell, S. Mittal, J.C. Lai, A. Saberi, E. Beard, S. Jing, D. Zhong, S.R. Steele, K. Sun, T. Jain, E. Zhao, C.R. Neimeth, W.G. Viana, J. Tang, D. Sivaraj, J. Padmanabhan, M. Rodrigues, D.P. Perrault, A. Chattopadhyay, Z.N. Maan, M.C. Leeolou, C.A. Bonham, S.H. Kwon, H.C. Kussie, K.S. Fischer, G. Gurusankar, K. Liang, K. Zhang, R. Nag, M.P. Snyder, M. Januszyk, G.C. Gurtner, Z. Bao, Wireless, closed-loop, smart bandage with integrated sensors and stimulators for advanced wound care and accelerated healing, *Nat. Biotechnol.* 41 (2023) 652–662. <https://doi.org/10.1038/s41587-022-01528-3>.

Comment 6: In Figures 5 and S47, the hydrogel was designed to face biological tissue (please correct me if not). In this case, some of physical sensors (e.g., strain and pressure gauges) would be fine, but how to figure out operation of other sensors/stimulators that need direct contacts to tissues? For example, the authors mentioned here electrical stimulation for the bladder, but is it possible to stimulate the bladder through the adhesive layer and PAAM hydrogel? That would be not. Can the authors explain how to achieve such functional operations?

Our response: We thank the reviewer’s comment regarding the configuration of soft robotic gripper for bladder control. Here, regarding the concerns about the operation of stimulators that require direct tissue contact, we would like to clarify the design of our stimulator module.

This stimulator module utilizes gold (Au) electrodes that are specially designed to establish direct contact with bladder tissue, as shown in **Fig. R5**. This direct contact is critical for the effective delivery of therapeutic interventions, such as electrical stimulation for bladder control. Moreover, even in instances where minor gaps might exist at the interface, body fluids naturally act as a conductive bridge, ensuring efficient electrical stimulation.

The hydrogel component of our device is applied primarily to the device’s arms for actuation purposes, which is intentionally designed not to interfere with the stimulator module’s direct tissue contact (**Fig. R5B**). We have clarified this point in revised supporting information, aiming to offer clearer insights into the device’s design and its capability for direct tissue engagement through the simulator module.

Fig. R5. Soft robotic gripper for monitoring bladder volume. (A) Exploded view of the soft robotic finger including actuator, sensor and stimulator components. (B) Detailed schematic illustration for the soft robotic finger, highlighting the direct tissue contact by the stimulation electrode (indicated with a red dashed line). Here the hydrogel actuators are intentionally positioned on the device’s arms, marked by the blue lines, ensuring that they do not hinder the electrodes’ direct engagement with the tissue, facilitating efficient electrical stimulation.

Modification to the manuscript:

(1) On page 44, in the revised supporting information, we added **Fig. R5B** as **Fig. S33B**.

(2) On page 44, in the revised supporting information, we modified the **Fig. S33A&B**'s captions, "(A) Schematic illustration showing exploded layout of the soft robotic gripper that incorporates actuator, sensor and stimulator components. The actuation component includes a passive layer made of an Au/PI bilayer with heat functionality, and an active layer of PNIPAM hydrogel. The sensing component includes an elastic poly(acrylamide) (~PAAm) hydrogel film and a serpentine Au/PI resistor to form a buckled strain sensor. The stimulator component is two pairs of pacing electrodes, employing Au for its exceptional electrical conductivity, biocompatibility, and stability in biological settings. (B) Detailed schematic illustration for the soft robotic finger, highlighting the direct tissue contact by the stimulation electrode (indicated with a red dashed line). Here the hydrogel actuators are intentionally positioned on the device's arms, marked by the blue lines, ensuring that they do not hinder the electrodes' direct engagement with the tissue, facilitating efficient electrical stimulation."

Comment 7: In Figure 5g, it is unclear what the author intended to demonstrate through the programming. Can the authors provide significance of the programming for the control of the bladder. (It is quite strange that E-stim ON when the bladder is void, and E-stim OFF after expansion/contraction cycle... Shouldn't it turn on when the volume reaches maximum?).

Our response: We thank the reviewer's comment regarding the programming strategy depicted in **Fig. 5G**, particularly concerning the application of electrical stimulation (E-stim) in bladder control. Our soft robotic device integrates bladder volume sensing with E-stim to offer a targeted treatment modality for underactive bladder. This strategy aligns with recent advances in bioengineering and medical devices, aiming to offer more personalized treatments for patients with urinary disorders.

Indeed, our programming logic operates as the reviewer has anticipated: The E-stim is activated when the bladder volume surpasses a predefined threshold, set here at 100 mL, targeting the bladder muscles to facilitate urination. Once the bladder empties to below the threshold, indicating successful voiding, the system automatically deactivates the E-stim. Here, a slight delay in turning off was observed which could partially be attributed to the response time for the microcontroller unit (MCU) system to pick up the sensory change and update the output value. This programming cycle ensures that stimulation is provided only, when necessary, closely mimicking the natural urination process and enhancing patient comfort and device efficiency.

It appears that the reviewer's confusion may be due to the unclear labeling in **Fig. 5G**. To ensure better clarity, we have revised the labels as shown in **Fig. R6** and enhanced the explanation in the manuscript.

Fig. R6. A robotic gripper for bladder control. Programmed electrical stimulation (top) and measured volume of an artificial bladder based on a balloon (middle and bottom). The experimental demonstration is conducted using the following parameters: volume threshold of ~100 mL, electrical stimulation amplitude of 3 V. Here, a slight delay in the deactivation process could be partially attributed to the response time of microcontroller unit (MCU) system in detecting changes from the sensors and updating the output accordingly.

Modification to the manuscript:

- (1) On page 20, in the revised manuscript, we added **Fig. R6** as **Fig. 5G**.
- (2) On page 21, in the revised manuscript, we modified the figure caption of **Fig. 5G**, “Programmed electrical stimulation (top) and measured volume of an artificial bladder based on a balloon (middle and bottom). The experimental demonstration is conducted using the following parameters: volume threshold of ~100 mL, electrical stimulation amplitude of 3 V. Here, a slight delay in the deactivation process could be partially attributed to the response time of microcontroller unit (MCU) system in detecting changes from the sensors and updating the output accordingly.”
- (3) On page 18, in the revised manuscript, we added, “When the balloon’s volume reaches a predetermined threshold, set here at 100 mL, the control system initiates electrical stimulation. Following successful voiding to below the threshold, the system automatically deactivates the stimulation. While electrical stimulation has shown promising results in enhancing bladder control in various studies and clinical trials, its efficacy can differ across individuals. The effectiveness of electrical stimulation for bladder voiding and its required voltage levels requires further investigation beyond the scope of our current study. However, our prototype showcases the potential of integrating

sensing and actuation mechanisms to facilitate timely and adaptive interventions for bladder dysfunction.”

References

Grill, W. M. Electrical stimulation for control of bladder function. Proc. 31st Annu. Int. Conf. IEEE Eng. Med. Biol. Soc. Eng. Futur. Biomed. EMBC 2009 2369–2370 (2009) doi:10.1109/IEMBS.2009.5335001.

Coolen, R. L., Groen, J. & Blok, B. F. M. Electrical stimulation in the treatment of bladder dysfunction: Technology update. Med. Devices Evid. Res. 12, 337–345 (2019).

Comment 8: In Figs. 6 and S52A, it appears difficult or impossible for the electronic components at the end of each arm to form a strong coupling to living tissues, particularly to tissues whose volume changes over time, such as the bladder. How did the authors control or attach the robotic arms?

Our response: We thank the reviewer’s comment regarding the integration of electronic components at the end of each robotic arm with living tissues, especially those that undergo volume changes such as bladder.

Our design employs a multilayer fabrication approach, incorporating a layer of PNIPAM hydrogel. The inherent thermos-responsive property of PNIPAM hydrogel enables our device to dynamically adapt to the body’s natural thermal environment. This shape adaptability ensures that our soft robotic devices maintain a stable and effective interface with living tissues, seamlessly accommodating volumetric changes without compromising contract or functionality (**Fig. R7A**).

Fig. R7. shape conformability of soft robotic devices with biological tissues/organs. (A) Measured static force changes of the soft robotic finger under different temperatures. **(B)** Adhesion strength of a PNIPAM-co-PAAm hydrogel with an LCST of 36 °C across different temperatures.

In addition, the inherent adhesiveness of the hydrogel layer significantly enhances the device’s ability to securely couple with tissue surfaces. Upon deformation and subsequent

contact with tissue, this adhesiveness strengthens the coupling between the device and the tissue, ensuring a strong attachment even as the tissue undergoes volumetric changes, thereby preserving the functional integrity of the device's functional modules (Fig. R7B).

Fig. R8. *in vivo* validation of a soft robotic gripper for epicardial sensing and pacing. (A) Image of a soft robotic thera-gripper grasping on the epicardial surface of a living mouse heart. Scale bar, 5 mm. (B) Temperature measurements from the thera-gripper during its deployment onto the mouse heart. (C) The surface ECG trace during electrical stimulation using a pair of Au pacing electrodes. (D) Representative measurements of local cardiac contractions using a soft robotic thera-gripper wrapping onto a living mouse heart.

Furthermore, as demonstrated in **Fig. R8**, our device has successfully detected key physiological parameters, including heart rate, contractile ability, and temperature variations, and effectively provided electrical stimulation. These findings not only showcase the device's operational capabilities but also confirm the effective coupling contact between the device and the living tissues/organs.

Modification to the manuscript:

(1) On page 10, in the revised manuscript, we modified, "**Fig. S19B** shows that the static force exhibits a noticeable increase with rising temperature. At a temperature of 40 °C, the force reaches a maximum of 32 mN. Additionally, it is observed that the generated force remains consistent throughout 40 cycles of alternating power on and off (0.35 W), indicating the robust reversibility of the soft robot (**Fig. S19C**). When compared to similar

hydrogel-based soft actuators, our design consistently achieves a relatively high output force, as shown in Table S1.”

(2) On page 52, in the revised supporting information, we added **Fig. R7B** as **Fig. S40B**.

(3) On pages 21-22, in the revised manuscript, we added, “Additionally, our device offers versatile adaptability for diverse application scenarios through customizable dimensions, sensor positioning, and geometric layouts, ensuring it aligns with the unique morphologies and functional demands of targeted tissues/organs. This flexibility is essential for enabling minimally invasive deployment. As illustrated in **Fig. S38&S39**, **Table S2**, and **Supplementary Note S6**, the design’s adaptability enhances soft robotic technologies for effective integration in a broad spectrum of biomedical applications. We also explored the bioadhesive behavior of our device on targeted tissues/organs. We observed that hydrogel’s inherent adhesiveness is significantly related to its water content and temperature. As shown in **Fig. S40**, there is a decline in adhesive strength as temperatures approach the hydrogel’s LCST. While this inherent adhesive capability contributes to the initial secure placement of the device, it’s noteworthy that solely relying on this property might not guarantee a durable bond, especially as the hydrogel experiences dehydration. However, this temperature-responsive adhesiveness can play a complementary role in enhancing the device’s grasp by counterbalancing any potential decrease in force due to hydrogel reswelling.”

(4) On page 28, in the revised manuscript, we added, “Adhesion force was tested by the standard 180° peel test with the Instron machine (Mark-10 ESM303). All tests were conducted with a constant peeling speed of 13 mm/min.”

Reference

Y. Jiang, A.A. Trotsyuk, S. Niu, D. Henn, K. Chen, C.C. Shih, M.R. Larson, A.M. Mermin-Bunnell, S. Mittal, J.C. Lai, A. Saberi, E. Beard, S. Jing, D. Zhong, S.R. Steele, K. Sun, T. Jain, E. Zhao, C.R. Neimeth, W.G. Viana, J. Tang, D. Sivaraj, J. Padmanabhan, M. Rodrigues, D.P. Perrault, A. Chattopadhyay, Z.N. Maan, M.C. Leeolou, C.A. Bonham, S.H. Kwon, H.C. Kussie, K.S. Fischer, G. Gurusankar, K. Liang, K. Zhang, R. Nag, M.P. Snyder, M. Januszyk, G.C. Gurtner, Z. Bao, Wireless, closed-loop, smart bandage with integrated sensors and stimulators for advanced wound care and accelerated healing, *Nat. Biotechnol.* 41 (2023) 652–662. <https://doi.org/10.1038/s41587-022-01528-3>.

Comment 9: Page 3, line 86 It is not ‘overactive’, but ‘underactive’ bladder.

Our response: We appreciate the reviewer’s comment.

Modification to the manuscript:

On page 3, in the revised manuscript, we have corrected “overactive” to “underactive”.

Comment 10: In fig. 2c, is there a motivation for using such diverse materials for different components? This referee can’t find outstanding characteristics of those materials for sensing components.

Our response: We appreciate the reviewer’s comment regarding the selection of diverse materials for different components as depicted in **Fig. 2C** and the aims behind their use in sensing components of our soft robots.

The motivation for utilizing a variety of materials in our device lies in mimicking the hierarchical structures and functionalities of biological systems, such as skin and muscle. Here, we highlight the integration of multi-electronic modules and thermally actuated hydrogels within a single platform. This integration enables our soft robotic devices to not only perform receptor-like sensing functions for detecting various stimuli but also exhibit muscle-like contractions for generating physically adaptive motion, thereby endowing the soft robotic devices with the capabilities to autonomously navigate complex environments.

The selection of materials for the flexible composite layer is driven by the need to fabricate a multi-modal electronic skin (e-skin) that incorporates distinct sensors (e.g., strain, pressure, pH, and temperature sensors) and stimulators (thermal and electrical). Each chosen material contributes both individual properties and a synergistic effect to enhance the overall functionality of the system. For instance, AgNWs provide excellent conductivity and flexibility, making them ideal for wearable sensors. Graphene stands out for its high surface area, electrical conductivity, and mechanical strength, enhancing sensitivity and selectivity in biosensing applications. Meanwhile, MXene offers high conductivity and hydrophilicity, making it an advantageous choice for biosensor interfaces.¹⁻⁴ Through such strategic material integration, we ensure our soft robotic devices can be well equipped to sensitively and accurately perceive and interact their surroundings.

Moreover, the diversity of materials showcases the versatility of our solution-based method, enabling the integration of a wide range of advanced functional materials. Such versatility surpasses the capabilities of 3D printing or other traditional fabrication methods⁵⁻⁷, allowing us to engineer devices via fine-tuning each component’s properties to specific functional needs, and optimizing the performance and utility of our soft robotic systems.

Modification to the manuscript:

On pages 6-7, in the revised manuscript, we modified, “Furthermore, compared with a make-and-transfer method, our *in situ* solution-based fabrication approach enables the seamless integration of sensors into the e-skin matrix in a single step, enhancing mechanical and electrical performance by reducing interfacial resistances and improving mechanical conformity, thereby significantly improving sensitivity and responsiveness. This approach also offers a versatile platform that can be constructed using a broad range of functional nanomaterials hybridized with a polymeric matrix to form a multi-modal sensing system. By selecting materials that offer unique functional attributes, from AgNWs known for their conductivity and flexibility to graphene and MXene for their high surface area, electrical conductivity, and hydrophilicity, this system can emulate the skin with complex somatosensory system, where various mechanoreceptors and thermoreceptors distributed in the epidermal and dermal layers enable the spatiotemporal recognition of the magnitude and location of touch and temperature stimuli.”

References

- Basarir, F., Madani, Z. & Vapaavuori, J. Recent Advances in Silver Nanowire Based Flexible Capacitive Pressure Sensors: From Structure, Fabrication to Emerging Applications. *Adv. Mater. Interfaces* **9**, (2022).
- Chauhan, N., Maekawa, T. & Kumar, D. N. S. Graphene based biosensors - Accelerating medical diagnostics to new-dimensions. *J. Mater. Res.* **32**, 2860–2882 (2017).
- Amara, U., Hussain, I., Ahmad, M., Mahmood, K. & Zhang, K. 2D MXene-Based Biosensing: A Review. *Small* **19**, 1–38 (2023).
- Driscoll, N. *et al.* MXene-infused bioelectronic interfaces for multiscale electrophysiology and stimulation. *Sci. Transl. Med.* **13**, eabf8629 (2021).
- Liu, H. *et al.* 3D Printed Flexible Strain Sensors: From Printing to Devices and Signals. *Adv. Mater.* **33**, 2004782 (2021).
- Zhu, Z., Park, H. S. & McAlpine, M. C. 3D printed deformable sensors. *Sci. Adv.* **6**, eaba5575 (2023).
- Reeder, J. *et al.* Mechanically Adaptive Organic Transistors for Implantable Electronics. *Adv. Mater.* **26**, 4967–4973 (2014).

Comment 11: In figs. 2f and 2g, the authors claimed that soft robots can form conformal contacts to tissues, but did the authors study or compare mechanical modulus between devices and tissues?

Our response: We thank the reviewer's comment concerning the mechanical compatibility of our soft robotic devices with biological tissues, as shown in **Fig. R9A (Fig. 2F)** and **Fig. R9B (Fig. 2G)**. We employ a multilayer design mimicking the relation between human skin and skeletal muscle to integrate the e-skin layer with an artificial muscle layer composed of PNIPAM-based hydrogel. This hydrogel layer, designed for direct tissue interface, is critical in our device's architecture.

In choosing materials for our device, especially those designed for implantable biomedical applications, our priorities are biocompatibility and the mechanical softness, essential for effective tissue integration. Hydrogels, exhibiting the above-mentioned properties, are ideally suitable for our application in implantable devices. Compared to other smart materials such as shape memory polymers and alloys, which exhibit the elastic moduli (E) in the order of several megapascals (MPa)⁸, hydrogels have an elastic modulus ranging from 1-100 kPa. This range is notably similar to that of soft tissues, which typically exhibit E from a few kPa to hundreds of kPa⁹. Specially, the PNIPAM hydrogel in our research exhibited an $E \sim 1.2$ kPa, as shown in **Fig. R9C**. This mechanical similarity facilitates a more seamless integration with surrounding tissues and minimizes the potential damage from mechanical mismatch¹⁰.

Fig. R9. Mechanical properties of PNIPAM hydrogel. (A&B) Optical image showing conformal attachment of the soft sensory robot onto human skin (A) and porcine tissue (B) with high mechanical compliance. (C) Tensile stress curve of PNIPAM hydrogel (Dimensions: thickness 4mm, width 10mm, initial length 15mm), highlighting an elastic modulus (E) of ~ 1.2 kPa. This value is within the range of soft tissues, which typically exhibit an E from a few kPa to hundreds of kPa, ensuring the hydrogel's compatibility with biological tissues and reducing risks associated with mechanical mismatches.

Modification to the manuscript:

(1) On page 13, in the revised supporting information, we added **Fig. R9C** as **Fig. S3C**. (C) Tensile stress curve of PNIPAM hydrogel (Dimensions: thickness 4 mm, width 10 mm, initial length 15 mm), highlighting an elastic modulus (E) of approximately 1.2 kPa. This value is within the range of soft tissues, which typically exhibit an E from a few kPa to hundreds of kPa, ensuring the hydrogel's compatibility with biological tissues and reducing risks associated with mechanical mismatches.

(2) On page 6, in the revised manuscript, we modified, “The as-fabricated soft robot can form a highly conformal interface with diverse biological surfaces, indicating its inherent mechanical softness and high biocompatibility. This adaptability minimizes potential risks related to mechanical incompatibility, facilitating its smooth integration with targeted tissues/organs (**Fig. 2F&G**, and **Fig. S3C**).”

(3) On page 28, in the revised manuscript, we added, “Mechanical tests were conducted on rectangular-shape specimens with the dimensions of 10 mm in width, 2 mm in thickness, and 16 mm in length) using the Instron machine (Mark-10 ESM303).”

References

Czerner, M., Fellay, L. S., Suárez, M. P., Frontini, P. M. & Fasce, L. A. Determination of Elastic Modulus of Gelatin Gels by Indentation Experiments. *Procedia Mater. Sci.* **8**, 287–296 (2015).

Xie, C., Wang, X., He, H., Ding, Y. & Lu, X. Mussel-Inspired Hydrogels for Self-Adhesive Bioelectronics. *Adv. Funct. Mater.* **30**, 1909954 (2020).

Comment 12: In fig. S12B, such layered structures are likely to interfere with each other's signals. Is there any reason to measure both parameters at the same time? Or, even if simultaneous measurements are made, does each sensor operate independently?

Our response: We thank the reviewer's comment regarding the potential for signal interference in the layered structures as shown in **Fig. S12B** (**Fig. S11B** in the updated supporting information).

As shown in **Fig. S11B**, we employed a dual-transfer printing method to fabricate the functional modules located in different layers. This strategy allows for the integration of a minimal amount of polyimide with functional materials to form composite functional modules. This not only preserves the functional integrity of the materials within each module but also leverages the remaining polyimide to serve as a dual purpose. Beyond its inclusion in the composite, polyimide acts as an insulating barrier between adjacent modules. This barrier is crucial for our device's functionality, as it prevents electrical and chemical signal interference between modules, thereby ensuring that each functional module operates independently and as intended.

The primary purpose of measuring multiple parameters simultaneously is to showcase the multifunctional capabilities of our solution-based approach, which closely mimicking the complex functionality of natural model (e.g., human skin), enabling the device to perform concurrent sensing operations. By demonstrating the device's ability to interact comprehensively with its environment, we not only highlight its potential for various applications but also validate our fabrication method's versatility in fabricating multifunctional e-skin. This strategy surpasses conventional fabrication methods like 3D printing, offering new possibilities for advancements in robotics, wearable technologies, and biomedical devices.

Moreover, each functional module operates independently even during simultaneous measurements. As aforementioned, the integration of a polyimide layer between modules, serves as an effective barrier against any potential signal interference, ensuring that despite the simultaneous operation of multiple sensors, each module retains its individual functionality and delivers accurate, independent measurements.

Modification to the manuscript:

On page 7, in the revised manuscript, we modified, "**Fig. 2M** displays a more complicated integration paradigm with multi-layer stacking, where different electronic components (e.g., PEDOT:PSS/PI-based conductive electrodes and RGO/PI temperature sensors) can be distributed in different layers of the e-skin to achieve simultaneous functional versatility and compactness. This assembly technique ensures the e-skin remarkable thinness and flexibility, enhancing its effective performance and seamless integration in implantable applications (**Fig. S11B**). The X-ray photoelectron spectroscopy (XPS) characterization on the e-skin layers reveals the precise nanoscale integration of active

materials within a polymer matrix, as detailed in **Fig. S12-S14** and **Supplementary Note S1**. It showcases the optimal distribution and intermolecular bonding of the composite components, effectively addressing the common challenge of uneven dispersion of nanomaterials, which usually undermines the performance of conventional composites. Our approach minimizes the polymer amount required to integrate nanomaterials into composite functional modules and utilizes excess polymer as an insulating layer to separate modules, preventing interference between their electrical and chemical signals, thereby ensuring that each functional module operates independently and effectively. This approach harmoniously combines the distinct properties of each constituent, achieving a balance between structural integrity and functional versatility. This advanced level of integration would be of great value for soft robots that seek to achieve multifunctionality and local sensing capabilities approaching skin.”

Comment 13: there's no description for Figs. 3H&3I.

Our response: We thank the reviewer's comment regarding Fig. 3H&3I. We have added detailed descriptions for both figures.

Modification to the manuscript:

On page 11, in the revised manuscript, we added, “**Fig. 3H** illustrates the resistive change in a relatively linear relation with temperature for the RGO/PI thermal sensor. The temperature coefficient of the resistance (TCR) of the RGO/PI thermal sensor is $> 0.5\%/^{\circ}\text{C}$, featuring its high thermal sensitivity. On the other hand, the RGO/PI-based thermal sensor exhibits a stable performance after 1000 bending cycles, and even after immersing in PBS solution. **Fig. 3I** and **Fig. S21F** show performance of the thermal sensor in response to cycles of temperature rise and drop, indicating good sensing stability.”

Comment 14: We usually do not include information about SI figures in main figure captions. For example, in fig.3, there is information about fig. S18 and 25, but these should be removed.

Our response: We thank the reviewer's comment. We have removed these SI figures from the main figure captions based on reviewer's advice.

Modification to the manuscript:

We have updated figure captions in the revised manuscript.

Comment 15: Please do double check all the figures including SI, whose order is a total mess.

Our response: We appreciate the reviewer's comment to help us significantly enhance the clarity and coherence of our manuscript. We have conducted a thorough review and reorganization of all figures and supplementary information to ensure a logical flow and clear presentation throughout our submissions.

Modification to the manuscript: We have reorganized and updated all supplementary figures for clarity.

Responses to comments of Referee #2

Comments from Referee #2:

Summary Comment: Lin Zhang et al. developed various types of soft robots, utilizing an electronic skin (e-skin) made of materials like silver nanowires, reduced graphene oxide, MXene, and conductive polymers integrated into a polymer matrix. This e-skin is designed to mimic skin receptors, enabling the perception of various stimuli. The robots, inspired by natural forms like starfish, chiral seedpod, and others, are equipped with features for bending, twisting, and expanding, and include an artificial muscle for dynamic movement. This study also integrated wireless modules to control and communicate without tethering. Demonstrated in various medical scenarios, these devices showed capabilities such as blood pressure sensing and adapting to organ surfaces for diagnostic and therapeutic functions, exemplifying their potential as multifunctional, minimally invasive medical tools. However, the devices used materials commonly used in practical applications, with the exception of an *in situ* solution-based fabrication approach, showing no significant material novelty. This article also lacks either long-term feasibility or detailed information about the mechanism of each sensor like the relationships between pH, pressure and resistance. Despite such some disadvantages, the positive aspect lies in its multifunctional ability to be implanted in real scenarios, possibly leading to successful demonstrations for various organ shapes. In specific, massive integration of multifunctional soft robots entailing high-performance functional nanomaterials and actuating modules is highly impressive. Additionally, this article demonstrates a high level of systemic completeness, considering real clinical situations. Therefore, the reviewer believes that with some modifications addressing specific comments included in the following, this paper has the potential to be accepted in Nature Communications.

Our response: We thank the reviewer for these positive comments and for these helpful suggestions for revision. We carefully addressed the issues, as listed below, and revised our manuscript accordingly.

Modification to the manuscript: We have revised both the manuscript and supporting information based on the reviewer's concerns and suggestions.

Comment 1: In the introduction part, some references (Nature Materials volume 15, pages937–950 (2016), Nature Nanotechnology volume 9, pages397–404 (2014), and Nature Electronics volume 6, pages779–789 (2023)) regarding stable tissue-interfacing performance of wearable/implantable devices should be cited to justify the authors' research purpose.

Our response: We thank the reviewer's suggestion. We have carefully integrated the recommended references into the introduction section.

Modification to the manuscript:

On page 2, in the revised manuscript, we have cited all suggested references.

References

D. Son, J. Lee, S. Qiao, R. Ghaffari, J. Kim, J.E. Lee, C. Song, S.J. Kim, D.J. Lee, S.W. Jun, S. Yang, M. Park, J. Shin, K. Do, M. Lee, K. Kang, C.S. Hwang, N. Lu, T. Hyeon, D.H. Kim, Multifunctional wearable devices for diagnosis and therapy of movement disorders, *Nat. Nanotechnol.* 9 (2014) 397–404. <https://doi.org/10.1038/nnano.2014.38>.

A. Chortos, J. Liu, Z. Bao, Pursuing prosthetic electronic skin, *Nat. Mater.* 15 (2016) 937–950. <https://doi.org/10.1038/nmat4671>.

H. Choi, Y. Kim, S. Kim, H. Jung, S. Lee, K. Kim, H.S. Han, J.Y. Kim, M. Shin, D. Son, Adhesive bioelectronics for sutureless epicardial interfacing, *Nat. Electron.* 6 (2023) 779–789. <https://doi.org/10.1038/s41928-023-01023-w>.

Comment 2: The reviewer believes that sensory-motor integration is valuable when the robot can control motors in response to signals received from sensors. However, it seems that controlling the motors and performing sensor functions operate independently. In that case, can this robot still be considered as the presence of responsiveness as mentioned in the introduction?

Our response: We thank the reviewer’s question regarding sensory-motor integration within our soft robotic system. Our design can be engineered to offer dual-mode functionality, adapting to a broad spectrum of operational requirements, enhancing the system’s versatility and application range.

We have demonstrated that the integrated design that leverages on the synergistic interplay between sensors and soft robotic actuators enables volatile adaptivity and responsiveness of the robotic implant in ever-changing conditions. This design allows sensory inputs to directly influence actuator outputs in real-time, creating a feedback loop that allows for automatic adjustment based on environmental stimuli. This feature underscores our system’s responsiveness and adaptability, allowing with the principles of sensor-actuator integration highlighted by the reviewer.

As an example, we present an adaptively controlled responsive robotic gripper that maintains optimal actuation temperatures responsive to external temperature changes. The responsive self-adaptation in different thermal environments ensures optimal and predictable heating patterns regardless of conditions, as well as increasing patient safety and preventing potential burn hazard caused by overheating. **Fig. R10A** shows an integrative system that provides temperature sensory readout and, through feedback-controlled power delivery, allows actuation responsive to external thermal environment. **Fig. R10B** demonstrates the working principles of controlling hardware and software. The resistive temperature sensor was connected in series with a reference resistor. The voltage drops over the sensor is read and converted by the on-chip ADC and is processed by the microcontroller to convert to a temperature value. Upon receipt of the temperature value, the control algorithm compares it with the target temperature to produce an error value. The error value is multiplied with a predefined coefficient to produce a corrective

factor and is added to the current output value to generate a new output value. The on-chip pulse-width modulation (PWM) module converts the output value to a PWM voltage signal, which is then amplified by the on-board power MOSFET, forming a controller current source to produce a current through the heater. When the external temperature changes, a large corrective factor is produced to allow rapid adaptations. When the target temperature was reached, a constant current is produced to maintain the optimal temperature. **Fig. R11** showed the time-synchronized current and temperature plot during different responsive phases after a sudden change in temperature. During the underheating phase, a rapid increase in output current was generated by the controlled current source in response to the sudden change in temperature. When the target temperature was first reached, the current output remains constant until the actuator was overheated, during which the current output drops again to reach the static phase at which the current levels off.

Fig. R10. Soft robots designed integrated sensing and actuation capabilities. (A) Schematic diagram of the responsive robotic gripper with control circuitry. **(B)** Functional block diagram of the algorithmic and hardware design of the adaptive control loop.

Moreover, our robotic system also supports operation in an open-loop configuration, where actuation is not directly governed by immediate sensory inputs but can be pre-programmed or controlled externally. This flexibility is particularly beneficial for scenarios where actions need to be acquired based on predefined conditions rather than dynamic sensory input, offering stability in a controlled environment. An example of this functionality is our bladder control system, designed to maintain a consistent grip post-

implantation. However, we have also integrated a closed-loop mechanism between the stimulator and sensor modules for this application. Electrical stimulation is activated when the bladder reaches its maximum volume threshold, targeting the bladder muscles to facilitate urination. Upon complete voiding of the bladder, the integrated sensors detect the reduction in bladder volume, which subsequently triggers the deactivation of the electrical stimulation. This programming cycle ensures that stimulation is provided only, when necessary, closely mimicking the natural urination process and enhancing patient comfort and device efficiency (**Fig. 5A-5G, Fig. S34-S35**).

Fig. R11. Soft robots designed integrated sensing and actuation capabilities. (A) Infrared and optical images of the responsive robotic grippers at different stages of controlled actuation. (B&C) Time synchronized current (B) and temperature plot (C) capturing the controlled actuation changes after a sudden change in ambient temperature.

Our robotic system is engineered to support both responsive closed-loop feedback and stable open-loop operation for sensor-actuator integration. This not only meets the initial responsiveness criteria but also significantly expands its utility across diverse fields, ranging from precise medical interventions to versatile robotic applications.

Modification to the manuscript:

- (1) On page 36, in the revised supporting information, we added **Fig. R10** as **Fig. S25**.
- (2) On page 37, in the revised supporting information, we added **Fig. R11** as **Fig. S26**.
- (3) On page 5, in the revised supporting information, we added,“

Supplementary Note S3: The sensory-motor integration within the soft robotic system

The integrated design that leverages on the synergistic interplay between sensors and soft robotic actuators enables volatile adaptivity and responsiveness of the robotic implant in ever-changing conditions. This design allows sensory inputs to directly influence actuator outputs in real-time, creating a feedback loop that allows for automatic adjustment based on environmental stimuli. This feature underscores our system's responsiveness and adaptability, allowing with the principles of sensor-actuator integration.

As an example, we present an adaptively controlled responsive robotic gripper that maintains optimal actuation temperatures responsive to external temperature changes. The responsive self-adaptation in different thermal environments ensures optimal and predictable heating patterns regardless of conditions, as well as increasing patient safety and preventing potential burn hazard caused by overheating. **Fig. S25A** shows an integrative system that provides temperature sensory readout and, through feedback-controlled power delivery, allows actuation responsive to external thermal environment. **Fig. S25B** demonstrates the working principles of controlling hardware and software. The resistive temperature sensor was connected in series with a reference resistor. The voltage drops over the sensor is read and converted by the on-chip ADC and is processed by the microcontroller to convert to a temperature value. Upon receipt of the temperature value, the control algorithm compares it with the target temperature to produce an error value. The error value is multiplied with a predefined coefficient to produce a corrective factor and is added to the current output value to generate a new output value. The on-chip pulse-width modulation (PWM) module converts the output value to a PWM voltage signal, which is then amplified by the on-board power MOSFET, forming a controller current source to produce a current through the heater. When the external temperature changes, a large corrective factor is produced to allow rapid adaptations. When the target temperature was reached, a constant current is produced to maintain the optimal temperature. **Fig. S26** showed the time-synchronized current and temperature plot during different responsive phases after a sudden change in temperature. During the underheating phase, a rapid increase in output current was generated by the controlled current source in response to the sudden change in temperature. When the target temperature was first reached, the current output remains constant until the actuator was overheated, during which the current output drops again to reach the static phase at which the current levels off."

(4) On page 12, in the revised manuscript, we added, "Furthermore, our soft robotic system exemplifies advanced sensory-motor integration, leveraging the synergistic relationship between embedded sensors and actuators to achieve dynamic adaptivity and responsiveness to environmental changes. A prime example is a temperature-sensitive control system, as shown in **Fig. S25A**, which utilizes real-time sensory feedback to dynamically adjust heating in response to environmental temperature changes. The operational principle, as detailed in **Fig. S25B** and **Supplementary Note S3**, involves a

microcontroller-driven algorithm that interprets temperature input collected by a resistive temperature sensor, and modulates the electric heater's current accordingly, enabling rapid adaptations to achieve and maintain a preset temperature. **Fig. S26** presents a soft robotic finger's real-time response to temperature variations, ensuring stable shape adaptation through this regulatory mechanism. Moreover, this intelligent control significantly improves safety by preventing the risk of overheating, thereby ensuring the system's safe operation in various thermal conditions, highlighting our device's ability to provide precise thermal management, enhancing both efficacy and safety in its applications."

(5) On page 29, in the revised manuscript, we added, "

The sensory-motor integration within the soft robotic system

The robotic gripper and the external circuitry were connected in series with an NI DMM amperemeter set for DC current measurement. The device was cooled to 22 °C in ambient temperature before the system was started up to capture its response to a sudden decrease in ambient temperature. The temperature readouts recorded by the device's integrated sensor were logged via a microcontroller unit (MCU) and cross-referenced with data from FLIR thermal camera. Both the current and temperature data were analyzed using custom Python script designed specifically for this purpose."

References

G.C. van Rhoon, T. Samaras, P.S. Yarmolenko, M.W. Dewhurst, E. Neufeld, N. Kuster, CEM43°C thermal dose thresholds: a potential guide for magnetic resonance radiofrequency exposure levels?, *Eur. Radiol.* 23 (2013) 2215–2227. <https://doi.org/10.1007/s00330-013-2825-y>.

M.W. Dewhurst, B.L. Viglianti, M. Lora-Michiels, P.J. Hoopes, M. Hanson, THERMAL DOSE REQUIREMENT FOR TISSUE EFFECT: EXPERIMENTAL AND CLINICAL FINDINGS., *Proc. SPIE--the Int. Soc. Opt. Eng.* 4954 (2003) 37. <https://doi.org/10.1117/12.476637>.

Comment 3: It seems good that the robot is designed differently for each target organ. The reviewer is curious about the variation in curvature among different organs. For instance, in the case of organs like the heart and bladder, squeezing too much might make it challenging to grasp effectively. How was this consideration taken into account in the design?

Our response: We thank the reviewer's question regarding how our soft robotic systems accommodate the anatomical diversity and curvature of different target organs. Our design process incorporated multiple factors to ensure that our device can achieve shape adaptability, effective data acquisition, and potential therapeutic interventions across various applications.

Fig. R12. Illustration of soft robotic implants exhibiting significant modularity and adaptability. (A-D) A starfish-inspired device featuring twelve arms. (E-H) A hand-structured soft robotic device that can be compactly housed in and released from a catheter. (I-L) A twisted soft robotic device designed for storage and deployment through a 14-Gauge Tuohy needle. Scale bars, 1 cm.

As mentioned by the reviewer, our device is highly customizable. We design specific configurations of the device to align the morphology of different organs, considering their shape (e.g., tubular, spherical), size, and the intended functionalities. The customization extends to the device's dimensions, the selection and distribution of sensors, as well as their geometrical arrangement, as illustrated in **Fig. R12**, ensuring that our device can conform naturally to different organs and tissues.

Beyond their thermo-responsive property, we have selected PNIPAM hydrogel as a key material due to their mechanical properties, which closely resemble those of body tissues/organs. Unlike other smart materials such as shape memory polymers and alloys, which possess elastic moduli (E) on the order of several megapascals (MPa)⁸, hydrogels feature an elastic modulus within the 1-100 kPa range. This range closely aligns with the E of soft tissues, typically spanning from a few kPa to hundreds of kPa⁹. The PNIPAM hydrogel in our research exhibited an $E \sim 1.2$ kPa, as shown in **Fig. R13A**. Such mechanical similarity facilitates a more seamless integration with surrounding tissues, substantially minimizing the potential damage from mechanical mismatch¹⁰.

Fig. R13. Mechanical behavior of soft robotic device. (A) Tensile stress curve of PNIPAM hydrogel (Dimensions: thickness 4mm, width 10 mm, initial length 15 mm). (B&C) Optical image showing conformal attachment of the soft sensory robot onto human skin (B) and porcine tissue (C) with high mechanical compliance. (D) Measured static force changes of the soft robotic finger under different temperatures. Scale bars, 5 mm.

This compatibility ensures that our devices can naturally conform to the contours of organs and tissues, as demonstrated in **Fig. R13B&C**. Furthermore, the intrinsic flexibility and low stiffness of hydrogels prevent an excessive force on the organ or tissue (**Fig. R13D**). Additionally, we found that the hydrogel's intrinsic adhesiveness, further enhancing the device's integration with biological tissues, ensuring that the device can maintain effective grip and functionality without exerting damaging pressure on the organ, addressing concerns related to squeezing or over-compression (**Fig. R14**).

Fig. R14. Demonstration of hydrogel adhesion to biological tissues. (A&B) Adhesion behavior of a PNIPAM-co-PAAm hydrogel with an LCST of 36 °C across different temperatures. Below the LCST, the hydrogel exhibits robust adhesion below its LCST, which gradually loses above this threshold. All tests were conducted after overnight incubation simulating physiological conditions at 34 °C, 37 °C, and 40 °C. Scale bars, 5 mm.

In summary, our design strategy emphasizes customization, careful material selection, and empirical validation to address the complexities of interacting with various organs. In this way, we ensure that our soft robotic system is not only versatile and adaptable but also safe and effective for a wide range of biomedical applications.

Modification to the manuscript:

(1) On page 21, in the revised manuscript, we added, “Additionally, our device offers versatile adaptability for diverse application scenarios through customizable dimensions, sensor positioning, and geometric layouts, ensuring it aligns with the unique morphologies and functional demands of targeted tissues/organs. This flexibility is essential for enabling minimally invasive deployment. As illustrated in **Fig. S38&S39**, **Table S2**, and **Supplementary Note S6**, the design’s adaptability enhances soft robotic technologies for effective integration in a broad spectrum of biomedical applications.”

(2) On page 13, in the revised supporting information, we added **Fig. R13A** as **Fig. S3C**.

(3) On page 6, in the revised manuscript, we modified, “The as-fabricated soft robot can form a highly conformal interface with diverse biological surfaces, indicating its inherent mechanical softness and high biocompatibility. This adaptability minimizes potential risks related to mechanical incompatibility, facilitating its smooth integration with targeted tissues/organs (**Fig. 2F&G**, and **Fig. S3C**).”

(4) On page 10, in the revised manuscript, we modified, “We further evaluated the mechanical force generated by the soft robotic finger which incorporates a PNIPAM hydrogel layer roughly 1 mm thick, under various input powers. **Fig. S19B** shows that the static force exhibits a noticeable increase with rising temperature. At a temperature of 40 °C, the force reaches a maximum of 32 mN. Additionally, it is observed that the generated force remains consistent throughout 40 cycles of alternating power on and off (0.35 W), indicating the robust reversibility of the soft robot (**Fig. S19C**). When compared to similar hydrogel-based soft actuators, our design consistently achieves a relatively high output force, as shown in **Table S1**.”

(5) On page 52, in the revised supporting information, we added **Fig. R14** as **Fig. S40**.

(6) On pages 21-22, in the revised manuscript, we added, “We also explored the bioadhesive behavior of our device on targeted tissues/organs. We observed that hydrogel’s inherent adhesiveness is significantly related to its water content and temperature. As shown in **Fig. S40**, there is a decline in adhesive strength as temperatures approach the hydrogel’s LCST. While this inherent adhesive capability contributes to the initial secure placement of the device, it’s noteworthy that solely relying on this property might not guarantee a durable bond, especially as the hydrogel experiences dehydration. However, this temperature responsive adhesiveness can play a complementary role in enhancing the device’s grasp by counterbalancing any potential decrease in force due to hydrogel reswelling.”

(7) On page 29, in the revised manuscript, we added, “Adhesion force was tested by the standard 180° peel test with the Instron machine (Mark-10 ESM303). All tests were conducted with a constant peeling speed of 13 mm/min.

Mechanical tests were conducted on rectangular-shape specimens with the dimensions of 10 mm in width, 2 mm in thickness, and 16 mm in length) using the Instron machine (Mark-10 ESM303).”

Reference

Y. Jiang, A.A. Trotsyuk, S. Niu, D. Henn, K. Chen, C.C. Shih, M.R. Larson, A.M. Mermin-Bunnell, S. Mittal, J.C. Lai, A. Saberi, E. Beard, S. Jing, D. Zhong, S.R. Steele, K. Sun, T. Jain, E. Zhao, C.R. Neimeth, W.G. Viana, J. Tang, D. Sivaraj, J. Padmanabhan, M. Rodrigues, D.P. Perrault, A. Chattopadhyay, Z.N. Maan, M.C. Leeolou, C.A. Bonham, S.H. Kwon, H.C. Kussie, K.S. Fischer, G. Gurusankar, K. Liang, K. Zhang, R. Nag, M.P. Snyder, M. Januszyk, G.C. Gurtner, Z. Bao, Wireless, closed-loop, smart bandage with integrated sensors and stimulators for advanced wound care and accelerated healing, *Nat. Biotechnol.* 41 (2023) 652–662. <https://doi.org/10.1038/s41587-022-01528-3>.

Comment 4: Why does the parallel strips configuration result in twisting? Is there any reference for this, or was it an original concept?

Our response: We thank the reviewer’s question regarding the underlying mechanism by which the “parallel strips” configuration results in a twisting motion in our device. This phenomenon arises from the differential expansion and contraction of materials when subjected to stimuli, such as temperature changes.

Initially, PNIPAM hydrogel is a thermo-responsive material whose volume and shape significantly change with temperature variations. Above its LCST, PNIPAM undergoes a shrinking process, leading to a reduction in the hydrogel’s volume. Conversely, polyimide (PI) is a high-performance polymer known for its thermal stability, exhibiting minimal expansion or contraction upon heating, especially compared to the significant changes observed in PNIPAM hydrogel.

When these inclined PI strips are bonded to the PNIPAM hydrogel, a hybrid bilayer structure is constructed (**Fig. S8A**). Upon heating, the differential thermal shrinkage of PNIPAM and the relative stability of the PI layer induce uneven expansion or contraction across the structure. Especially, as the PNIPAM hydrogel shrinks with heat, the connected PI strips generate a tilting torque, causing the entire structure to twist. This design leverages the disparate thermal response characteristics of the materials combined with a strategic structural arrangement to achieve unique mechanical behavior. The experimental results correspond very well with simulation results (**Fig. R15**).

While the application of this principle has been explored in various studies^{11,12}, our implementation within a soft robotic system, especially for implantable devices, represents a sophisticated adaption aimed at achieving specific functional objectives. We have achieved an integration of structural design with sensors, representing a more advanced extension of this concept.

Fig. R15. FEA simulation results of a bilayer soft robotic device mimicking a chiral seedpod. (A-F) Stress distribution across the device during the shape-shifting process, facilitated by strategic stripe patterns on the 3D bilayer robot. This visualization highlights the material property mismatches including mechanical and thermal responsiveness that drive the device's complex morphological transformations.

We have incorporated a more detailed discussion on the design mechanism behind the parallel strips' configuration, ensuring a comprehensive understanding of the concept and its application within our work.

Modification to the manuscript:

- (1) On page 18, in the revised supporting information, we added **Fig. R15** as **Fig. S8C**.
- (2) On page 18, in the revised supporting information (**Fig. S8B**'s caption), we added "The PNIPAM hydrogel contracts while PI maintains relative stability upon heating. The contraction of the PNIPAM hydrogel activates a tilting torque through the bonded PI strips, resulting in asymmetric deformation throughout the device."
- (3) On page 6, in the revised manuscript, we added, "Upon thermal stimulation, the PNIPAM hydrogel contracts while the PI maintains stability, generating a differential contraction across the structure. The differential thermal response induces a tilting torque, leading to local saddle-like curvature and twisting motion of the integrated robotic systems (**Fig. 2J**, **Fig. S8B** and **Supplementary Movie S3**)."

References

- Shian, S., Bertoldi, K. & Clarke, D. R. Dielectric Elastomer Based 'grippers' for Soft Robotics. *Adv. Mater.* **27**, 6814–6819 (2015).
- Shojaeifard, M., Niroumandi, S. & Baghani, M. Programming shape-shifting of flat bilayers composed of tough hydrogels under transient swelling. *Acta Mech.* **233**, 213–232 (2022).

Comment 5: On page 6, if the transfer method is well-designed, it could potentially enable multi-modal implementation. Is there a particular advantage in this solution-based approach over transfer methods?

Our response: We thank the reviewer’s question regarding the specific advantages of our *in-situ* solution-based fabrication approach over traditional make-transfer methods, especially concerning the integration of sensors into the e-skin layer for achieving multi-modal functionality. Our method is designed to facilitate seamless and monolithic structuring of functional materials with a passive polymer-based skin, thereby transforming the e-skin into a highly versatile platform. Unlike make-and-transfer methods that may introduce interface issues or compromise material compatibility, our solution-based process facilitates the uniform integration of a wide array of functional nanomaterials within a polymeric matrix. This capability is crucial for constructing e-skin that emulates the complexity of the human skin’s somatosensory system.

(1) Our solution-based method allows for the direct integration of sensors and the e-skin matrix in a single step, ensuring a seamless interface between different components. This monolithic integration is essential for reproducing the integrated somatosensory functions of natural skin, thereby significantly enhancing the e-skin’s sensitivity and responsiveness.

(2) The flexibility to incorporate a broad range of functional nanomaterials into the polymeric matrix without the constraints of transfer compatibility of thermal/mechanical sensitivity, broadens the scope of achievable functionalities. This versatility enables the e-skin to more closely mimic the multi-modal sensory capabilities of natural skin.

(3) The direct integration of sensors within the e-skin matrix, as facilitated by our solution-based approach, results in improved mechanical and electrical performance. This improvement is due to the elimination of interfacial resistances and better mechanical conformity, which are critical for sensitive and accurate sensory detection.

Our approach offers an effective and versatile method for developing e-skin systems capable of sophisticated multi-modal sensory functions. We have further clarified these advantages in the revised manuscript and supplementary information to provide a clearer understanding of our methods and their benefits over traditional make-and-transfer methods.

Modification to the manuscript:

On pages 6-7, in the revised manuscript, we modified, “Furthermore, compared with a make-and-transfer method, our *in situ* solution-based fabrication approach enables the seamless integration of sensors into the e-skin matrix in a single step, enhancing mechanical and electrical performance by reducing interfacial resistances and improving mechanical conformity, thereby significantly improving sensitivity and responsiveness. This approach also offers a versatile platform that can be constructed using a broad range of functional nanomaterials hybridized with a polymeric matrix to form a multi-modal sensing system. By selecting materials that offer unique functional attributes, from AgNWs known for their conductivity and flexibility to graphene and MXene for their high surface area, electrical conductivity, and hydrophilicity, this system can emulate the skin with complex somatosensory system, where various mechanoreceptors and thermoreceptors distributed in the epidermal and dermal layers enable the spatiotemporal recognition of the magnitude and location of touch and temperature stimuli.”

References

- Basarir, F., Madani, Z. & Vapaavuori, J. Recent Advances in Silver Nanowire Based Flexible Capacitive Pressure Sensors: From Structure, Fabrication to Emerging Applications. *Adv. Mater. Interfaces* **9**, (2022).
- Chauhan, N., Maekawa, T. & Kumar, D. N. S. Graphene based biosensors - Accelerating medical diagnostics to new-dimensions. *J. Mater. Res.* **32**, 2860–2882 (2017).
- Amara, U., Hussain, I., Ahmad, M., Mahmood, K. & Zhang, K. 2D MXene-Based Biosensing: A Review. *Small* **19**, 1–38 (2023).
- Driscoll, N. *et al.* MXene-infused bioelectronic interfaces for multiscale electrophysiology and stimulation. *Sci. Transl. Med.* **13**, eabf8629 (2021).
- Liu, H. *et al.* 3D Printed Flexible Strain Sensors: From Printing to Devices and Signals. *Adv. Mater.* **33**, 2004782 (2021).
- Zhu, Z., Park, H. S. & McAlpine, M. C. 3D printed deformable sensors. *Sci. Adv.* **6**, eaba5575 (2023).
- Reeder, J. *et al.* Mechanically Adaptive Organic Transistors for Implantable Electronics. *Adv. Mater.* **26**, 4967–4973 (2014).

Comment 6: On Page 7, Figure 2H, the volume of the PNIPAM hydrogel in Figure 2H is shown to dramatically decrease between approximately 35°C and 45°C. Given that the normal human body temperature is around 36-37°C, even if actuation is induced by an electrothermal method, maintaining its shape seems crucial for attaching it to various organs within the body. To substantiate this point, it would be necessary for the authors to demonstrate volume changes starting from a range lower than 35°C.

Our response: We appreciate the reviewer's comment regarding the temperature-dependent volume changes of PNIPAM hydrogel. As reviewer's suggested, we have included data on volume changes of PNIPAM hydrogel starting from temperatures lower than 35 °C, extending through the normal human body temperature range to beyond its LCST.

As shown in **Fig. R16**, the expanded data clearly demonstrate PNIPAM hydrogel's inherent characteristic of water retention and swelling at temperatures below its LCST, which approximately 32 °C to 34 °C for pure PNIPAM. Beyond this threshold, a significant and rapid deswelling process occurs due to expulsion of water, resulting in a dramatic volume reduction. This thermos-responsive deswelling behavior is the basis for the actuation mechanism used in our device designs.

We have updated the manuscript to better reflect the PNIPAM hydrogel's behavior across relevant physiological temperature ranges. It suggests the hydrogel's applicability and adaptability in developing implantable biomedical devices, capable of dynamically interacting with the human body's thermal environment.

Fig. R16. The thermal responsiveness of PNIPAM hydrogel. (A) The volumetric shrinkage of PNIPAM hydrogel during the heating cycle across a temperature range of 25 °C to 60 °C. (B) The hydrogel's volumetric recovery during the cooling cycle, as temperatures revert to 25 °C, showcasing its reversible thermal behavior and resilience.

Modification to the manuscript:

- (1) On page 8, in the revised manuscript, we added **Fig. R16A** as **Fig. 2H**.
- (2) On page 13, in the revised supporting information, we added **Fig. R16B** as **Fig. S3B**.
- (3) On pages 5-6, in the revised manuscript, we modified, “The PNIPAM hydrogel can undergo a dramatic volumetric reduction of about 90% as the temperature shifts from 25 °C to 60 °C. Notably, this significant and rapid deswelling behavior is initiated only when temperature is beyond its lower critical solution temperature (LCST 32 °C~34 °C), enabling versatile actuation capabilities within biological environments (**Fig. S3A&B**, and **Fig. 2H**).”

Comment 7: It was confirmed that the composite was implemented through XPS analysis. Is there a concern that this might result in lower performance compared to a full composite?

Our response: We appreciate the reviewer's comment regarding the performance of our composite materials, particularly considering its implementation and characterization through XPS analysis. We wish to clarify that our work indeed successfully constructed a full composite tailored specifically for electronic skin (e-skin) applications. Unlike traditional bulk composites, our composite has a thin profile to meet the specific requirements of e-skin functionalities, where excessive thickness would compromise practicality and performance.

Our fabrication process involves an initial step of drop-casting followed by laser patterning, which forms a thin and porous network of the functional materials (**Fig. R17A**). This method contributes to fabricating precise and suitable fine features required for high-performance e-skin. The subsequent step involves spin casting a polymer solution over

this preformed structure. This process ensures the polymer solution thoroughly permeates the functional material layer, resulting in the formation of a full composite (**Fig. R17B**).

Fig. R17. Fabrication of functional nanocomposite based on *in situ* solution integration method. (A) Top: Schematic illustration of patterning AgNWs network with a laser; Bottom: SEM image of as-synthesized AgNWs. (B) Top: Schematic illustration of spin coating PI solution and forming AgNW/PI nanocomposite; Bottom: SEM image of AgNW/PI nanocomposite with full PI infiltration, demonstrating a thorough infiltration of PI within the AgNW network, resulting in a cohesive nanocomposite structure.

The choice of XPS for surface composition and chemical state analysis (**Fig. S13-S15**), supplemented by SEM for morphology insights (**Fig. 2C** and **Fig. 2D**, **Fig. S17A-C**, **Fig. S21A-C**), FTIR for chemical bond identification (**Fig. S17D** and **Fig. S21D**), and XRD for crystalline structure information (**Fig. S17E** and **Fig. S21E**), commonly validates the composite's integrity and the efficacy of the fabrication process (**Supplementary Note S1-S2**).

Our design and fabrication strategies align with the requirement for developing high-performance e-skin layer. By prioritizing a thin profile, we ensure that our composite not only meets the mechanical flexibility and electrical performance demands to e-skin applications but also remains lightweight and conformable to various surfaces. This approach facilitates a seamless integration of functional materials into a polymer matrix, forming a uniform and cohesive composite. Such a composition is critical for the optimal functionality of e-skin, enabling it to closely mimic the sensitivity and versatility of nature skin while ensuring durability and reliability in various operational situations.

In the revised manuscript and supporting information, we have further explained the fabrication process and its advantages, enhancing understanding of our *in situ* solution-based method and its contribution to advancing e-skin technology.

Modification to the manuscript:

On page 7, in the revised manuscript, we modified, “The X-ray photoelectron spectroscopy (XPS) characterization on the e-skin layers reveals the precise nanoscale integration of active materials within a polymer matrix, as detailed in **Fig. S12-S14** and **Supplementary Note S1**. It showcases the optimal distribution and intermolecular bonding of the composite components, effectively addressing the common challenge of uneven dispersion of nanomaterials, which usually undermines the performance of conventional composites. Our approach minimizes the polymer amount required to integrate nanomaterials into composite functional modules and utilizes excess polymer as an insulating layer to separate modules, preventing interference between their electrical and chemical signals, thereby ensuring that each functional module operates independently and effectively. This simple approach combines the distinct properties of each constituent, achieving a balance between structural integrity and functional versatility [56–60]. This advanced level of integration would be of great value for soft robots that seek to achieve multifunctionality and local sensing capabilities approaching skin.”

Comment 8: After nanomaterials are created through the laser pattern, the inherent properties of the material make handling difficult. How was the surrounding material removed or treated?

Our response: We appreciate the reviewer’s concern regarding the post-treatment process surrounding unwanted nanomaterials following laser patterning. Our approach to addressing this challenge hinges on the precise control of laser parameters during the patterning process, which effectively eliminates the need for extensive post-patterning removal steps.

Regarding the laser patterning, the underlying mechanism relies on the laser’s thermal effect, which induces localized heating upon interaction material. This heating is intense and concentrated within a precise area, corresponding to the laser’s focal point. We precisely adjust the laser settings to ensure complete elimination of the nanomaterials outside the desired pattern through hatching process. This laser ablation process, characterized by its high resolution of several tens of micrometers, allows for the selective removal of unwanted nanomaterials while leaving the desired pattern intact. The accuracy and efficiency of this method significantly reduces the presence of residual materials that could potentially interfere with subsequent processing steps. Notably, any residual materials left post-hatching are minimal and do not compromise the structural integrity or functionality of the as-formed composite. By employing this laser patterning technique, we ensure the precise and clean deposition of nanomaterials in alignment with our designs, which is crucial for the operational effectiveness of the device.

We have included the detailed laser patterning parameters applicable for various nanomaterials in the revised manuscript and supporting information, aiming to provide clear insights and facilitate a better understanding for researchers engaging in similar future endeavors.

Modification to the manuscript:

(1) On page 28, in the revised manuscript, we added, “The thermal effects of laser patterning ensure minimal residual materials remain, as localized heating effectively eliminates any leftovers, facilitating seamless progression to subsequent processing steps.”

(2) On page 28, in the revised manuscript, we added, “Here, laser hatching parameters for patterning nanomaterials are set: for AgNW/PEDOT:PSS/RGO, an infrared laser power of 12% (50 W) with a hatching speed of 5000 mm/s and frequency of 40 kHz; for Au/MXene, the settings are adjusted to an infrared laser power of 10% (50 W), a hatching speed of 1000 mm/s, and frequency of 40 kHz, ensuring precision in the material's functional structuring.”

Comment 9: In Figs 3C, D, please indicate in the caption whether SEM images represent the surface or cross-section. Additionally, to determine the presence of voids, it may be necessary to examine cross-sectional TEM images.

Our response: We thank the reviewer's comments. We have updated the captions of **Fig. 3C** and **Fig. 3D** to clearly indicate that these SEM images represent the surface of our samples.

To further address concerns regarding void detection within the composite, we conducted cross-section SEM analysis to examine the presence of voids within the composite. Our findings from the SEM images, as shown in **Fig. R18B&C**, provide a comprehensive understanding of the composite structure, including the void distribution. The SEM images are fully capable of capturing the necessary information regarding voids presence, thus obviating the need for supplementary TEM studies for this purpose.

We have included cross-sectional SEM images in our revised supporting information. These images offer an in-depth view of the composite integrity, allowing us to verify the absence of voids within the material structure, demonstrating the composite's uniformity that is essential for e-skin application requiring high sensitivity and reliability.

Fig. R18. SEM characterization of functional nanocomposite based on *in situ* solution integration method. (A) Top surface SEM view of the AgNW/PI nanocomposite. (B&C) Cross-sectional SEM images of AgNW/PI nanocomposite, revealing the complete penetration of PI throughout the AgNW matrix with the absence of voids.

Modification to the manuscript:

(1) On page 13, in the revised manuscript, we modified the figure caption of **Fig. 3C&D**, “(C&D) SEM images of the surface of nanocomposite films used in constructing sensory robots.”

(2) On page 27, in the revised supporting information, we added **Fig. R18B&C** as **Fig. S17D&F**, respectively.

(3) On page 27, in the revised supporting information, we modified **Fig. S17B&C**.

(4) On page 32, in the revised supporting information, we modified **Fig. S21B&C**.

(3) On pages 9-10, in the revised manuscript, we modified, “The scanning electron microscope (SEM) images, including both top and side views as shown in **Fig. 3C&D**, **Fig. S17A-F**, and **Fig. S21A-C**, present that the curing process fully buries all the nanomaterials inside the PI matrix, resulting in a uniform composite free from observable voids.”

Comment 10: In Fig 3E, it is understood that accurately observing the exact temperature by performing twisting and bending may be challenging with an infrared camera. However, even considering this, it seems that the temperature distribution is not uniform.

Our response: We thank the reviewer’s comment regarding the observed non-uniform temperature distribution in **Fig. 3E**, captured using an infrared camera. As reviewer said, it is challenging to accurately capture temperature variations during dynamic movements such as twisting and bending due to the limitations of thermal imaging technology.

Fig. R19. Infrared thermograph of an AgNW/PI-based heater undergoing bending and twisting motions.

Upon detailed analysis, we identified that the primary cause for the observed non-uniformity was related to the positioning of the sample during imaging. It appears that the sample was not maintained in a perfectly flat orientation nor consistently aligned at a uniform distance from the infrared camera. Such positioning discrepancies can significantly affect the accuracy of temperature measurements, leading to the appearance

of non-uniform temperature distributions due to the angular and distance-related variations inherent in infrared thermography.

To ensure more accurate temperature measurement, we have refined our experimental setup. We made efforts to ensure that the sample is evenly positioned on a flat surface, minimizing any potential distortions or variations due to uneven distances between different parts of the sample and the camera. The updated figures have been included in the revised manuscript.

Modification to the manuscript:

On page 12, in the revised manuscript, we added **Fig. R19** as **Fig. 3E**.

Comment 11: In Fig 3F, G, the temperature converges over time, but what is the reason for the continuous change in bending force? Additionally, once bent, how long does the bending force persist?

Our response: We thank the reviewer's question about the dynamics of temperature stabilization and the resultant bending state as shown in **Fig. R20A (Fig. 3F)** and **Fig. R20B (Fig. 3G)**. **Fig. R20A (Fig. 3F)** presents the surface temperature of the AgNW/PI heater, showing its rapid thermal response and ability to quickly reach and stabilize at the set temperature due to the heater's efficient heat generation capabilities. This rapid thermal stabilization allows for efficient energy transfer and thermal equilibrium.

Fig. R20. (A) Surface temperature of the AgNW/PI-based heater as a function of the input electric power. Notably, the AgNW/PI nanocomposite heater can function under relatively low input electric power. **(B)** The resultant bending angle of a soft robotic arm as a function of the input electric power.

Conversely, **Fig. R20B (Fig. 3G)** shows the actuator's deformation over time. The transmission of heat from the heater to the entire sample requires a certain duration to propagate through the material. This apparent discrepancy in response times between the graphs arises from the difference in thermal propagation and the actuator's

mechanical response. While **Fig. R20A (Fig. 3F)** illustrates the heater's immediate temperature stabilization, **Fig. R20B (Fig. 3G)** reveals a more gradual progression in the actuator's deformation.

Furthermore, it is important to note that **Fig. R20B (Fig. 3G)** shows the deformation state of the actuator within a defined timeframe. Our empirical observations indicate that, upon reaching a thermally stable state, the actuator achieves its maximum deformation. This maximum bending is not transient, instead it is maintained as long as the thermal conditions remain unchanged. This ensures that once activated to a certain threshold, it retains its deformed state indefinitely under steady thermal environments.

To enhance clarity and provide a comprehensive understanding of these phenomena, we have updated **Fig. R20B (Fig. 3G)** in the revised manuscript to include a plateau segment, demonstrating the actuator's deformation state over an extended period. This clearly illustrates the actuator's capability to maintain its maximum bending force upon achieving thermal equilibrium, offering a clearer depiction of its long-term stability and performance.

Modification to the manuscript:

(1) On page 13, in the revised manuscript, we added **Fig. R20B as Fig. 3G**.

(2) On page 10, in the revised manuscript, we added, “ **Fig. 3G** also indicates that the actuator reaches its peak deformation upon achieving thermal equilibrium, and importantly, this maximum bend is maintained as long as there are no changes in thermal conditions.”

Comment 12: In addition to comment 9, is it necessary to keep the heater on continuously to maintain actual bending when attaching it to an organ?

Our response: We appreciate the reviewer's comment regarding the necessity of continuous heating to maintain the actuator's bending state when interfaced with biological tissues/organs. For PNIPAM hydrogel, with LCST between 32-34°C, the normal human body temperature is sufficient to trigger the desired deformation. The initial heating is primarily employed to accelerate the actuation process. Once actuation is achieved, continuous heating using the electrical heater becomes unnecessary. This feature ensures that our device can efficiently adapt and function within the physiological temperature range without the need for sustained thermal input.

Furthermore, we can design the device with a thermal feedback mechanism that automatically adjusts the heating, thereby preventing potential overheating and ensuring a rapid yet safe actuation (**Fig. R10&11**). This built-in feature, combined with hydrogel's responsiveness to body temperature, enables our developing device not only to be effective but also safe for extended biomedical utilization. We have highlighted this point in the revised manuscript.

Modification to the manuscript:

(1) On page 22, in the revised manuscript, we added, “The design employed PNIPAM hydrogel with a LCST 34 °C that is closely aligned with natural body temperature to

achieve necessary shape deformation. The initial heating serves primarily to accelerate the actuation, but after achieving the desired state, continuous electrical heating becomes unnecessary. This feature allows the device to effectively adapt and function within the physiological temperature range without the need for ongoing thermal input.”

(2) On page 36, in the revised supporting information, we added **Fig. R10** as **Fig. S25**.

(3) On page 37, in the revised supporting information, we added **Fig. R11** as **Fig. S26**.

(4) On page 5, in the revised supporting information, we added,“

Supplementary Note S3: The sensory-motor integration within the soft robotic system

The integrated design that leverages on the synergistic interplay between sensors and soft robotic actuators enables volatile adaptivity and responsiveness of the robotic implant in ever-changing conditions. This design allows sensory inputs to directly influence actuator outputs in real-time, creating a feedback loop that allows for automatic adjustment based on environmental stimuli. This feature underscores our system’s responsiveness and adaptability, allowing with the principles of sensor-actuator integration.

As an example, we present an adaptively controlled responsive robotic gripper that maintains optimal actuation temperatures responsive to external temperature changes. The responsive self-adaptation in different thermal environments ensures optimal and predictable heating patterns regardless of conditions, as well as increasing patient safety and preventing potential burn hazard caused by overheating. **Fig. S25A** shows an integrative system that provides temperature sensory readout and, through feedback-controlled power delivery, allows actuation responsive to external thermal environment. **Fig. S25B** demonstrates the working principles of controlling hardware and software. The resistive temperature sensor was connected in series with a reference resistor. The voltage drops over the sensor is read and converted by the on-chip ADC and is processed by the microcontroller to convert to a temperature value. Upon receipt of the temperature value, the control algorithm compares it with the target temperature to produce an error value. The error value is multiplied with a predefined coefficient to produce a corrective factor and is added to the current output value to generate a new output value. The on-chip pulse-width modulation (PWM) module converts the output value to a PWM voltage signal, which is then amplified by the on-board power MOSFET, forming a controller current source to produce a current through the heater. When the external temperature changes, a large corrective factor is produced to allow rapid adaptations. When the target temperature was reached, a constant current is produced to maintain the optimal temperature. **Fig. S26** showed the time-synchronized current and temperature plot during different responsive phases after a sudden change in temperature. During the underheating phase, a rapid increase in output current was generated by the controlled current source in response to the sudden change in temperature. When the target temperature was first reached, the current output remains constant until the actuator was

overheated, during which the current output drops again to reach the static phase at which the current levels off.”

(5) On page 12, in the revised manuscript, we added, “Furthermore, our soft robotic system exemplifies advanced sensory-motor integration, leveraging the synergistic relationship between embedded sensors and actuators to achieve dynamic adaptivity and responsiveness to environmental changes. A prime example is a temperature-sensitive control system, as shown in **Fig. S25A**, which utilizes real-time sensory feedback to dynamically adjust heating in response to environmental temperature changes. The operational principle, as detailed in **Fig. S25B** and **Supplementary Note S3**, involves a microcontroller-driven algorithm that interprets temperature input collected by a resistive temperature sensor, and modulates the electric heater’s current accordingly, enabling rapid adaptations to achieve and maintain a preset temperature. **Fig. S26** presents a soft robotic finger’s real-time response to temperature variations, ensuring stable shape adaptation through this regulatory mechanism. Moreover, this intelligent control significantly improves safety by preventing the risk of overheating, thereby ensuring the system’s safe operation in various thermal conditions, highlighting our device’s ability to provide precise thermal management, enhancing both efficacy and safety in its applications.”

(6) On page 29, in the revised manuscript, we added, “

The sensory-motor integration within the soft robotic system

The robotic gripper and the external circuitry were connected in series with an NI DMM amperemeter set for DC current measurement. The device was cooled to 22 °C in ambient temperature before the system was started up to capture its response to a sudden decrease in ambient temperature. The temperature readouts recorded by the device’s integrated sensor were logged via a microcontroller unit (MCU) and cross-referenced with data from FLIR thermal camera. Both the current and temperature data were analyzed using custom Python script designed specifically for this purpose.”

References

G.C. van Rhoon, T. Samaras, P.S. Yarmolenko, M.W. Dewhurst, E. Neufeld, N. Kuster, CEM43°C thermal dose thresholds: a potential guide for magnetic resonance radiofrequency exposure levels?, *Eur. Radiol.* 23 (2013) 2215–2227. <https://doi.org/10.1007/s00330-013-2825-y>.

M.W. Dewhurst, B.L. Viglianti, M. Lora-Michiels, P.J. Hoopes, M. Hanson, THERMAL DOSE REQUIREMENT FOR TISSUE EFFECT: EXPERIMENTAL AND CLINICAL FINDINGS., *Proc. SPIE--the Int. Soc. Opt. Eng.* 4954 (2003) 37. <https://doi.org/10.1117/12.476637>.

Comment 13: The dual-axis graphs presented in this figure (Figure 3I, Figure 5F, K, N etc.) are difficult to intuitively comprehend. It is unclear whether the changes on the left axis correspond to the results on the right axis or if they were determined through variations in resistance, pressure, or pH values. Therefore, it seems necessary for the authors to modify the graphs for better intuitive understanding and provide explanations regarding the extent of numerical changes made.

Our response: We appreciate the reviewer’s comment on the clarity of dual-axis graphs in our figures (Fig. 3I, Fig. 5F, Fig. 5K and Fig. 5N). These graphs are intended to showcase the correlation between sensor responses (left axis) and the varying external conditions (right axis) they measure, such as temperature, volume, pressure, or pH values.

For instance, Fig. R21A (Fig. 3I) presents the static cycling test of the RGO/PI-based thermal sensor, which operates on resistance changes. In this scenario, as we vary the environmental temperature over time, the resistance changes (indicated on the left y-axis) correspond directly to different temperature values (shown on the right y-axis). Similarly, for Fig. R21B (Fig. 5F), Fig. R21C (Fig. 5K) and Fig. R21D (Fig. 5N), where resistance-based sensors capture real-time changes in resistance as a direct response to external variations in volume, pressure, or pH.

We have refined these figures to more clearly illustrate the correlation between resistance changes and the physical parameters under observation, thereby clarifying the direct linkage between sensor outputs and variations in environmental stimuli (Fig. R21). Moreover, we have incorporated detailed explanations on the numerical changes and their significance into the manuscript to provide comprehensive insights into the sensor’s operational mechanisms.

Fig. R21. (A) Static cycling test of the RGO/PI-based thermal sensor. Here, the left y-axis represents the change in resistance, while the right y-axis corresponds to the associated temperature changes. (B) Representative test of the 3D buckling strain sensor in real-time monitoring of volumetric change of the artificial bladder during cyclic movements of filling and emptying. Here, the left y-axis is the change in resistance, while the right y-axis corresponds to the associated volume changes. (C) Representative measurement of fluidic pressure of the artificial artery system using the soft robotic cuff. Here, the left y-axis is the change in resistance,

while the right y-axis corresponds to the pressure changes. (D) Electrical response of PEDOT:PSS/PVA hydrogel to pH change ranging from 3 to 7 over time. Here, the left y-axis is the change in resistance, while the right y-axis corresponds to the pH value changes.

Modification to the manuscript:

(1) On page 13, in the revised manuscript, we added **Fig. R21A** as **Fig. 3I**.

(2) On page 14, in the revised manuscript, we modified the caption of **Fig. 3I**, “(I) Static cycling test of the RGO/PI-based thermal sensor. Here, the left y-axis represents the change in resistance, while the right y-axis corresponds to the associated temperature changes.”

(3) On page 20, in the revised manuscript, we added **Fig. R21B-D** as **Fig. 5F**, **Fig. 5K**, and **Fig. 5N**, respectively.

(4) On page 21, in the revised manuscript, we modified the caption of **Fig. 5F**, “(F) Representative test of the 3D buckling strain sensor in real-time monitoring of volumetric change of the artificial bladder during cyclic movements of filling and emptying. Here, the left y-axis is the change in resistance, while the right y-axis corresponds to the associated volume changes.”

(5) On page 21, in the revised manuscript, we modified the caption of **Fig. 5K**, “(K) Representative measurement of fluidic pressure of the artificial artery system using the soft robotic cuff. Here, the left y-axis is the change in resistance, while the right y-axis corresponds to the pressure changes.”

(6) On page 21, in the revised manuscript, we modified the caption of **Fig. 5N**, “(N) Electrical response of PEDOT:PSS/PVA hydrogel to pH change ranging from 3 to 7 over time. Here, the left y-axis is the change in resistance, while the right y-axis corresponds to the pH value changes.”

Comment 14: Using heat as a stimulus might potentially cause damage to the tissue. However, considering the hydrogel is thick enough, it may prevent such damage. How efficiently does heat transfer throughout the entire hydrogel, and is the heat dissipation effective enough to be considered safe?

Our response: We appreciate the reviewer’s concern about the use of heat as stimulus and its potential to cause tissue damage. As previously discussed in response to Comment 12, our device does not necessitate continuous heating. We employ initial heating to accelerate the actuation process, eliminating the need for continuous thermal input once the desired actuation is achieved. This strategy ensures that the device’s adaptability and safe operation is within the physiological temperature range.

To further address safety concerns, we can design the device with an integrated thermal feedback mechanism that meticulously regulates the heating, thereby preventing any risk of overheating (**Fig. R10&11**). Moreover, our design considerations ensure that heat transfer through the hydrogel is efficiently managed, maintaining operational temperatures well below the 43 °C threshold known to prevent thermal tissue damage^{13,14}.

Our *in vitro* and *in vivo* studies also confirm the device's safety. *In vitro* exposure of fibroblast-like cells (3T3-J-2) to 39 °C for 48 hours demonstrated maintained cellular health and viability (**Fig. 6E&F**). *In vivo* experiments involving the device's application around a mouse's heart, facilitated by wireless power transfer at 0.1W to achieve 40 °C thermal stimulation, revealed no detectable damage or inflammation in critical organs (**Fig. 4I** and **Fig. S45**).

We have updated the manuscript to highlight these aspects, providing a better understanding of the thermal dynamics and safety profiles of our hydrogel-based systems.

Modification to the manuscript:

(1) On page 22, in the revised manuscript, we added, "The design employed PNIPAM hydrogel with a LCST 34 °C that is closely aligned with natural body temperature to achieve necessary shape deformation. The initial heating serves primarily to accelerate the actuation, but after achieving the desired state, continuous electrical heating becomes unnecessary. This feature allows the device to effectively adapt and function within the physiological temperature range without the need for ongoing thermal input."

(2) On page 36, in the revised supporting information, we added **Fig. R10** as **Fig. S25**.

(3) On page 37, in the revised supporting information, we added **Fig. R11** as **Fig. S26**.

(4) On page 5, in the revised supporting information, we added,"

Supplementary Note S3: The sensory-motor integration within the soft robotic system

The integrated design that leverages on the synergistic interplay between sensors and soft robotic actuators enables volatile adaptivity and responsiveness of the robotic implant in ever-changing conditions. This design allows sensory inputs to directly influence actuator outputs in real-time, creating a feedback loop that allows for automatic adjustment based on environmental stimuli. This feature underscores our system's responsiveness and adaptability, allowing with the principles of sensor-actuator integration.

As an example, we present an adaptively controlled responsive robotic gripper that maintains optimal actuation temperatures responsive to external temperature changes. The responsive self-adaptation in different thermal environments ensures optimal and predictable heating patterns regardless of conditions, as well as increasing patient safety and preventing potential burn hazard caused by overheating. **Fig. S25A** shows an integrative system that provides temperature sensory readout and, through feedback-controlled power delivery, allows actuation responsive to external thermal environment. **Fig. S25B** demonstrates the working principles of controlling hardware and software. The resistive temperature sensor was connected in series with a reference resistor. The voltage drops over the sensor is read and converted by the on-chip ADC and is processed by the microcontroller to convert to a temperature value. Upon receipt of the temperature value, the control algorithm compares it with the target temperature to produce an error

value. The error value is multiplied with a predefined coefficient to produce a corrective factor and is added to the current output value to generate a new output value. The on-chip pulse-width modulation (PWM) module converts the output value to a PWM voltage signal, which is then amplified by the on-board power MOSFET, forming a controller current source to produce a current through the heater. When the external temperature changes, a large corrective factor is produced to allow rapid adaptations. When the target temperature was reached, a constant current is produced to maintain the optimal temperature. **Fig. S26** showed the time-synchronized current and temperature plot during different responsive phases after a sudden change in temperature. During the underheating phase, a rapid increase in output current was generated by the controlled current source in response to the sudden change in temperature. When the target temperature was first reached, the current output remains constant until the actuator was overheated, during which the current output drops again to reach the static phase at which the current levels off.”

(5) On page 12, in the revised manuscript, we added, “Furthermore, our soft robotic system exemplifies advanced sensory-motor integration, leveraging the synergistic relationship between embedded sensors and actuators to achieve dynamic adaptivity and responsiveness to environmental changes. A prime example is a temperature-sensitive control system, as shown in **Fig. S25A**, which utilizes real-time sensory feedback to dynamically adjust heating in response to environmental temperature changes. The operational principle, as detailed in **Fig. S25B** and **Supplementary Note S3**, involves a microcontroller-driven algorithm that interprets temperature input collected by a resistive temperature sensor, and modulates the electric heater’s current accordingly, enabling rapid adaptations to achieve and maintain a preset temperature. **Fig. S26** presents a soft robotic finger’s real-time response to temperature variations, ensuring stable shape adaptation through this regulatory mechanism. Moreover, this intelligent control significantly improves safety by preventing the risk of overheating, thereby ensuring the system’s safe operation in various thermal conditions, highlighting our device’s ability to provide precise thermal management, enhancing both efficacy and safety in its applications.”

(6) On page 29, in the revised manuscript, we added, “

The sensory-motor integration within the soft robotic system

The robotic gripper and the external circuitry were connected in series with an NI DMM amperemeter set for DC current measurement. The device was cooled to 22 °C in ambient temperature before the system was started up to capture its response to a sudden decrease in ambient temperature. The temperature readouts recorded by the device’s integrated sensor were logged via a microcontroller unit (MCU) and cross-referenced with data from FLIR thermal camera. Both the current and temperature data were analyzed using custom Python script designed specifically for this purpose.”

References

G.C. van Rhooen, T. Samaras, P.S. Yarmolenko, M.W. Dewhirst, E. Neufeld, N. Kuster, CEM43°C thermal dose thresholds: a potential guide for magnetic resonance radiofrequency exposure levels?, *Eur. Radiol.* 23 (2013) 2215–2227. <https://doi.org/10.1007/s00330-013-2825-y>.

M.W. Dewhirst, B.L. Viglianti, M. Lora-Michiels, P.J. Hoopes, M. Hanson, THERMAL DOSE REQUIREMENT FOR TISSUE EFFECT: EXPERIMENTAL AND CLINICAL FINDINGS., *Proc. SPIE--the Int. Soc. Opt. Eng.* 4954 (2003) 37. <https://doi.org/10.1117/12.476637>.

Comment 15: While applying temperature, can the thermal sensor accurately measure the temperature from the tissue?

Our response: We appreciate the reviewer’s concern regarding the precision of our thermal sensors in capturing tissue temperatures during application. To validate our sensor’s performance, we conducted comparative tests against commercial thermal resistors (ERT-J0ET102H), demonstrating a high level of correlation and match in the readings, as shown in **Fig. R22A (Fig. 3K)** and **Fig. R22B (Fig. S21G)**.

As illustrated in **Fig. R22C (Fig. 3H)**, we extensively characterized the thermal sensor’s responsiveness across a wide temperature range, from 23 °C to 92 °C, which spans the spectrum of internal body temperature. This evaluation also included scenarios where the sensor was immersed in a phosphate-buffered saline (PBS) solution, closely simulating *in vivo* environmental conditions. The sensor exhibited exceptional stability and reliability even in this simulated physiological environment.

Furthermore, we employed *in situ* solution-based methods to fabricate the functional modules located in different layers. This strategy allows for the integration of a minimal amount of polyimide with functional materials to form composite functional modules. This not only preserves the functional integrity of the materials within each module but also leverages the remaining polyimide to serve as a dual purpose. Beyond its inclusion in the composite, polyimide acts as an insulating barrier between adjacent modules. This barrier is crucial for our device’s functionality, as it prevents electrical and chemical signal interference between modules, thereby ensuring that each functional module operates independently and as intended.

Fig. R22. (A) Temperature measurement on the MXene/PI thermal sensor and a commercial thermal resistor (ERT-J0ET102H). (B) Temperature measurement on the RGO/PI thermal sensor and the commercial thermal resistor. (C) Resistive response at various temperatures ranging from

23 °C to 92 °C, for the RGO/PI-based thermal sensor undergoing bending and twisting motions, and immersed in a solution of PBS.

These findings solidify the capability of our thermal sensor for precise monitoring of tissue temperatures. The updated manuscript further emphasizes the sensor's applicability and stability, highlighting its potential in medical applications.

Modification to the manuscript:

(1) On page 7, in the revised manuscript, we modified, “**Fig. 2M** displays a more complicated integration paradigm with multi-layer stacking, where different electronic components (e.g., PEDOT:PSS/PI-based conductive electrodes and RGO/PI temperature sensors) can be distributed in different layers of the e-skin to achieve simultaneous functional versatility and compactness. This assembly technique ensures the e-skin remarkable thinness and flexibility, enhancing its effective performance for implantable applications (**Fig. S11B**). The X-ray photoelectron spectroscopy (XPS) characterization on the e-skin layers reveals the precise nanoscale integration of active materials within a polymer matrix, as detailed in **Fig. S12-S14** and **Supplementary Note S1**. It showcases the optimal distribution and intermolecular bonding of the composite components, effectively addressing the common challenge of uneven dispersion of nanomaterials, which usually undermines the performance of conventional composites. Our approach minimizes the polymer amount required to integrate nanomaterials into composite functional modules and utilizes excess polymer as an insulating layer to separate modules, preventing interference between their electrical and chemical signals, thereby ensuring that each functional module operates independently and effectively. This simple approach combines the distinct properties of each constituent, achieving a balance between structural integrity and functional versatility. This advanced level of integration would be of great value for soft robots that seek to achieve multifunctionality and local sensing capabilities approaching skin.”

(2) On page 11, in the revised manuscript, we added, “**Fig. 3H** illustrates the resistive change in a relatively linear relation with temperature for the RGO/PI thermal sensor. The temperature coefficient of the resistance (TCR) of the RGO/PI thermal sensor is $> 0.5\%/^{\circ}\text{C}$, featuring its high thermal sensitivity. On the other hand, the RGO/PI-based thermal sensor exhibits a stable performance after 1000 bending cycles, and even after immersing in PBS solution. **Fig. 3I** and **Fig. S21F** show performance of the thermal sensor in response to cycles of temperature rise and drop, indicating good sensing stability. In addition, **Fig. S21G** shows consistent measurements of RGO/PI nanocomposite sensing performance in comparison with a commercial thermal resistor (ERT-J0ET102H), indicating excellent sensing accuracy.”

Comment 16: What is the reason for the difference in the transfer method in S12(b) compared to the dual printing method in S16?

Our response: We appreciate the reviewer’s concern about the methods utilized in our fabrication process, specifically the transfer method illustrated in **Fig. S12B** (**Fig. S11B**

in the updated supporting information), and the dual printing method detailed in **Fig. S16** (**Fig. S15** in the updated supporting information). The essential differentiation between these methods lies not in their fundamental principles but in their application and the resulting properties of the e-skin layer. The choice of method is tailored to the specific requirements of the application, ensuring that the e-skin fabricated is not only functionally effective but also optimally designed for its intended use.

The transfer method, as shown in **Fig. S12B** (**Fig. S11B** in the updated supporting information), allows for the fabrication of an e-skin that is notably thinner and possesses enhanced flexibility. This is especially beneficial for implantable applications where e-skin needs to conform closely to the complex contour of tissues/organs, ensuring optimal functionality and integration. Conversely, the dual printing method offers advantages in producing e-skin with multiple layers, allowing for the incorporation of multiple functionalities within a single e-skin platform.

Both methods demonstrate the versatility and adaptability of our solution-based fabrication strategy, highlighting its superiority over traditional fabrication techniques, such as 3D printing. These two methods indicate the operational ease and the broad applicability of our method in creating e-skins that can be tailored to meet the diverse requirements of both implantable devices and wearable sensors, thereby broadening the scope of potential applications in the field of soft robotics and biomedical devices.

In the revised manuscript, we have clarified these two methods, further emphasizing our approach's flexibility in accommodating the needs of advanced e-skin systems.

Modification to the manuscript:

(1) On page 7, in the revised manuscript, we modified, "**Fig. 2M** displays a more complicated integration paradigm with multi-layer stacking, where different electronic components (e.g., PEDOT:PSS/PI-based conductive electrodes and RGO/PI temperature sensors) can be distributed in different layers of the e-skin to achieve simultaneous functional versatility and compactness. This assembly technique ensures the e-skin remarkable thinness and flexibility, enhancing its effective performance for implantable applications (**Fig. S11B**)."

(2) On page 9, in the revised manuscript, we modified, "**Fig. S15A** shows layer-by-layer stacking as a simple and effective approach for fabricating the e-skin. This approach stands out for its capability to fabricate multi-layered e-skin integrating diverse functionalities within an unified e-skin framework, offering a sophisticated level of device customization."

Comment 17: On Page 11, the author mentioned that "The electrothermal stimulus along with distributed sensing capabilities enables programmed actuation not only on demand but also regulated simultaneously by the sensing feedback". How was actuation implemented based on sensing feedback?

Our response: We thank the reviewer's question regarding the implementation of actuation based on sensing feedback. Our system design integrates a network of distributed sensors within e-skin, enabling the detection of various environmental stimuli. Upon detecting a change in external environment, these sensors convert environmental changes into electrical signals relayed to a control unit.

We have demonstrated that the integrated design that leverages on the synergistic interplay between sensors and soft robotic actuators enables volatile adaptivity and responsiveness of the robotic implant in ever-changing conditions. This design allows sensory inputs to directly influence actuator outputs in real-time, creating a feedback loop that allows for automatic adjustment based on environmental stimuli. This feature underscores our system's responsiveness and adaptability, allowing with the principles of sensor-actuator integration highlighted by the reviewer.

As an example, we present an adaptively controlled responsive robotic gripper that maintains optimal actuation temperatures responsive to external temperature changes. The responsive self-adaptation in different thermal environments ensures optimal and predictable heating patterns regardless of conditions, as well as increasing patient safety and preventing potential burn hazard caused by overheating. **Fig. R10A** shows an integrative system that provides temperature sensory readout and, through feedback-controlled power delivery, allows actuation responsive to external thermal environment. **Fig. R10B** demonstrates the working principles of controlling hardware and software. The resistive temperature sensor was connected in series with a reference resistor. The voltage drops over the sensor is read and converted by the on-chip ADC and is processed by the microcontroller to convert to a temperature value. Upon receipt of the temperature value, the control algorithm compares it with the target temperature to produce an error value. The error value is multiplied with a predefined coefficient to produce a corrective factor and is added to the current output value to generate a new output value. The on-chip pulse-width modulation (PWM) module converts the output value to a PWM voltage signal, which is then amplified by the on-board power MOSFET, forming a controller current source to produce a current through the heater. When the external temperature changes, a large corrective factor is produced to allow rapid adaptations. When the target temperature was reached, a constant current is produced to maintain the optimal temperature. **Fig. R11** showed the time-synchronized current and temperature plot during different responsive phases after a sudden change in temperature. During the underheating phase, a rapid increase in output current was generated by the controlled current source in response to the sudden change in temperature. When the target temperature was first reached, the current output remains constant until the actuator was overheated, during which the current output drops again to reach the static phase at which the current levels off.

To clarify this feedback mechanism, we have included comprehensive details in the revised manuscript and supporting information, providing examples of how this feedback mechanism is implemented within our soft robotic system.

Modification to the manuscript:

- (1) On page 36, in the revised supporting information, we added **Fig. R10** as **Fig. S25**.
- (2) On page 37, in the revised supporting information, we added **Fig. R11** as **Fig. S26**.
- (3) On page 5, in the revised supporting information, we added,

Supplementary Note S3: The sensory-motor integration within the soft robotic system

The integrated design that leverages on the synergistic interplay between sensors and soft robotic actuators enables volatile adaptivity and responsiveness of the robotic implant in ever-changing conditions. This design allows sensory inputs to directly influence actuator outputs in real-time, creating a feedback loop that allows for automatic adjustment based on environmental stimuli. This feature underscores our system's responsiveness and adaptability, allowing with the principles of sensor-actuator integration.

As an example, we present an adaptively controlled responsive robotic gripper that maintains optimal actuation temperatures responsive to external temperature changes. The responsive self-adaptation in different thermal environments ensures optimal and predictable heating patterns regardless of conditions, as well as increasing patient safety and preventing potential burn hazard caused by overheating. **Fig. S25A** shows an integrative system that provides temperature sensory readout and, through feedback-controlled power delivery, allows actuation responsive to external thermal environment. **Fig. S25B** demonstrates the working principles of controlling hardware and software. The resistive temperature sensor was connected in series with a reference resistor. The voltage drops over the sensor is read and converted by the on-chip ADC and is processed by the microcontroller to convert to a temperature value. Upon receipt of the temperature value, the control algorithm compares it with the target temperature to produce an error value. The error value is multiplied with a predefined coefficient to produce a corrective factor and is added to the current output value to generate a new output value. The on-chip pulse-width modulation (PWM) module converts the output value to a PWM voltage signal, which is then amplified by the on-board power MOSFET, forming a controller current source to produce a current through the heater. When the external temperature changes, a large corrective factor is produced to allow rapid adaptations. When the target temperature was reached, a constant current is produced to maintain the optimal temperature. **Fig. S26** showed the time-synchronized current and temperature plot during different responsive phases after a sudden change in temperature. During the underheating phase, a rapid increase in output current was generated by the controlled current source in response to the sudden change in temperature. When the target temperature was first reached, the current output remains constant until the actuator was overheated, during which the current output drops again to reach the static phase at which the current levels off."

(4) On page 12, in the revised manuscript, we added, “Furthermore, our soft robotic system exemplifies advanced sensory-motor integration, leveraging the synergistic relationship between embedded sensors and actuators to achieve dynamic adaptivity and responsiveness to environmental changes. A prime example is a temperature-sensitive control system, as shown in **Fig. S25A**, which utilizes real-time sensory feedback to dynamically adjust heating in response to environmental temperature changes. The operational principle, as detailed in **Fig. S25B** and **Supplementary Note S3**, involves a microcontroller-driven algorithm that interprets temperature input collected by a resistive temperature sensor, and modulates the electric heater’s current accordingly, enabling rapid adaptations to achieve and maintain a preset temperature. **Fig. S26** presents a soft robotic finger’s real-time response to temperature variations, ensuring stable shape adaptation through this regulatory mechanism. Moreover, this intelligent control significantly improves safety by preventing the risk of overheating, thereby ensuring the system’s safe operation in various thermal conditions, highlighting our device’s ability to provide precise thermal management, enhancing both efficacy and safety in its applications.”

(5) On page 29, in the revised manuscript, we added, “

The sensory-motor integration within the soft robotic system

The robotic gripper and the external circuitry were connected in series with an NI DMM amperemeter set for DC current measurement. The device was cooled to 22 °C in ambient temperature before the system was started up to capture its response to a sudden decrease in ambient temperature. The temperature readouts recorded by the device’s integrated sensor were logged via a microcontroller unit (MCU) and cross-referenced with data from FLIR thermal camera. Both the current and temperature data were analyzed using custom Python script designed specifically for this purpose.”

References

G.C. van Rhoon, T. Samaras, P.S. Yarmolenko, M.W. Dewhurst, E. Neufeld, N. Kuster, CEM43°C thermal dose thresholds: a potential guide for magnetic resonance radiofrequency exposure levels?, *Eur. Radiol.* 23 (2013) 2215–2227. <https://doi.org/10.1007/s00330-013-2825-y>.

M.W. Dewhurst, B.L. Viglianti, M. Lora-Michiels, P.J. Hoopes, M. Hanson, THERMAL DOSE REQUIREMENT FOR TISSUE EFFECT: EXPERIMENTAL AND CLINICAL FINDINGS., *Proc. SPIE--the Int. Soc. Opt. Eng.* 4954 (2003) 37. <https://doi.org/10.1117/12.476637>.

Comment 18: On Page 14, Figure S31, in Figure S31B, it is evident that there is a tendency for the coupling coefficient to decrease as the separation distance increases. However, in Figure S31D, there is a trend showing an increase in output power relative to input power as the distance increases. This observation appears to be theoretically inconsistent. If this graph is accurate, the authors need to provide additional explanations for these results.

Our response: We thank the reviewer’s comment regarding the discrepancy in the coupling efficiency and output power trends with increasing separation distance, as

shown in **Fig. R23A (Fig. S31B)** and **Fig. R23B (Fig. S31D)**. After thorough review, we found a labeling error in **Fig. S31D** that misrepresented the trend of output power relative to input power as the distance increases.

Fig. R23. The effect of separation distance, horizontal offset and orientation angle on the performance of WPT system. (A) FEA results showing the variations in the coupling coefficient of the WPT coils under different separation distances. **(B)** Experimental evaluations of power transmission at resonance frequency (16 MHz) under different separation distances

We have corrected this error to accurately illustrate the anticipated decrease in output efficiency with an increase in separation distance, thus realigning our results with theoretical expectations and ensuring consistency across our data presentation.

Modification to the manuscript:

On page 42, in the revised supporting information, we revised **Fig. S31D**.

Comment 19: On Page 16, in order for the sensory robots developed by the authors to conformally adhere to tissues or organs, they require not only curvilinear surfaces but also effective adhesion. While this article mentions the adhesiveness of PNIPAM, a recently published paper indicates that PNIPAM shows tissue adhesion at low temperatures (around 25°C) but loses almost all adhesive strength when it transitions to a gel form near 40 °C (Nature Biotechnology 2023, 41, 652-662). To showcase the tissue adhesion capability of the authors' device, it becomes imperative to gauge the adhesive strength of the hydrogel by varying temperatures. This would confirm that there are no issues with adhesion even after attaching it to an organ.

Our response: We thank the reviewer's comment regarding the adhesive properties of PNIPAM hydrogels, particularly in light of recent literature indicating significant changes in adhesive strength at varying temperatures¹⁵. It is crucial to distinguish the type of adhesion we refer to within the scope of our device's application from the strong adhesive forces comparable to those of glues, as discussed in the mentioned Nature Biotechnology paper¹⁵.

Our device relies on the shape adaptability induced by the thermal response of PNIPAM hydrogels rather than on adhesive forces for effective tissue interfacing where the former

self-assembles into a conformal shape to reduce dependence on strong adhesion force and mitigate interfacial stress, thus increasing biocompatibility. The concern regarding adhesion arises in the context of ensuring that the implanted device maintains its position and functionality without being compromised by potential reswelling effects post-actuation, especially at the temperatures around the body's physiological range.

Our approach capitalizes on the unique temperature-responsive deformation properties of PNIPAM hydrogels to maintain device efficacy, emphasizing a synergistic integration of shape adaptability and minimal adhesion to secure tissue engagement. Our updated manuscript and supporting information now more clearly illustrate this perspective.

Fig. R24. Demonstration of hydrogel adhesion to biological tissues. (A&B) Adhesion behavior of a PNIPAM-co-PAAm hydrogel with an LCST of 36 °C across different temperatures. Below the LCST, the hydrogel exhibits robust adhesion below its LCST, which gradually loses above this threshold. All tests were conducted after overnight incubation simulating physiological conditions at 34 °C, 37 °C, and 40 °C. Scale bars, 5 mm.

Additionally, as the reviewer's suggested, we conducted additional characterizations of the adhesive force exhibited by a PNIPAM-co-PAAm hydrogel with an LCST of 36 °C across a range of temperatures, providing a better understanding of the hydrogel's physical properties and aiming to contribute valuable insights to the development of hydrogels for biomedical applications in designing effective implantable devices.

Modification to the manuscript:

(1) On page 10, in the revised manuscript, we modified, "We further evaluated the mechanical force generated by the soft robotic finger which incorporates a PNIPAM hydrogel layer roughly 1 mm thick, under various input powers. **Fig. S19B** shows that the static force exhibits a noticeable increase with rising temperature. At a temperature of 40 °C, the force reaches a maximum of 32 mN. Additionally, it is observed that the generated force remains consistent throughout 40 cycles of alternating power on and off (0.35 W), indicating the robust reversibility of the soft robot (**Fig. S19C**). When compared to similar

hydrogel-based soft actuators, our design consistently achieves a relatively high output force, as shown in **Table S1**.”

(2) On page 52, in the revised supporting information, we added **Fig. R24** as **Fig. S40**.

(3) On pages 21-22, in the revised manuscript, we added, “We also explored the bioadhesive behavior of our device on targeted tissues/organs. We observed that hydrogel’s inherent adhesiveness is significantly related to its water content and temperature. As shown in **Fig. S40**, there is a decline in adhesive strength as temperatures approach the hydrogel’s LCST. While this inherent adhesive capability contributes to the initial secure placement of the device, it’s noteworthy that solely relying on this property might not guarantee a durable bond, especially as the hydrogel experiences dehydration. However, this temperature-responsive adhesiveness can play a complementary role in enhancing the device’s grasp by counterbalancing any potential decrease in force due to hydrogel reswelling.”

(6) On page 29, in the revised manuscript, we added, “Adhesion force was tested by the standard 180° peel test with the Instron machine (Mark-10 ESM303). All tests were conducted with a constant peeling speed of 13 mm/min.”

Reference

Y. Jiang, A.A. Trotsyuk, S. Niu, D. Henn, K. Chen, C.C. Shih, M.R. Larson, A.M. Mermin-Bunnell, S. Mittal, J.C. Lai, A. Saberi, E. Beard, S. Jing, D. Zhong, S.R. Steele, K. Sun, T. Jain, E. Zhao, C.R. Neimeth, W.G. Viana, J. Tang, D. Sivaraj, J. Padmanabhan, M. Rodrigues, D.P. Perrault, A. Chattopadhyay, Z.N. Maan, M.C. Leeolou, C.A. Bonham, S.H. Kwon, H.C. Kussie, K.S. Fischer, G. Gurusankar, K. Liang, K. Zhang, R. Nag, M.P. Snyder, M. Januszyk, G.C. Gurtner, Z. Bao, Wireless, closed-loop, smart bandage with integrated sensors and stimulators for advanced wound care and accelerated healing, *Nat. Biotechnol.* 41 (2023) 652–662. <https://doi.org/10.1038/s41587-022-01528-3>.

Comment 20: In Figs 5, 6, the use of Au electrodes for electrical stimulation in electrotherapy seems somewhat divergent from the other sensing materials of this paper (Fig. S52B).

Our response: We thank the reviewer’s concern regarding the use of Au electrodes for electrical stimulation in electrotherapy, as depicted in **Fig. 5** and **Fig. 6**. The incorporation of Au electrodes within our device exemplifies our design to functional diversity and integration. The choice of Au, known for its superior electrical conductivity, biocompatibility, and stability within biological environments. These attributes render them ideally suited for delivering reliable and precise electrical stimulation, which is a critical function in therapeutic applications.

Our study explores a variety of materials with notable sensing and actuation capabilities. The selection of Au electrodes for demonstration emphasizes the system’s modular design. This showcases our device’s ability to seamlessly integrate various functional materials, enhancing its applicability across a wide range of therapeutic and monitoring scenarios. This adaptability allows for the customization of materials in alignment with

distinct application requirements, thereby providing a versatile platform that can be tailored to a broad spectrum of biomedical applications.

Modification to the manuscript:

(1) On page 44, in the revised supporting information (**Fig. S33A**'s caption), we added, "The stimulator component is two pairs of pacing electrodes, employing Au for its exceptional electrical conductivity, biocompatibility, and stability in biological settings."

(2) On page 53, in the revised supporting information (**Fig. S41**'s caption), we added, "The use of Au ensures superior electrical conductivity, biocompatibility, and stability within biological environments, guaranteeing reliable and precise electrical stimulation essential for effective therapeutic interventions."

Comment 21: In Fig 5G, the electrical stimulation off voltage is not 0.0 V. Is there a specific reason for this? To help readers to understand, it appears that the authors should provide additional explanations regarding the correlation between electrical stimulation and volume changes. Specifically, a clear clarification is needed on whether the specific electrical stimulation is causing an actual increase in volume or if it is transmitting signals to patients having urinary dysfunction.

Our response: We thank the reviewer's questions regarding the electrical stimulation off voltage in **Fig. 5G** and the need for additional explanations about the correlation between electrical stimulation and volume changes. We have come into realization that in the previous experiment, the voltage electrical stimulation voltage was measured without a MOSFET power buffer. The high internal resistance of the microcontroller GPIO pins has resulted in fluctuations in the voltage measured at shut-off state. Through the application of a MOSFET power buffer, we have successfully decreased the amount of current drawn at the GPIO pin and decreased the internal resistance of the electrical stimulation output, and thereby corrects the small offset observed at shut-off state (**Fig. R25**).

Fig. R25. A robotic gripper for bladder control. Programmed electrical stimulation (top) and measured volume of an artificial bladder based on a balloon (middle and bottom). The experimental demonstration is conducted using the following parameters: volume threshold of ~100 mL, electrical stimulation amplitude of 3 V. Here, a slight delay in the deactivation process could be partially attributed to the response time of microcontroller unit (MCU) system in detecting changes from the sensors and updating the output accordingly.

Electrical stimulation therapy has been used in various forms to manage bladder dysfunction, including sacral nerve stimulation (SNS), posterior tibial nerve stimulation (PTNS), and direct stimulation of the bladder or surrounding tissues^{16,17}. Our device utilizes direct stimulation on the bladder muscle to facilitate bladder contraction and assist in urination, addressing conditions where bladder muscle control is involuntary and the sensory feedback to the central nervous system is disrupted.

Our developed soft robotic gripper provides both real-time bladder volume assessment and voiding treatment in a wireless closed-loop control manner. It features a flexible hydrogel-based actuator for conformal wrapping around the bladder, an integrated strain sensor for continuous bladder volume detection, an electrical stimulator for on-demand electrotherapy, and a control module for programmed operation (**Fig. 5A-G, Fig. S33-35**). Electrical stimulation (E-stim) is activated when the bladder reaches its maximum volume threshold, targeting the bladder muscles to facilitate urination. Upon complete voiding of the bladder, the integrated sensors detect the reduction in bladder volume, which subsequently triggers the deactivation of the E-stim. This programming cycle ensures that stimulation is provided only, when necessary, closely mimicking the natural urination process and enhancing patient comfort and device efficiency.

In several studies and clinical trials, electrical stimulation has shown positive effects on bladder control functions for some patients. However, the effectiveness of this treatment may vary among individuals and requires evaluation and supervision by healthcare professionals based on the patient's specific condition. The effectiveness of electrical stimulation for bladder voiding and its required voltage levels requires further investigation beyond the scope of our current study. However, our system demonstrates the feasibility of integrating sensing and actuation components to facilitate timely and responsive treatments for bladder dysfunction.

In the revised manuscript, we have clarified these points and provided additional explanations regarding the operational principles of our device, its potential applications in bladder dysfunction treatment. Our goal is to contribute to the broader understanding and development of integrative solutions for managing urinary disorders, emphasizing the significance of continuous monitoring and responsive treatment mechanisms in the field of soft robotics and biomedical engineering.

Modification to the manuscript:

(1) On page 20, in the revised manuscript, we added **Fig. R5** as **Fig. 5G**.

(2) On page 21, in the revised manuscript, we modified the figure caption of **Fig. 5G**, "Programmed electrical stimulation (top) and measured volume of an artificial bladder based on a balloon (middle and bottom). The experimental demonstration is conducted using the following parameters: volume threshold of ~100 mL, electrical stimulation amplitude of 3 V. Here, a slight delay in the deactivation process could be partially attributed to the response time of microcontroller unit (MCU) system in detecting changes from the sensors and updating the output accordingly."

(3) On page 18, in the revised manuscript, we added, "When the balloon's volume reaches a predetermined threshold, set here at 100 mL, the control system initiates electrical stimulation. Following successful voiding to below the threshold, the system automatically deactivates the stimulation. While electrical stimulation has shown promising results in enhancing bladder control in various studies and clinical trials, its efficacy can differ across individuals. The effectiveness of electrical stimulation for bladder voiding and its required voltage levels requires further investigation beyond the scope of our current study. However, our prototype showcases the potential of integrating sensing and actuation mechanisms to facilitate timely and adaptive interventions for bladder dysfunction."

References

Grill, W. M. Electrical stimulation for control of bladder function. Proc. 31st Annu. Int. Conf. IEEE Eng. Med. Biol. Soc. Eng. Futur. Biomed. EMBC 2009 2369–2370 (2009) doi:10.1109/IEMBS.2009.5335001.

Coolen, R. L., Groen, J. & Blok, B. F. M. Electrical stimulation in the treatment of bladder dysfunction: Technology update. Med. Devices Evid. Res. 12, 337–345 (2019).

Comment 22: As the authors mentioned in the context, PNIPAM hydrogel has reversibly contract and relax property upon electrothermal trigger. If that's the case, is there no risk of delamination if the device is not well-adhered to the tissue when the internal body temperature decreases? Additionally, wouldn't there be a possibility that continuous electrothermal stimulation, applied to prevent deformation of the device after it is attached to the tissue, could have adverse effects on the tissue?

Our response: We appreciate the reviewer's concerns regarding the potential for delamination and the impact of continuous electrothermal stimulation with the use of PNIPAM hydrogel in our device. The inherent reversible contraction and relaxation properties of PNIPAM hydrogel, activated by electrothermal triggers, are indeed central to our device's operational design.

In our study, we utilize a bilayer structure for our soft robotic designs, where bending motion is primarily driven by the anisotropic response to the temperature change. As shown in **Fig. S2B**, our bilayer system mainly consists of two layers: an electronic skin (e-skin) layer and a thermal-responsive PNIPAM hydrogel artificial muscle layer. The PNIPAM undergoes volume and phase transitions in response to temperature changes. Conversely, the e-skin layer serves as a static, non-responsive layer that does not undergo deformation, thereby providing a constraint to the expansive or contractive movements of the PNIPAM layer. Upon surpassing PNIPAM's lower critical solution temperature (LCST), typically within the range of 32-34 °C, a threshold easily reached by body natural temperature (~37 °C), the device exhibits a bending motion towards the hydrogel layer, achieving a sustained shape deformation under these conditions.¹⁸⁻²⁰ This dynamic is facilitated by the distinct material properties of both layers, allowing for significant shape transformation while maintaining the device's overall flexibility. Therefore, the device design ensures tissue integration through the mechanical architecture that promotes conformal contact with tissues for sensing, communication, or control systems.

We can incorporate an integrated thermal feedback system that precisely regulates the device's heating to further address concerns of delamination, especially in scenarios where body temperature significantly diverges from the LCST (32~34 °C). **Fig. R10A** shows our adaptively controlled robotic gripper, engineered to autonomously maintain the required actuation temperature under varying environmental conditions. The system is configured to activate the heater to preserve the actuation temperature whenever external temperatures fall below the LCST. **Fig. R11** details the synchronization of current and temperature during thermal adjustments, highlighting our device's capability to rapidly respond to temperature variations and maintain stability thereafter.

Regarding the concern of continuous electrothermal stimulation, we can carefully control the duration and intensity of the stimulation to remain within safe limits. We have conducted both *in vivo* and *in vitro* evaluations to confirm the device's safety. *In vitro* analysis involving fibroblast-like cells (3T3-J-2) subjected to a 39°C environment for 48 hours showed no compromise in cell health or viability (**Fig. 6E-F**). Similarly, *in vivo*

application involving thermal stimulation around the mice's heart revealed no significant tissue damage or inflammatory response (**Fig. S45**). While our studies have primarily explored the effect of a few hours of continuous heating, both theoretical and empirical evidence suggest that as long as the temperature is maintained below the critical threshold of 43 °C based on previous studies^{13,14}, a risk of thermal damage to tissues can be effectively avoided. As aforementioned, the integration of a thermal feedback mechanism further ensures precise heating control, thereby preventing any risk of overheating or insufficient heating (**Fig. R10&11**).

In the revised manuscript, we have provided an enhanced clarification on the operational principles of our device, emphasizing its safety and reliability profile. We hope to offer a clearer understanding of the mechanisms underlying our device's functionality, and the applicability in biomedical application through engineering solutions.

Modification to the manuscript:

(1) On page 10, in the revised manuscript, we modified, "We further evaluated the mechanical force generated by the soft robotic finger which incorporates a PNIPAM hydrogel layer roughly 1 mm thick, under various input powers. **Fig. S19B** shows that the static force exhibits a noticeable increase with rising temperature. At a temperature of 40 °C, the force reaches a maximum of 32 mN. Additionally, it is observed that the generated force remains consistent throughout 40 cycles of alternating power on and off (0.35 W), indicating the robust reversibility of the soft robot (**Fig. S19C**). When compared to similar hydrogel-based soft actuators, our design consistently achieves a relatively high output force, as shown in **Table S1**."

(2) On page 22, in the revised manuscript, we added, "The design employed PNIPAM hydrogel with a LCST 34 °C that is closely aligned with natural body temperature to achieve necessary shape deformation. The initial heating serves primarily to accelerate the actuation, but after achieving the desired state, continuous electrical heating becomes unnecessary. This feature allows the device to effectively adapt and function within the physiological temperature range without the need for ongoing thermal input."

(3) On page 36, in the revised supporting information, we added **Fig. R10** as **Fig. S25**.

(4) On page 37, in the revised supporting information, we added **Fig. R11** as **Fig. S26**.

(5) On page 5, in the revised supporting information, we added,"

Supplementary Note S3: The sensory-motor integration within the soft robotic system

The integrated design that leverages on the synergistic interplay between sensors and soft robotic actuators enables volatile adaptivity and responsiveness of the robotic implant in ever-changing conditions. This design allows sensory inputs to directly influence actuator outputs in real-time, creating a feedback loop that allows for automatic adjustment based on environmental stimuli. This feature underscores our system's

responsiveness and adaptability, allowing with the principles of sensor-actuator integration.

As an example, we present an adaptively controlled responsive robotic gripper that maintains optimal actuation temperatures responsive to external temperature changes. The responsive self-adaptation in different thermal environments ensures optimal and predictable heating patterns regardless of conditions, as well as increasing patient safety and preventing potential burn hazard caused by overheating. **Fig. S25A** shows an integrative system that provides temperature sensory readout and, through feedback-controlled power delivery, allows actuation responsive to external thermal environment. **Fig. S25B** demonstrates the working principles of controlling hardware and software. The resistive temperature sensor was connected in series with a reference resistor. The voltage drops over the sensor is read and converted by the on-chip ADC and is processed by the microcontroller to convert to a temperature value. Upon receipt of the temperature value, the control algorithm compares it with the target temperature to produce an error value. The error value is multiplied with a predefined coefficient to produce a corrective factor and is added to the current output value to generate a new output value. The on-chip pulse-width modulation (PWM) module converts the output value to a PWM voltage signal, which is then amplified by the on-board power MOSFET, forming a controller current source to produce a current through the heater. When the external temperature changes, a large corrective factor is produced to allow rapid adaptations. When the target temperature was reached, a constant current is produced to maintain the optimal temperature. **Fig. S26** showed the time-synchronized current and temperature plot during different responsive phases after a sudden change in temperature. During the underheating phase, a rapid increase in output current was generated by the controlled current source in response to the sudden change in temperature. When the target temperature was first reached, the current output remains constant until the actuator was overheated, during which the current output drops again to reach the static phase at which the current levels off.”

(6) On page 12, in the revised manuscript, we added, “Furthermore, our soft robotic system exemplifies advanced sensory-motor integration, leveraging the synergistic relationship between embedded sensors and actuators to achieve dynamic adaptivity and responsiveness to environmental changes. A prime example is a temperature-sensitive control system, as shown in **Fig. S25A**, which utilizes real-time sensory feedback to dynamically adjust heating in response to environmental temperature changes. The operational principle, as detailed in **Fig. S25B** and **Supplementary Note S3**, involves a microcontroller-driven algorithm that interprets temperature input collected by a resistive temperature sensor, and modulates the electric heater’s current accordingly, enabling rapid adaptations to achieve and maintain a preset temperature. **Fig. S26** presents a soft robotic finger’s real-time response to temperature variations, ensuring stable shape adaptation through this regulatory mechanism. Moreover, this intelligent control significantly improves safety by preventing the risk of overheating, thereby ensuring the system’s safe operation in various thermal conditions, highlighting our device’s ability to

provide precise thermal management, enhancing both efficacy and safety in its applications.”

(7) On page 29, in the revised manuscript, we added, “

The sensory-motor integration within the soft robotic system

The robotic gripper and the external circuitry were connected in series with an NI DMM amperemeter set for DC current measurement. The device was cooled to 22 °C in ambient temperature before the system was started up to capture its response to a sudden decrease in ambient temperature. The temperature readouts recorded by the device’s integrated sensor were logged via a microcontroller unit (MCU) and cross-referenced with data from FLIR thermal camera. Both the current and temperature data were analyzed using custom Python script designed specifically for this purpose.”

References

G.C. van Rhooen, T. Samaras, P.S. Yarmolenko, M.W. Dewhurst, E. Neufeld, N. Kuster, CEM43°C thermal dose thresholds: a potential guide for magnetic resonance radiofrequency exposure levels?, *Eur. Radiol.* 23 (2013) 2215–2227. <https://doi.org/10.1007/s00330-013-2825-y>.

M.W. Dewhurst, B.L. Viglianti, M. Lora-Michiels, P.J. Hoopes, M. Hanson, THERMAL DOSE REQUIREMENT FOR TISSUE EFFECT: EXPERIMENTAL AND CLINICAL FINDINGS., *Proc. SPIE--the Int. Soc. Opt. Eng.* 4954 (2003) 37. <https://doi.org/10.1117/12.476637>.

Comment 23: On Page 24, Figure 6F, what is the indication T1 and T2 in Figure 6F account for? The authors need to explain these in the captions.

Our response: We thank the reviewer’s questions regarding the indications of T1 and T2 in **Fig. 6F (Fig. 6G** in the updated manuscript). T1 and T2 represent two distinct channels, each linked to a thermal sensor positioned on separate arms of our cardiac gripper, as illustrated in **Fig. 6A** and **Fig. S41A&B**. These sensors are designed to simultaneously capture the heart’s temperature, facilitating a detailed assessment of temperature fluctuation across different segments of the organ. The multi-channel design exemplifies our device’s capability for precise and localized temperature monitoring that is essential for a range of biomedical applications.

Fig. R26. (A) schematic illustration of the soft robotic thera-gripper consisting of two temperature sensors made of thermal resistors. (B) Temperature measurements from the thera-gripper’s dual-channel sensors during its deployment onto the mouse heart, demonstrating the device’s capability to monitor thermal variations in real-time.

Modification to the manuscript:

- (1) On page 24, in the revised manuscript, we added **Fig. R26B** as **Fig. 6G**.
- (2) On page 25, in the revised manuscript (**Fig. 6G**’s caption), we modified, “Temperature measurements from the thera-gripper’s dual-channel sensors during its deployment onto the mouse heart, demonstrating the device’s capability to monitor thermal variations in real-time.”

Comment 24: In Fig 6G, it is challenging to consider it as functioning like a pacemaker, as pacing does not seem to lead to actual cardiac capture.

Our response: We thank the reviewer’s concern regarding the pacing functionality described for our cardiac gripper. We agree that our gripper’s electrical stimulation electrodes do not function in the traditional sense of a pacemaker. As correctly pointed out, our device introduces a general concept of an electrical stimulator module (**Fig. R27A**), designed to cater to a broader spectrum of potential therapeutic applications that can benefit from electrical stimulation (E-stim) (e.g., enhancing myocardial contractility), beyond the specific function of pacing^{21,22}. Our in vivo studies have demonstrated that the gripper’s ability to successfully deliver various E-stim patterns (**Fig. R27B-D**), showing its potential in therapeutic settings.

To address this point and clarify our intention, we have revised the descriptions within our manuscript and adjusted the annotations in the figures (**Fig. R27B**) accordingly. This modification aims to explore the versatility of our soft robotic gripper in delivering targeted electrical stimulation for various therapeutic scenarios.

Fig. R27. (A) Flexible pacing electrodes implantation. (B) The surface ECG trace during electrical stimulation using a pair of Au pacing electrodes. (C&D) Representative voltage traces of the cardiac electrical activity during E-stim with various parameters. (C) 500 mV voltage with 1 ms width at 6.5 Hz. (D) 500 mV voltage with 5 s width at 0.1 Hz.

Modification to the manuscript:

- (1) On page 24, in the revised manuscript, we added **Fig. R27B** as **Fig. 6H**.
- (2) On pages 22-23, in the revised manuscript, we modified, “. The e-skin layer consists of microelectrodes for capturing electrical activity of the heart, which serves as essential guidance in operating electrical stimulation (**Fig. S47**). **Fig. S48&S49** showcase the simultaneous sensing and stimulation capabilities on a beating heart with an *in vivo* mouse model, demonstrating its capability in a broad spectrum of potential therapeutic applications.”
- (3) In both the revised manuscript and supporting information. We have replaced “pacing” with “electrical stimulating” to describe the functionality of our device more accurately.

References

- Monteiro, L. M., Vasques-Nóvoa, F., Ferreira, L., Pinto-Do-ó, P. & Nascimento, D. S. Restoring heart function and electrical integrity: Closing the circuit. *npj Regen. Med.* **2**, 1–13 (2017).
- Cao, H., Kang, B. J., Lee, C. A., Shung, K. K. & Hsiai, T. K. Electrical and Mechanical Strategies to Enable Cardiac Repair and Regeneration. *IEEE Rev. Biomed. Eng.* **8**, 114–124 (2015).

Comment 25: In Figs 6 M, N, it was mentioned that changes were observed in S2 and S4. However, it seems that only changes in frequency are visible along the x-axis. Shouldn't the focus be on the resistance changes along the y-axis, which may represent actual strain changes?

Our response: We appreciate the reviewer's concern regarding the data presented in **Fig. 6M** and **Fig. 6N**. These figures are intended to illustrate the contractility patterns of the heart's chambers under both normal and myocardial infarction (MI) conditions, as detected by our strategically positioned strain sensors on the epicardial surfaces (**Fig. 6A** and **Fig. S41A&B**). The output from these sensors, closely related to the strain they experience, provides the mechanical behaviors of specific heart chambers.

Notably, the sensor S4, located on the left ventricle (LV), exhibits the largest amplitude of strain changes. This indicates the LV's exceptional myocardium strength, aligning with the physiological role of the LV in bearing the greatest burdens in circulating blood throughout the body's systemic circulation. Additionally, **Fig. 6I** offers a visual representation of the infarction area two weeks post-MI surgery, complementing the quantitative data provided by strain sensors. The decreased strain changes detected by sensors S2 (RV) and S4(LV) illustrate the impact of MI, characterized by a loss of contractile myocardium, a decrease in the force of myocardial contractility, and an altered heart rate.

Fig. R28. (A&B) Representative measurements of local cardiac contractions before **(A)** and after myocardial infarction **(B)** using a soft robotic thera-gripper wrapping onto a living mouse heart.

We have revised the figures to feature the resistance changes more prominently along the y-axis, ensuring a balanced emphasis on both the strain measurements and the associated frequency changes, offering a comprehensive view of our device's capability to monitor and diagnose cardiac conditions through advanced sensor integration, and demonstrating the potential of custom-engineered solutions in cardiac health.

Modification to the manuscript:

On page 24, in the revised manuscript, we updated **Fig. R28A&B** as **Fig. 6M&N**, respectively.

Comment 26: The paper lacks details on the system used to generate water flow simulating blood flow in the vascular system, including names of the equipment and the setting of flow rate. Information on the properties and thickness of the rubber used is also insufficient.

Our response: We appreciate the reviewer's concern regarding the details on our experimental setup for simulating blood flow in our study. As shown in **Fig. R29**, to mimic the arterial environment, we employed a silicone tube (Transparent Silicone Tube 4mm ID x 5mm OD, wall thickness ~0.5mm) with large stretchability and flexibility as an arterial artery. The pulsatile water flow (30~260 ml/min) is generated with two flow rate controllable pumps. Pump 1 maintains a constant flow rate to establish a baseline pressure, while Pump 2, connected to a solenoid valve, is regulated by a relay. This valve opens and closes periodically, replicating the pulsatile pressure of blood flow. The relay is further controlled with a pre-programmed microcontroller. All the parts for the setup were purchased from local vendors.

Fig. R29. In vitro setup for artificial artery model.

We have included the detailed information regarding the experiment setup in the revised manuscript and supporting information.

Modification to the manuscript:

(1) On page 47, in the revised supporting information, we added **Fig. R29** as **Fig. 36B**.

(2) On page 31, in the revised manuscript, we added, “*Measurement of biomimetic blood pressure: As shown in Fig. S36B*, to mimic the arterial environment, we employed a silicone tube (Transparent Silicone Tube 4mm ID x 5mm OD, wall thickness ~0.5mm) with large stretchability and flexibility as an arterial artery. The pulsatile water flow (30~260 ml/min) is generated with two flow rate controllable pumps. Pump 1 maintains a constant flow rate to establish a baseline pressure, while Pump 2, connected to a solenoid valve, is regulated by a relay. This valve opens and closes periodically, replicating the

pulsatile pressure of blood flow. The relay is further controlled with a pre-programmed microcontroller. All the parts for the setup were purchased from local vendors.”

Comment 27: This work validated through H&E staining data that a single-day stimulation has a minimal impact on the tissue. However, it remains unclear what effects may occur with continued stimulation over longer period.

Our response: We appreciate the reviewer’s concern regarding the long-term effects of continuous thermal stimulation. As previously discussed in Comments 12, 14, and 22, our device is engineered to operate around the body’s natural temperature, significantly reducing the reliance on constant thermal input. Moreover, we can design the device with an advanced thermal feedback mechanism that precisely regulates temperature, thus mitigating risks associated with overheating or insufficient heating (**Fig. R10&11**).

Furthermore, the existing literature supports that controlled thermal stimulation within safe limits typically does not lead to adverse effects even thermal input over extended periods.^{13,14} We have further explained the immediate scope of our current research phase. We focus on establishing a solid foundation for the device’s safety and functionality before extending our investigation into long-term usage scenarios.

Modification to the manuscript:

(1) On page 22, in the revised manuscript, we added, “The design employed PNIPAM hydrogel with a LCST 34 °C that is closely aligned with natural body temperature to achieve necessary shape deformation. The initial heating serves primarily to accelerate the actuation, but after achieving the desired state, continuous electrical heating becomes unnecessary. This feature allows the device to effectively adapt and function within the physiological temperature range without the need for ongoing thermal input.”

(2) On page 36, in the revised supporting information, we added **Fig. R10** as **Fig. S25**.

(3) On page 37, in the revised supporting information, we added **Fig. R11** as **Fig. S26**.

(4) On page 5, in the revised supporting information, we added,“

Supplementary Note S3: The sensory-motor integration within the soft robotic system

The integrated design that leverages on the synergistic interplay between sensors and soft robotic actuators enables volatile adaptivity and responsiveness of the robotic implant in ever-changing conditions. This design allows sensory inputs to directly influence actuator outputs in real-time, creating a feedback loop that allows for automatic adjustment based on environmental stimuli. This feature underscores our system’s responsiveness and adaptability, allowing with the principles of sensor-actuator integration.

As an example, we present an adaptively controlled responsive robotic gripper that maintains optimal actuation temperatures responsive to external temperature changes. The responsive self-adaptation in different thermal environments ensures optimal and

predictable heating patterns regardless of conditions, as well as increasing patient safety and preventing potential burn hazard caused by overheating. **Fig. S25A** shows an integrative system that provides temperature sensory readout and, through feedback-controlled power delivery, allows actuation responsive to external thermal environment. **Fig. S25B** demonstrates the working principles of controlling hardware and software. The resistive temperature sensor was connected in series with a reference resistor. The voltage drops over the sensor is read and converted by the on-chip ADC and is processed by the microcontroller to convert to a temperature value. Upon receipt of the temperature value, the control algorithm compares it with the target temperature to produce an error value. The error value is multiplied with a predefined coefficient to produce a corrective factor and is added to the current output value to generate a new output value. The on-chip pulse-width modulation (PWM) module converts the output value to a PWM voltage signal, which is then amplified by the on-board power MOSFET, forming a controller current source to produce a current through the heater. When the external temperature changes, a large corrective factor is produced to allow rapid adaptations. When the target temperature was reached, a constant current is produced to maintain the optimal temperature. **Fig. S26** showed the time-synchronized current and temperature plot during different responsive phases after a sudden change in temperature. During the underheating phase, a rapid increase in output current was generated by the controlled current source in response to the sudden change in temperature. When the target temperature was first reached, the current output remains constant until the actuator was overheated, during which the current output drops again to reach the static phase at which the current levels off.”

(5) On page 12, in the revised manuscript, we added, “Furthermore, our soft robotic system exemplifies advanced sensory-motor integration, leveraging the synergistic relationship between embedded sensors and actuators to achieve dynamic adaptivity and responsiveness to environmental changes. A prime example is a temperature-sensitive control system, as shown in **Fig. S25A**, which utilizes real-time sensory feedback to dynamically adjust heating in response to environmental temperature changes. The operational principle, as detailed in **Fig. S25B** and **Supplementary Note S3**, involves a microcontroller-driven algorithm that interprets temperature input collected by a resistive temperature sensor, and modulates the electric heater’s current accordingly, enabling rapid adaptations to achieve and maintain a preset temperature. **Fig. S26** presents a soft robotic finger’s real-time response to temperature variations, ensuring stable shape adaptation through this regulatory mechanism. Moreover, this intelligent control significantly improves safety by preventing the risk of overheating, thereby ensuring the system’s safe operation in various thermal conditions, highlighting our device’s ability to provide precise thermal management, enhancing both efficacy and safety in its applications.”

(6) On page 29, in the revised manuscript, we added, “

The sensory-motor integration within the soft robotic system

The robotic gripper and the external circuitry were connected in series with an NI DMM amperemeter set for DC current measurement. The device was cooled to 22 °C in ambient temperature before the system was started up to capture its response to a sudden decrease in ambient temperature. The temperature readouts recorded by the device's integrated sensor were logged via a microcontroller unit (MCU) and cross-referenced with data from FLIR thermal camera. Both the current and temperature data were analyzed using custom Python script designed specifically for this purpose."

References

G.C. van Rhoon, T. Samaras, P.S. Yarmolenko, M.W. Dewhirst, E. Neufeld, N. Kuster, CEM43°C thermal dose thresholds: a potential guide for magnetic resonance radiofrequency exposure levels?, *Eur. Radiol.* 23 (2013) 2215–2227. <https://doi.org/10.1007/s00330-013-2825-y>.

M.W. Dewhirst, B.L. Viglianti, M. Lora-Michiels, P.J. Hoopes, M. Hanson, THERMAL DOSE REQUIREMENT FOR TISSUE EFFECT: EXPERIMENTAL AND CLINICAL FINDINGS., *Proc. SPIE--the Int. Soc. Opt. Eng.* 4954 (2003) 37. <https://doi.org/10.1117/12.476637>.

Comment 28: The 2-week histological analysis of cardiac tissue with a hydrogel-based thera-gripper, as presented in the study, raises questions. It would be insightful to know the condition of the device after this period –whether the device remained in an operable state and the device remained properly attached as intended while withstanding repeated pulsations.

Our response: We appreciate the reviewer's concern regarding the long-term stability and functionality of the hydrogel-based gripper after a 2-week period. After this time, we carefully examined the device and found it to be intact and effectively attached to the cardiac tissue as intended, as shown in **Fig. R30A**.

We also conducted further evaluation focused on the device's performance post-implantation. As shown in **Fig. R30B-D**, the pacing electrodes and thermal sensor maintained exemplary performance, indicating their capability to deliver electrical stimulation and accurately sense temperature changes after two weeks. These results suggest that our hydrogel-based thera-gripper not only maintains its structural and functional integrity over an extended period but also holds promise for long-term applications in cardiac therapy and monitoring.

We have updated our manuscript and supporting information to include these insights and highlight the device's robustness and reliability for potential long-term use.

Fig. 30. Overview of device performance after two-week implantation. (A) Visualization of the device accurately positioned on the mouse heart, indicating its stability post two-week implantation. (B&C) Voltage response traces from the pacing electrodes embedded in the implanted device, demonstrating its operational integrity over the two-week period. Here, (B) shows a sine wave configuration at a frequency of 1 Hz and amplitude of 1 V, while (C) depicts pulse modulation with a height of 500 mV, pulse width of 0.01 s, and a maximum repeating rate of 1 Hz. (D) Comparative analysis of the resistive response from the thermal sensor before and after the two-week implantation period, illustrating the device's consistent performance and sensor integrity over time.

Modification to the manuscript:

- (1) On page 64, in the revised supporting information, we added **Fig. R30** as **Fig. S52**.
- (2) On page 24, in the revised manuscript, we added, “Our post-implantation evaluation revealed that the hydrogel-based thera-gripper remained intact and securely attached to cardiac tissue as intended, demonstrating its durability and effectiveness over time (**Fig. S52A**). Notably, the E-stim electrodes and thermal sensor maintained optimal performance, effectively delivering electrical stimulation (**Fig. S52B&C**) and precisely sensing temperature fluctuations (**Fig. S52D**) even after a two-week period. These results support its feasibility for long-term therapeutic and diagnostic applications.”

Comment 29: Caption in figure S11 (a) : AgNW/PI -> AgNW/PDMS

Our response: We thank the reviewer's attention to detail and have corrected the caption in Fig. S11(a) (Fig. S10A in the updated supporting information) from AgNW/PI to AgNW/PDMS as suggested.

Modification to the manuscript:

On page 20, in the revised manuscript, we have corrected "AgNW/PI" to "AgNW/PDMS".

Comment 30: Page 21, the third line from the bottom spelling error : PINPAM -> PNIPAM

Our response: We thank the reviewer's attention to detail and have corrected the spelling error.

Modification to the manuscript:

On page 22, in the revised manuscript, we have corrected "PINPAM" to "PNIPAM".

Comment 31: Figure S51D, indication spelling error : Basi -> Basic

Our response: We thank the reviewer's attention to detail and have corrected the spelling error.

Modification to the manuscript:

On page 48, in the revised manuscript, we have corrected "Basi" to "Basic".

Comment 32: There should be a space before the units in the text and figures.

Our response: We thank the reviewer's attention to detail.

Modification to the manuscript: We have checked and revised all units in the updated manuscript and supporting information.

References

1. Basarir, F., Madani, Z. & Vapaavuori, J. Recent Advances in Silver Nanowire Based Flexible Capacitive Pressure Sensors: From Structure, Fabrication to Emerging Applications. *Adv. Mater. Interfaces* **9**, (2022).
2. Chauhan, N., Maekawa, T. & Kumar, D. N. S. Graphene based biosensors - Accelerating medical diagnostics to new-dimensions. *J. Mater. Res.* **32**, 2860–2882 (2017).
3. Amara, U., Hussain, I., Ahmad, M., Mahmood, K. & Zhang, K. 2D MXene-Based Biosensing: A Review. *Small* **19**, 1–38 (2023).
4. Driscoll, N. *et al.* MXene-infused bioelectronic interfaces for multiscale electrophysiology and stimulation. *Sci. Transl. Med.* **13**, eabf8629 (2021).
5. Liu, H. *et al.* 3D Printed Flexible Strain Sensors: From Printing to Devices and Signals. *Adv. Mater.* **33**, 2004782 (2021).
6. Zhu, Z., Park, H. S. & McAlpine, M. C. 3D printed deformable sensors. *Sci. Adv.* **6**, eaba5575 (2023).

7. Reeder, J. *et al.* Mechanically Adaptive Organic Transistors for Implantable Electronics. *Adv. Mater.* **26**, 4967–4973 (2014).
8. Xu, J. & Song, J. Thermal Responsive Shape Memory Polymers for Biomedical Applications. in (ed. Fazel-Rezai, R.) Ch. 6 (IntechOpen, 2011). doi:10.5772/19256.
9. Czerner, M., Fellay, L. S., Suárez, M. P., Frontini, P. M. & Fasce, L. A. Determination of Elastic Modulus of Gelatin Gels by Indentation Experiments. *Procedia Mater. Sci.* **8**, 287–296 (2015).
10. Xie, C., Wang, X., He, H., Ding, Y. & Lu, X. Mussel-Inspired Hydrogels for Self-Adhesive Bioelectronics. *Adv. Funct. Mater.* **30**, 1909954 (2020).
11. Shian, S., Bertoldi, K. & Clarke, D. R. Dielectric Elastomer Based ‘grippers’ for Soft Robotics. *Adv. Mater.* **27**, 6814–6819 (2015).
12. Shojaeifard, M., Niroumandi, S. & Baghani, M. Programming shape-shifting of flat bilayers composed of tough hydrogels under transient swelling. *Acta Mech.* **233**, 213–232 (2022).
13. Dewhurst, M. W., Viglianti, B. L., Lora-Michiels, M., Hoopes, P. J. & Hanson, M. THERMAL DOSE REQUIREMENT FOR TISSUE EFFECT: EXPERIMENTAL AND CLINICAL FINDINGS. *Proc. SPIE--the Int. Soc. Opt. Eng.* **4954**, 37 (2003).
14. van Rhoon, G. C. *et al.* CEM43°C thermal dose thresholds: a potential guide for magnetic resonance radiofrequency exposure levels? *Eur. Radiol.* **23**, 2215–2227 (2013).
15. Jiang, Y. *et al.* Wireless, closed-loop, smart bandage with integrated sensors and stimulators for advanced wound care and accelerated healing. *Nat. Biotechnol.* **41**, 652–662 (2023).
16. Grill, W. M. Electrical stimulation for control of bladder function. *Proc. 31st Annu. Int. Conf. IEEE Eng. Med. Biol. Soc. Eng. Futur. Biomed. EMBC 2009* 2369–2370 (2009) doi:10.1109/IEMBS.2009.5335001.
17. Coolen, R. L., Groen, J. & Blok, B. F. M. Electrical stimulation in the treatment of bladder dysfunction: Technology update. *Med. Devices Evid. Res.* **12**, 337–345 (2019).
18. Lanzalaco, S., Mingot, J., Torras, J., Alemán, C. & Armelin, E. Recent Advances in Poly(N-isopropylacrylamide) Hydrogels and Derivatives as Promising Materials for Biomedical and Engineering Emerging Applications. *Adv. Eng. Mater.* **25**, 2201303 (2023).
19. Liu, J., Jiang, L., He, S., Zhang, J. & Shao, W. Recent progress in PNIPAM-based multi-responsive actuators: A mini-review. *Chem. Eng. J.* **433**, 133496 (2022).
20. Ansari, M. J. *et al.* Poly(N-isopropylacrylamide)-Based Hydrogels for Biomedical Applications: A Review of the State-of-the-Art. *Gels (Basel, Switzerland)* **8**, (2022).
21. Monteiro, L. M., Vasques-Nóvoa, F., Ferreira, L., Pinto-Do-ó, P. & Nascimento, D.

- S. Restoring heart function and electrical integrity: Closing the circuit. *npj Regen. Med.* **2**, 1–13 (2017).
22. Cao, H., Kang, B. J., Lee, C. A., Shung, K. K. & Hsiai, T. K. Electrical and Mechanical Strategies to Enable Cardiac Repair and Regeneration. *IEEE Rev. Biomed. Eng.* **8**, 114–124 (2015).

REVIEWERS' COMMENTS

Reviewer #1 (Remarks to the Author):

I believe that the revisions address the points that were raised in the original review. The paper is now suitable to be accepted in nature communications.

Reviewer #2 (Remarks to the Author):

I thank the authors for fully and thoroughly revising the manuscript. I believe their manuscript has improved significantly, and I have no further comments.

Responses to comments of Referee #1

Summary Comment: I believe that the revisions address the points that were raised in the original review. The paper is now suitable to be accepted in nature communications.

Our response: We thank the reviewer for the positive comment.

Responses to comments of Referee #2

Summary Comment: I thank the authors for fully and thoroughly revising the manuscript. I believe their manuscript has improved significantly, and I have no further comments.

Our response: We thank the reviewer for the positive comment.

Description of Additional Supplementary Files

File name: Supplementary movie 1

Description: Soft robot with a nature-inspired starfish design

File name: Supplementary movie 2

Description: Shape transformation of soft robots with various configurations

File name: Supplementary movie 3

Description: Soft robot with a helical structure biomimicking a chiral seedpod

File name: Supplementary movie 4

Description: The shape transformation of a soft robotic pill

File name: Supplementary movie 5

Description: A three-arm soft robotic gripper via sequentially programming input power

File name: Supplementary movie 6

Description: A four-arm soft robotic gripper via simultaneously programming input power